# Effect of Activation Functions on the Training of Overparametrized Neural Nets

**Abhishek Panigrahi**
Microsoft Research India
t-abpani@microsoft.com

**Abhishek Shetty** *
Cornell University
shetty@cs.cornell.edu

**Navin Goyal**
Microsoft Research India
navingo@microsoft.com

## Abstract

It is well-known that overparametrized neural networks trained using gradient-based methods quickly achieve small training error with appropriate hyperparameter settings. Recent papers have proved this statement theoretically for highly overparametrized networks under reasonable assumptions. These results either assume that the activation function is ReLU or they depend on the minimum eigenvalue of a certain Gram matrix. In the latter case, existing works only prove that this minimum eigenvalue is non-zero and do not provide quantitative bounds which require that this eigenvalue be large. Empirically, a number of alternative activation functions have been proposed which tend to perform better than ReLU at least in some settings but no clear understanding has emerged. This state of affairs underscores the importance of theoretically understanding the impact of activation functions on training.

In the present paper, we provide theoretical results about the effect of activation function on the training of highly overparametrized 2-layer neural networks. A crucial property that governs the performance of an activation is whether or not it is smooth:

- For non-smooth activations such as ReLU, SELU, ELU, which are not smooth because there is a point where either the first order or second order derivative is discontinuous, all eigenvalues of the associated Gram matrix are large under minimal assumptions on the data.
- For smooth activations such as tanh, swish, polynomial, which have derivatives of all orders at all points, the situation is more complex: if the subspace spanned by the data has small dimension then the minimum eigenvalue of the Gram matrix can be small leading to slow training. But if the dimension is large and the data satisfies another mild condition, then the eigenvalues are large. If we allow deep networks, then the small data dimension is not a limitation provided that the depth is sufficient.

We discuss a number of extensions and applications of these results.

## 1 Introduction

It is now well-known that overparametrized feedforward neural networks trained using gradient-based algorithms with appropriate hyperparameter choices reliably achieve near-zero training error, e.g., Neyshabur et al. (2015). Importantly, overparametrization also often helps with generalization; but our central concern here is the training error which is an important component in understanding generalization. We study the effect of the choice of activation function (we often just say activation) on the training of overparametrized neural networks. By overparametrized setting we roughly mean that the number of parameters or weights in the networks is much larger than the number of data samples.

The well-known universal approximation theorem for feedforward neural networks states that any continuous function on a bounded domain can be approximated arbitrarily well by a finite neural network with one hidden layer. This theorem is generally stated for specific activation functions such as sigmoid or ReLU. A more general form of the theorem shows this for essentially all non-polynomial activations (Leshno et al., 1993; Pinkus, 1999; Sonoda & Murata, 2017). This theorem

---

*Work done when the author was at Microsoft Research India.

concerns only the expressive power and does not address how the training and generalization of neural networks are affected by the choice of activation, nor does it provide quantitative information about the size of the network needed for the task.

Traditionally, sigmoid and tanh had been the popular activations but a number of other activations have also been considered including linear and polynomial activations (Arora et al., 2019a; Du & Lee, 2018; Kileel et al., 2019). One of the many innovations in the resurgence of neural networks in the last decade or so has been the realization that ReLU activation generally performs better than the traditional choices in terms of training and generalization. ReLU is now the de facto standard for activation functions for neural networks but many other activations are also used which may be advantageous depending on the situation (e.g. (Goodfellow et al., 2016, Chapter 6)). In practice, most activation functions often achieve reasonable performance. To quote Goodfellow et al. (2016): *In general, a wide variety of differentiable functions perform perfectly well. Many unpublished activation functions perform just as well as the popular ones.* Concretely, Ramachandran et al. (2018) provides a list of ten non-standard functions which all perform close to the state of the art at some tasks. This hints at the possibility of a universality phenomenon for training neural networks similar to the one for expressive power mentioned above.

**Search for activation functions.** A number of recent papers have proposed new activations—such as ELU, SELU, penalized tanh, SiLU/swish—based on either theoretical considerations or automated search using reinforcement learning and other methods; e.g. Clevert et al. (2016); Klambauer et al. (2017); Xu et al. (2016); Elfwing et al. (2017); Ramachandran et al. (2018). For definitions, see Section 2 and Appendix B. These activation functions have been found to be superior to ReLU in many settings. See e.g. Eger et al. (2018); Nwankpa et al. (2018) for overview and evaluation. We quote once more from Goodfellow et al. (2016): *The design of hidden units is an extremely active area of research and does not yet have many definitive guiding theoretical principles.*

**Theoretical analysis of training of highly overparametrized neural networks.** Theoretical analysis of neural network training has seen vigorous activity of late and significant progress was made for the case of highly overparametrized networks. At a high level, the main insight in these works is that when the network is large, small changes in weights can already allow the network to fit the data. And yet, since the weights change very little, the training dynamics approximately behaves as in kernel methods and hence can be analyzed (e.g. Jacot et al. (2018); Li & Liang (2018); Du et al. (2019a); Allen-Zhu et al. (2019); Du et al. (2019b); Allen-Zhu et al. (2018); Arora et al. (2019c); Oymak & Soltanolkotabi (2019)). There are also many other approaches for theoretical analysis, e.g. Brutzkus et al. (2018); Mei et al. (2018); Chizat & Bach (2018). Because of the large number of papers on this topic, we have chosen to list only the most closely related ones.

Analyses in many of these papers involve a matrix $\mathbf{G}$, either explicitly (Jacot et al., 2018; Du et al., 2019a;b) or implicitly (Allen-Zhu et al., 2019). (This matrix also occurs in earlier works (Xie et al., 2017; Tsuchida et al., 2018).) $\lambda_{\min}(\mathbf{G})$, the minimum eigenvalue of $\mathbf{G}$, is an important parameter that directly controls the rate of convergence of gradient descent training: the higher the minimum eigenvalue the faster the convergence. Jacot et al. (2018); Du et al. (2019b) show that $\lambda_{\min}(\mathbf{G}) > 0$ for certain activations assuming that no two data points are parallel. Unfortunately, these results do not provide any quantitative information. The result of Allen-Zhu et al. (2019), where the matrix $\mathbf{G}$ does not occur explicitly, can be interpreted as showing that the minimum eigenvalue is large under the reasonable assumption that the data is $\delta$-separated, meaning roughly that no two data points are very close, and the activation used is ReLU. This quantitative lower bound on the minimum eigenvalue implies fast convergence of training. So far, ReLU was the only activation for which such a proof was known.

**Our results in brief.** A general result one could hope for is that based on general characteristics of the activations, such as smoothness, convexity etc., one can determine whether the smallest eigenvalue of $\mathbf{G}$ is small or large. We prove results of this type. A crucial distinction turns out to be whether the activation is (a) not smooth (informally, has a "kink") or (b) is smooth (i.e. derivatives of all orders exist over $\mathbb{R}$). The two classes of functions above seem to cover all "reasonable" activations; in particular, to our knowledge, all functions used in practice seem to fall in one of the two classes above.

- Activations with a kink, i.e., those with a a jump discontinuity in the derivative of some constant order, have all eigenvalues large under minimal assumptions on the data. E.g., the first derivatives of ReLU and SELU, the second derivative of ELU have jump discontinuities at $0$. These results

imply that for such activations, training will be rapid. We also provide a new proof for ReLU with the best known lower bound on the minimum eigenvalue.

- For smooth activations such as polynomials, tanh and swish the situation is more complex: the minimum eigenvalue can be small depending on the dimension of the span of the dataset. We give a few examples: Let $n$ be size of the dataset which is a collection of points in $\mathbb{R}^d$, and let $d'$ be the dimension of the span of the dataset. For quadratic activation, if $d' = O(\sqrt{n})$, then the minimum singular value is 0. For tanh and swish, if $d' = O(\log^{0.75} n)$, then the minimum eigenvalue is inverse superpolynomially small ($\exp(-\Omega(n^{1/2d'}))$). In fact, a significant fraction of eigenvalues can be small. This implies that for such datasets, training using smooth activations will be slow if the dimension of the dataset is small. A trade off is possible: assuming stronger bounds on the dimension of the span gives stronger bounds on the eigenvalues. We also show that these results are tight in a precise sense. We further show that the above limitation of smooth activations disappears if one allows sufficiently deep networks.

Unless otherwise stated, we work with one hidden layer neural nets where only the input layer is trained. This choice is made to simplify exposition; extensions of various types including the ones that drop the above restriction are possible and discussed in Section 5.

## 2    PRELIMINARIES

Denote the unit sphere in $\mathbb{R}^n$ by $\mathbb{S}^{n-1} := \left\{ \mathbf{u} \in \mathbb{R}^n : \|\mathbf{u}\|_2 = 1 \right\}$ where $\|\mathbf{u}\|^2 := \|\mathbf{u}\|_2^2 := \sum_{i=1}^n u_i^2$. For $u, v \in \mathbb{R}^n$, define the standard inner product by $\langle u, v \rangle := \sum_i u_i v_i$. Given a set $S$, denote by $\mathcal{U}(S)$ the uniform distribution over $S$. $\mathcal{N}(\mu, \sigma^2)$ denotes the univariate normal distribution with mean $\mu$ and variance $\sigma^2$. Let $\mathcal{I}_S$ denote the indicator of the set $S$. For a matrix $\mathbf{A}$ containing elements $a_{ij}$, $\mathbf{a}_i$ denotes its $i$-th column. We define some of the popular non-traditional activation functions here: $\mathsf{swish}(x) := \frac{x}{1+e^{-x}}$ (Ramachandran et al., 2018) (called SiLU in Elfwing et al. (2017)); $\mathsf{ELU}(x) := \max(x, 0) + \min(x, 0)(e^x - 1)$ (Clevert et al., 2016); $\mathsf{SELU}(x) := \alpha_1 \max(x, 0) + \alpha_2 \min(x, 0)(e^x - 1)$, where $\alpha_1$ and $\alpha_2$ are two different constants (Klambauer et al., 2017). See Appendix B for more definitions.

We consider 2-layer neural networks

$$F(\mathbf{x}; \mathbf{a}, \mathbf{W}) := \frac{c_\phi}{\sqrt{m}} \sum_{k=1}^m a_k \phi\left(\mathbf{w}_k^T \mathbf{x}\right), \tag{1}$$

where $\mathbf{x} \in \mathbb{R}^d$ is the input and $\mathbf{W} = [\mathbf{w}_1, \ldots, \mathbf{w}_m] \in \mathbb{R}^{m \times d}$ is the hidden layer weight matrix and $\mathbf{a} \in \mathbb{R}^m$ is the output layer weight vector. Activation function $\phi : \mathbb{R} \to \mathbb{R}$ acts entrywise on vectors and $c_\phi^2 := \mathbb{E}_{z \in \mathcal{N}(0,1)} \phi(z)^2$. In the case of one hidden layer nets, we set $c_\phi = 1$ to simplify expressions; this is without loss of generality. For deeper networks we do not do this as this assumption would result in loss of generality. Elements of $\mathbf{W}$ and $\mathbf{a}$ have been initialized i.i.d. from the standard Gaussian distribution. This initialization and the parametrization in Equation 1 are from Du et al. (2019b). Together, these will be referred to as the DZPS setting. Parametrization in practice does not have $\frac{c_\phi}{\sqrt{m}}$ in Equation 1; standard initializations in practice include He (He et al., 2015) and Glorot initializations (Glorot & Bengio, 2010) and variants. In the DZPS setting, the argument of the activation can be larger compared to the standard initializations, making the analysis harder. Our theorems apply to the DZPS setting as well as to standard initializations and for the latter easily follow as corollaries to the analysis in the DZPS setting. We defer this discussion to the appendix. Unless otherwise stated, we work in the DZPS setting with one hidden layer neural nets with initialization as above with only the input layer trained.

Given labeled input data $\{(\mathbf{x}_i, y_i)\}_{i=1}^n$, where $\mathbf{x}_i \in \mathbb{R}^d$ and $y_i \in \mathbb{R}$, we want to find the best fit weights $\mathbf{W}$ so that the quadratic loss $\mathcal{L}(\{(\mathbf{x}_i, y_i)\}_{i=1}^n; \mathbf{a}, \mathbf{W}) := \frac{1}{2} \sum_{i=1}^n (y_i - F(\mathbf{x}_i; \mathbf{a}, \mathbf{W}))^2$ is minimized. We train the neural network by fixing the output weight vector $\mathbf{a}$ at random initialization and training the hidden layer weight matrix $\mathbf{W}$ (output layer being trained can be easily handled as in Du et al. (2019a;b); See Appendix M for details). In this paper, we focus on the gradient descent algorithm for this purpose. The gradient descent update rule for $\mathbf{W}$ is given by $\mathbf{W}^{(t+1)} := \mathbf{W}^{(t)} - \eta \nabla_{\mathbf{W}} \mathcal{L}(\mathbf{W}^{(t)})$, where $\mathbf{W}^{(t)}$ denotes the weight matrix after $t$ steps of gradient descent and $\eta > 0$ is the learning rate. The output vector $\mathbf{u}^{(t)} \in \mathbb{R}^n$ is defined by $u_i^{(t)} := F(\mathbf{x}_i; \mathbf{a}, \mathbf{W}^{(t)})$.

Next, we define the matrix alluded to earlier, the *Gradient Gram matrix* $\mathbf{G} \in \mathbb{R}^{n \times n}$ associated with the neural network defined in (1), referred to as the $G$-matrix in the sequel:

$$g_{i,j} := \frac{1}{m} \sum_{k \in [m]} a_k^2 \, \phi'(\mathbf{w}_k^T \mathbf{x}_i) \phi'(\mathbf{w}_k^T \mathbf{x}_j) \langle \mathbf{x}_i, \mathbf{x}_j \rangle. \tag{2}$$

We will often work with the related matrix $\mathbf{M} \in \mathbb{R}^{md \times n}$, whose $i$-th column is obtained by vectorizing $\nabla_{\mathbf{W}} \mathcal{F}(\mathbf{x}_i, y_i)$, i.e., $\mathbf{M}_{d(k-1)+1:dk,i} := a_k \phi'\left(\mathbf{w}_k^T \mathbf{x}_i\right) \mathbf{x}_i$ for $k \in [m]$. $\mathbf{G}$ is a Gram matrix: $\mathbf{G} = \frac{1}{m} \mathbf{M}^T \mathbf{M}$. Denote by $\lambda_i(\mathbf{G})$ the $i$-th eigenvalue of $\mathbf{G}$ with $\lambda_1 \geq \lambda_2 \geq \ldots$, and similarly by $\sigma_i(\mathbf{M})$ the $i$-th singular value of $\mathbf{M}$. These quantities are related by $\lambda_i(\mathbf{G}) = \frac{1}{m} \sigma_i^2(\mathbf{M})$.

Following Allen-Zhu et al. (2019), we make the following mild assumptions on data.

**Assumption 1.** $\|\mathbf{x}_i\|_2 = 1 \quad \forall i \in [n]$.

**Assumption 2.** $\|(\mathbf{I}_n - \mathbf{x}_i \mathbf{x}_i^T) \mathbf{x}_j\|_2 \geq \delta \quad \forall i, j \in [n], i \neq j$ *i.e., the distance between the subspaces spanned by $\mathbf{x}_i$ and $\mathbf{x}_j$ is at least $\delta > 0$.*

Assumption 1 can be easily satisfied by the following preprocessing: renormalize each $\mathbf{x}_i$ so that $\|\mathbf{x}_i\| \leq 1/\sqrt{2}$, add another coordinate to each $\mathbf{x}_i$ so that $\|\mathbf{x}_i\| = 1/\sqrt{2}$ and then add another coordinate with value $1/\sqrt{2}$ to each $\mathbf{x}_i$. This ensures that $\|\mathbf{x}_i - \mathbf{x}_j\|_2 \geq \delta$ implies Assumption 2 for $\mathbf{x}_i, \mathbf{x}_j$, which we later verify empirically for CIFAR10.

## 3 REVIEW OF RELEVANT PRIOR WORK

To motivate the importance of the $G$-matrix, let us first consider the continuous time gradient flow dynamics $\dot{\mathbf{W}}^{(t)} = -\nabla_{\mathbf{W}} \mathcal{L}(\mathbf{W}^{(t)})$, where $\mathcal{L}(\mathbf{W})$ denotes the loss function (in the notation we suppressed dependence on data and the weights of the output layer) and $\dot{\mathbf{W}}$ denotes the derivative with respect to $t$. Let $\mathbf{y} \in \mathbb{R}^n$ be the vector of outputs. It follows from an application of the chain rule that $\dot{\mathbf{u}}^{(t)} = \mathbf{G}^{(t)}(\mathbf{y} - \mathbf{u}^{(t)})$. Here $g_{i,j}^{(t)} := \frac{1}{m} \left\langle \mathbf{x}_i, \mathbf{x}_j \right\rangle \sum_{k=1}^m \phi'(\mathbf{w}_k^{(t)T} \mathbf{x}_j) \phi'(\mathbf{w}_k^{(t)T} \mathbf{x}_i)$. It can be shown that as $m \to \infty$, the matrix $\mathbf{G}^{(t)}$ remains close to its initial value $\mathbf{G}^{(0)}$ which is exactly the $G$-matrix (see e.g. Jacot et al. (2018); Arora et al. (2019b) for closely related results). This leads us to the approximate solution, which upon diagonalizing the PSD matrix $\mathbf{G}^{(0)}$ is given by

$$\mathbf{y} - \mathbf{u}^{(t)} = \sum_{i \in [n]} (e^{-\lambda_i t} \mathbf{v}_i \mathbf{v}_i^T)(\mathbf{y} - \mathbf{u}^{(0)}). \tag{3}$$

Thus, it can be seen that the eigenvalues of the $G$-matrix, in particular $\lambda_{\min}(\mathbf{G}^{(0)})$, control the rate of change of the output of the neural network towards the true labels. The following result plays a central role in the present paper.

**Theorem 3.1** (Theorem 4.1 of Du et al. (2019a)). *Let $\phi$ be* ReLU. *Define matrix $\mathbf{G}^\infty$ as $\mathbf{G}^\infty := \mathbb{E}_{\mathbf{w} \sim \mathcal{N}(\mathbf{0}, \mathbf{I}_d)} \mathcal{I}_{\mathbf{w}^T \mathbf{x}_i \geq 0, \mathbf{w}^T \mathbf{x}_j \geq 0} \left\langle \mathbf{x}_i, \mathbf{x}_j \right\rangle$. Assume $\lambda = \lambda_{\min}(\mathbf{G}^\infty) > 0$. If we set $m \geq \Omega\left(n^6 \lambda^{-4} \kappa^{-3}\right)$ and $\eta \leq \mathcal{O}\left(\lambda n^{-3}\right)$ and initialize $\mathbf{w}_k \sim \mathcal{N}(0, \mathbf{I}_d)$ and $a_k \sim \mathcal{U}\{-1, +1\}$ for $k \in [m]$, then with probability at least $1 - \kappa$ over the random initialization, for $t \geq 1$ we have $\|\mathbf{y} - \mathbf{u}^{(t)}\|_2^2 \leq (1 - 0.5\eta\lambda)^t \|\mathbf{y} - \mathbf{u}^{(0)}\|_2^2$.*

In the theorem above it can be seen that the time required to reach a desired amount of error is inversely proportional to $\lambda$. Du et al. (2019b) extended the above result to general real-analytic functions. While the definition of the $G$-matrix shows that it is positive semidefinite, it is not immediately clear that the matrix is non-singular. But the following theorem, from Du et al. (2019b) says that the matrix is indeed non-singular under very weak assumptions on the data and activation function. A similar result for the limit $m \to \infty$ but for more general non-polynomial Lipschitz activations was shown in Jacot et al. (2018) using techniques from Daniely et al. (2016).

**Lemma 3.2** (Lemma F.1 in Du et al. (2019b)). *If $\phi$ is a non-polynomial analytic function and $\mathbf{x}_i$ and $\mathbf{x}_j$ are not parallel for distinct $i, j \in [n]$, then $\lambda_{\min}(\mathbf{G}^\infty) > 0$.*

In these papers the number of neurons required and the rate of convergence depend on $\lambda_{\min}(\mathbf{G}^\infty)$ (e.g. Theorem 3.1 above) and thus it is necessary for the matrix to have large minimum singular value for their analysis to give useful quantitative bounds. Unfortunately, these papers do not provide quantitative lower bound for $\lambda_{\min}(\mathbf{G}^\infty)$.

Allen-Zhu et al. (2019) considered $L$-layer networks using ReLU (see Appendix D for details on their parametrization). A major step in their analysis is a lower bound on the gradient norm at each step.

**Theorem 3.3** (Theorem 3 in Allen-Zhu et al. (2019)). *With probability at least* $1 - \exp(-\Omega(m/\mathrm{poly}(n, \frac{1}{\delta})))$ *with respect to the initialization, for every* $\mathbf{W}$ *such that* $\|\mathbf{W} - \mathbf{W}^{(0)}\|_F \leq 1/\mathrm{poly}(n, \delta^{-1})$, *we have* $\|\nabla_{\mathbf{W}}\mathcal{L}(\mathbf{W})\|_F^2 \geq \Omega(\mathcal{L}(\mathbf{W})\,m\delta d^{-1}n^{-2})$.

We show that $\lambda_{\min}(\mathbf{G})$ is directly related to the lower bound on the gradient in the case of ReLU. It is not clear if the same method can be extended to other activation functions.

With this in mind, we aim to characterize the minimum eigenvalue of Gram matrix $\mathbf{G}^{\infty}$. Since, $\mathbf{G}^{\infty}$ is the same matrix as $\mathbf{G}^{(0)}$ in the limit $m \to \infty$, we will focus on proving high probability bounds for eigenvalues of $\mathbf{G}^{(0)}$.

## 4 MAIN RESULTS

### 4.1 ACTIVATIONS WITH A KINK

For any positive integer constant $r$, presence of a jump discontinuity in the $r$-th derivative of the activation leads to a large lower bound on the minimum eigenvalue. The activation function has the form $\phi(x) = \phi_1(x)\,\mathcal{I}_{x < \alpha} + \phi_2(x)\,\mathcal{I}_{x \geq \alpha}$, where $-1 \leq \alpha \leq 1$. Recall that $C^{r+1}$ denotes the set of $r + 1$ times continuously differentiable functions. We need $\phi_1$ and $\phi_2$ to satisfy the following set of conditions parametrized by $r$ and denoted $\mathbf{J_r}$:

- $\phi_1, \phi_2 \in C^{r+1}$ in the domains $(-\infty, \alpha]$ and $[\alpha, \infty)$, respectively.
- The first $(r + 1)$ derivatives of $\phi_1$ and $\phi_2$ are upper bounded in magnitude by 1 in $(-\infty, \alpha]$ and $[\alpha, \infty)$ respectively.
- For $0 \leq i < r$, we have $\phi_1^{(i)}(\alpha) = \phi_2^{(i)}(\alpha)$.
- $|\phi_1^{(r)}(\alpha) - \phi_2^{(r)}(\alpha)| = 1$, i.e., the $r$-th derivative has a jump discontinuity at $\alpha$.

*Remark.* We fix the constants in $\mathbf{J_r}$ to 1 for simplicity. We could easily parameterize these constants and make explicit the dependence of our bounds on these parameters. The requirement on the boundedness of derivatives is also not essential and can be relaxed as "all the action happens" in the interval $[-O(\sqrt{\log m}), O(\sqrt{\log m})]$.

$\mathbf{J_1}$ covers activations such as ReLU, SELU and LReLU, while $\mathbf{J_2}$ covers activations such as ELU. Below we state the bound explicitly for $\mathbf{J_2}$. Similar results hold for $\mathbf{J_r}$ for $r \geq 1$. See Section K for details.

**Theorem 4.1** ($\mathbf{J_2}$ activations). *: If the activation $\phi$ satisfies $\mathbf{J_2}$ then we have*

$$\lambda_{\min}(\mathbf{G}^{(0)}) \geq \Omega(\delta^3 n^{-7}(\log n)^{-1}),$$

*with probability at least* $1 - e^{-\Omega(\delta m/n^2)}$ *with respect to* $\{\mathbf{w}_k^{(0)}\}_{k=1}^m$ *and* $\{a_k^{(0)}\}_{k=1}^m$, *given that* $m > \max(\Omega(n^3\delta^{-1}\log(n\delta^{-1})), \Omega(n^2\delta^{-1}\log d))$.

The theorem above shows that the presence of a jump discontinuity in the derivative of activation function (or one of its higher derivatives) leads to fast training of the neural network. For the special case of ReLU we give a new proof. To our knowledge, lower bound on the minimum eigenvalue of the $G$-matrix below is the best known. The proof technique uses Hermite polynomials and is motivated by our results for smooth activations in the next section. See Section L for details.

**Theorem 4.2.** *If the activation is* ReLU *and* $m \geq \tilde{\Omega}(n^4\delta^{-3}\log^4 n)$, *then* $\lambda_{\min}(\mathbf{G}^{(0)}) \geq \Omega((\delta^{1.5}\log^{-1.5} n)$, *with probability at least* $1 - e^{-\tilde{\Omega}(m\delta^3 n^{-2}\log^{-3} n)}$.

The dependence on $n$ in the above bound is inverse-polylogarithmic as opposed to inverse-polynomial that seems to result from using the technique of Allen-Zhu et al. (2019). It implies that with $m = \tilde{\Omega}(n^6/\delta^6)$ in $\mathrm{poly}(\log(n/\epsilon), 1/\delta)$ steps gradient descent training achieves error less than $\epsilon$.

### 4.2 SMOOTH ACTIVATIONS

In contrast to activations with a kink, the situation is more complex for smooth activations and we can divide the results into positive and negative.

**Negative results for smooth activations.** The $G$-matrix of constant degree polynomial activations, such as quadratic activation, must have many zero eigenvalues; and of sufficiently smooth activations, such as tanh or swish, must have many small eigenvalues, if the dimension of the span of data is sufficiently small:

**Theorem 4.3** (restatement of Theorem F.2). *Let the activation be a degree-$p$ polynomial such that $\phi'(x) = \sum_{l=0}^{p-1} c_\ell x^\ell$ and let $d' = \dim(\text{span}\{\mathbf{x}_1 \ldots \mathbf{x}_n\}) = \mathcal{O}(n^{1/p})$. Then we have $\lambda_{\min}(\mathbf{G}^{(0)}) = 0$. Furthermore, $\lambda_k = 0$, for $k \geq \lceil n/d' \rceil$.*

**Theorem 4.4** (restatement of Theorem F.10). *Let the activation function be tanh and let $d' = \dim(\text{span}\{\mathbf{x}_1 \ldots \mathbf{x}_n\}) = \mathcal{O}(\log^{0.75} n)$. Then we have $\lambda_{\min}(\mathbf{G}^{(0)}) \leq \exp(-\Omega(n^{1/2d'}))$, with probability at least $1 - 1/n^{3.5}$ over the random choice of weight vectors $\{\mathbf{w}_k^{(0)}\}_{k=1}^m$ and $\{a_k^{(0)}\}_{k=1}^m$. Furthermore, the same upper bound is satisfied by $\lambda_k$, for $k \geq \lceil n/d' \rceil$.*

See Appendix F for proofs. Note that the bounds above do not make any assumption on the data other than the dimension of its span. The proof technique generalizes to give similar results for general classes of smooth activation such as swish (Section 6 and Appendix H). In contrast to the above result, the average eigenvalue of the $G$-matrix for all reasonable activation functions is large:

**Theorem 4.5** (informal version of Theorem E.1). *Let $\phi$ be a non-constant activation function, with Lipschitz constant $\alpha$ and let $\mathbf{G}$ be its $G$-matrix. Then, $\text{tr}(\mathbf{G}) = \Omega(n)$ with high probability.*

The previous two theorems together imply that the $G$-matrix is poorly conditioned when $d = \mathcal{O}(\log^{0.75} n)$ and the activation function is smooth, e.g., tanh. The effect on training of $G$-matrix being poorly conditioned can be easily seen in Equation 3 for the $m \to \infty$ case with gradient flow discussed earlier. For the finite $m$ setting, we show that the technique of Arora et al. (2019c) can be extended to the setting of smooth functions (see Appendix M.4) to prove the following.

**Theorem 4.6.** *Denote by $\mathbf{v}_i$ the eigenvectors of $\mathbf{G}^{(0)}$ and with $\lambda_i$ the corresponding eigenvalue. With probability at least $1 - \kappa$ over the random initialization, the following holds for $t \geq 0$, $\|\mathbf{y}-\mathbf{u}^{(t)}\|_2 \leq (\sum_{i=1}^n (1-\eta\lambda_i)^{2t}(\mathbf{v}_i^T(\mathbf{y}-\mathbf{u}^{(0)}))^2)^{1/2}+\epsilon$, provided $m \geq \Omega(n^5\kappa^{-1}\lambda_{\min}(\mathbf{G}^{(0)})^{-4}\epsilon^{-2})$ and $\eta \leq \mathcal{O}(n^{-2}\lambda_{\min}(\mathbf{G}^{(0)}))$.*

This result can be interpreted to mean that in the small perturbative regime of Du et al. (2019a); Arora et al. (2019c), smooth functions like tanh do not train fast. The learning rate in the above result is small as $\lambda_{\min}(\mathbf{G}^{(0)})$ is small. Analyzing the training for higher learning rates remains open.

**Positive results for smooth activations.** We show that in a certain sense the results of Theorem 4.3 and Theorem 4.4 are tight. Let us illustrate this for Theorem 4.4. Suppose that the activation function is tanh and that the dimension of the span $V$ of the data $\mathbf{x}_1, \ldots, \mathbf{x}_n$ is $\Omega(n^\gamma)$, for a constant $\gamma$. Furthermore, we assume that the data is *smoothed* in the following sense. We start with a preliminary dataset $\mathbf{x}_1', \ldots, \mathbf{x}_n'$ with the same span $V$, then we perturb each data point by adding i.i.d. Gaussian noise, i.e. $\mathbf{x}_i = \mathbf{x}_i' + \mathbf{n}_i$, and normalize to have unit Euclidean norm (see Assumption 3 for a precise statement). This Gaussian noise is obtained by taking the standard Gaussian variable on $V$ multiplied by a small factor. Thus the new data points have span $V$. For such datasets we show the following theorem. For more general theorems for polynomial activations and tanh, see Theorem I.3 and Theorem I.4 respectively.

**Theorem 4.7** (Informal version of Corollary I.4.2). *Let the activation be tanh, let the perturbation noise be of the order $(\delta/\sqrt{n})$ and $d' = \dim \text{span}\{\mathbf{x}_1, \ldots, \mathbf{x}_n\} \geq \Omega(n^\gamma)$, for a constant $\gamma$. Then we have $\lambda_{\min}(\mathbf{G}^{(0)}) \gtrsim \Omega(\delta^{(2/\gamma)}n^{-(3/\gamma)})$ with probability at least 0.99 w.r.t. the noise matrix $\mathbf{N}$, $\{\mathbf{w}_k^{(0)}\}_{k=1}^m$ and $\{a_k^{(0)}\}_{k=1}^m$, provided $m \gtrsim \tilde{\Omega}(n^{6/\gamma}\delta^{-4/\gamma})$.*

We now say a few words about our assumption that the data is smoothed. Smoothed analysis, originating from Spielman & Teng (2004), is a general methodology for analyzing efficiency of algorithms (often those that work well in practice) and can be thought of as a hybrid between worst-case and average-case analysis. Since in nature, problem instances are often subject to numerical and observational noise, one can model them by the process of smoothing. Smoothed analysis involves analyzing the performance of the algorithm on smoothed instances, which can be substantially better than the worst-case instances. Smoothed analysis has also been used in learning theory and our proof is inspired by Anderson et al. (2014) addressing a different problem, namely rank-1 decomposition

of tensors. In the present case, smoothness of the data rules out situations where the data has span $d'$ in a *non-robust* sense: most of the data points lie in small dimensional subspace and the dimension of the span is high because of a few points. Some such condition seems essential.

In another direction, we show that if the network is sufficiently deep for tanh, only the separability assumption on the data suffices. For deep networks, Du et al. (2019b) generalized the notion of $G$-matrix to be the $G$-matrix for the penultimate layer (see Section J). This matrix and its eigenvalues play similar role in the dynamics of training as for the one-hidden layer case discussed before; we continue to denote this matrix by $\mathbf{G}^{(0)}$. This result can be generalized to other smooth activations.

**Theorem 4.8** (informal version of Theorem J.6). *Let the activation be* tanh *and let the data satisfy Assumption 1 and Assumption 2. Let the depth $L$ satisfy $L = \Theta(\log 1/\delta)$. Then $\lambda_{\min}(\mathbf{G}^{(0)}) \geq e^{-\Omega(\sqrt{\log n})} \gg 1/\mathsf{poly}(n)$ with high probability, provided $m \geq \Omega\left(\mathsf{poly}(n, 1/\delta)\right)$.*

## 5 EXTENSIONS

For a large part of the paper we confine ourselves to the case of one hidden layer where only the input layer is trained. This is in order to focus on the core technical issues of the spectrum of the $G$-matrix. Indeed, our results can be extended along several axes, often by combining our results for the $G$-matrix with existing techniques from the literature. We now briefly discuss some of these extensions. Some of these are worked out in the appendix for completeness.

We can easily generalize to the case when the output layer is also trained (Section M.2). Also, we have focused on training with gradient descent, but training with stochastic gradient descent can also be analyzed for activations in $\mathbf{J_r}$ (Section M.5).

Generalization bounds from Arora et al. (2019c) can easily be extended to the set of functions satisfying $\mathbf{J_r}$ using techniques from Du et al. (2019b). Similarly, techniques from Allen-Zhu et al. (2019) for higher depth generalize to functions such as SELU, LReLU and ELU. We believe this also generalizes to $\mathbf{J_r}$. Other loss functions such as cross-entropy can be handled as well as activations with more than one kink. The case of multi-class classification can also be handled (Sec. M.3). We do not pursue these directions in this paper choosing to focus on the core issues about activations. We briefly discuss extension to more general classes of activations in Appendix H.

## 6 PROOF SKETCH

In this section, we provide a high level sketch of the proofs of our results.

**Activations with a kink.** We first sketch the proof of Theorem K.1, which shows that the minimum eigenvalue of the $G$-matrix is large for activations satisfying $\mathbf{J_1}$. As an illustrative example, consider ReLU. Its derivative, the step function, is discontinuous at 0. In their convergence proof for ReLU networks, Allen-Zhu et al. (2019) prove that the norm of the gradient for a $\mathbf{W}$ is large if the loss at $\mathbf{W}$ is large. We observe that their technique also shows a lower bound on the lowest singular value of the $\mathbf{M}$-matrix by considering the norm of all possible linear combinations of the columns. For $\zeta \in \mathbb{S}^{n-1}$, define the linear combination $f_\zeta(\mathbf{w}) := \sum_{i=1}^n \zeta_i\, \phi'(\mathbf{w}^T \mathbf{x}_i)\, \mathbf{x}_i$.

**Theorem 6.1** (Informal statement of Claim K.2). *Let $\phi \in \mathbf{J_1}$. For any $\zeta \in \mathbb{S}^{n-1}$, $f_\zeta(\mathbf{w})$ has large norm with high probability for a randomly chosen standard Gaussian vector $\mathbf{w}$.*

Using an $\epsilon$-net argument on $\zeta$, the above result implies a lower bound on the minimum singular value of $\mathbf{M}$. Allen-Zhu et al. (2019) write $\mathbf{w}$ as two independent Gaussian vectors $\mathbf{w}'$ and $\mathbf{w}''$, with large and small variances respectively. They isolate an event $E$ involving $\mathbf{w}'$. This event happens if all but one of the summands in $f_\zeta(\mathbf{w}) = \sum_{i=1}^n \zeta_i\, \phi'((\mathbf{w}' + \mathbf{w}'')^T \mathbf{x}_i)\, \mathbf{x}_i$ are fixed to constant values with good probability over the choice of $\mathbf{w}''$. For the exceptional summand, say $\zeta_j\, \phi'((\mathbf{w}' + \mathbf{w}'')^T \mathbf{x}_j)\mathbf{x}_j$, the choice of $\mathbf{w}'$ is such that the argument $(\mathbf{w}' + \mathbf{w}'')^T \mathbf{x}_j$ can be on either side of the jump discontinuity with large probability over the random choice of $\mathbf{w}''$. The random choice of $\mathbf{w}''$ now shows that the whole sum is not concentrated and so with significant probability has large norm. They show that $E$ has substantial probability over the choice of $\mathbf{w}'$, which implies that with significant probability $\|f_\zeta(\mathbf{w})\|$ is large. The property of all but one of the summands being fixed relies crucially on the fact that the derivative of ReLU is constant on both sides of the origin.

When generalizing this proof to activations in $\mathbf{J_1}$ we run into the difficulty that the derivative need not be a constant function on the two sides of the jump discontinuity. We are able to resolve this difficulty with additional technical ideas, in particular, using the assumption that $|\phi'(\cdot)|$ is bounded.

We work with event $E'$ involving $\mathbf{w}'$: in the sum defining $f_\zeta(\mathbf{w})$ there is one exceptional summand $\zeta_j \, \phi'((\mathbf{w}' + \mathbf{w}'')^T \mathbf{x}_j) \, \mathbf{x}_j$ such that the sum involving the rest of the summands—while not fixed to a constant value unlike for ReLU—does not change much over the random choice of $\mathbf{w}''$. Whereas the exceptional summand varies a lot with the random choice of $\mathbf{w}''$ because the argument moves around the jump discontinuity. We show that $E'$ has significant probability, which proves the theorem.

Now, we look at the proof of Theorem 4.1 which handles activations in $\mathbf{J_2}$. The goal again is to show that for any $\zeta \in \mathbb{S}^{n-1}$, the function $f_\zeta(\mathbf{w})$ has large norm with good probability for the random choice of $\mathbf{w}$. To this end, we consider the Taylor approximation of $g_\zeta(\mathbf{w}) := \sum_{i=1}^n \zeta_i \, \phi'(\mathbf{w}^T \mathbf{x}_i)$ around $\mathbf{w}'$, that is, $g_\zeta(\mathbf{w}' + \mathbf{w}'') = g_\zeta(\mathbf{w}') + (\nabla_\mathbf{w} g_\zeta(\mathbf{w}'))^T \mathbf{w}'' + H(\mathbf{w}', \mathbf{w}'')$ where $H$ is the error term. We show that $\left\| \nabla_\mathbf{w} g_\zeta(\mathbf{w}') \right\|$ is likely to be large over the random choice of $\mathbf{w}'$, and so $(\nabla_\mathbf{w} g_\zeta(\mathbf{w}'))^T \mathbf{w}''$ is likely to be not concentrated on any single value if the error term $H(\mathbf{w}', \mathbf{w}'')$ is sufficiently small, which we show. To prove that $\left\| \nabla_\mathbf{w} g_\zeta(\mathbf{w}') \right\|$ is large, note that $\nabla_\mathbf{w} g_\zeta(\mathbf{w}') = \sum_{i=1}^n \zeta_i \, \phi''(\mathbf{w}^T \mathbf{x}_i) \, \mathbf{x}_i$, which allows us to use the argument above for $f_\zeta(\mathbf{w})$ being large in the case of $\mathbf{J_1}$. This implies that $g_\zeta(\mathbf{w})$ is large with good probability, which implies, with further argument, that $f_\zeta(\mathbf{w})$ is large. Full proofs of these results can be found in Appendix K. As mentioned earlier, the argument can be generalized to condition $\mathbf{J_r}$ for any constant $r$; we omit the details.

**Smooth activations.** First we look at the proof sketch for Theorem 4.4. To understand the behavior of smooth activations under gradient descent, we first look at the behavior of a natural subclass of smooth functions: polynomials. The proof actually works with the $M$-matrix introduced earlier whose spectrum is closely related to that of $\mathbf{G} = \mathbf{M}^T \mathbf{M}/m$. In this case, the problem of computing the smallest eigenvalue reduces to finding a non-trivial linear combination of the columns of $\mathbf{M}$ resulting in $\mathbf{0}$. We show that if $d'$, the dimension of the span of the data, is sufficiently small, then this can be done implying that the smallest eigenvalue is 0. By a simple extension of the argument, we can show that in fact the $G$-matrix has low rank. This gives

**Theorem 6.2** (Informal version of Theorem F.2). *The $G$-matrix for polynomial activation functions has low rank if the data spans a low-dimensional subspace.*

Given that polynomials have singular $M$-matrices, a natural idea is to approximate the smooth function $\tanh'$ by a suitable family of polynomials, and then use the above theorem to "kill" the polynomial part using an appropriate linear combination and get an upper bound on the eigenvalue comparable to the error in the approximation. An immediate choice is Taylor's approximation. The Taylor series for $\tanh'$ around 0 has a radius of convergence $\pi/2$. Depending on the initialization and $m$, the argument of the function can take values outside $[-\pi/2, \pi/2]$. To circumvent this difficulty, we consider a different notion of approximation. Consider a series of Chebyshev polynomials $\sum a_n T_n(x)$ that approximates $\tanh'(x)$ in the $L^\infty$ norm in some finite interval. The fact that $\tanh'$ can be extended analytically to the complex plane can be used to show that the coefficients of the above series decay rapidly. The approximation is captured by the following theorem.

**Theorem 6.3** (Informal version of Theorem F.4). *$\tanh'$ is approximable on the interval $[-k, k]$ in the $L^\infty$-norm by (Chebyshev) polynomials to error $\epsilon > 0$ using a polynomial of degree $\mathcal{O}(k \log(k/\epsilon))$.*

When applying the lemma above, the degree required for approximation increases with the number of neurons $m$. This is because the interval $[-k, k]$, in which the approximation is required to hold, grows with $m$ (the maximum of $mn$ Gaussians grows as $\mathcal{O}(\sqrt{\log mn})$). This leads the degree of polynomial to become too large to be "killed" as $m$ becomes larger. Thus, for large $m$, this fails to give the required bound. To remedy this, we relax the approximation requirement. Since we are working with Gaussian initialization of weights a natural relaxation is the $L^2$-approximation under the Gaussian measure. This leads us to consider the Hermite expansion (see also Daniely et al. (2016)) of $\tanh'$. The $p$-th coefficient in Hermite expansion is an integral involving the $p$-th Hermite polynomial. For large $p$, these polynomials are highly oscillatory which makes evaluation of the coefficients difficult. Fortunately, a theorem of Hille (1940) comes to rescue. Again, the fact that $\tanh'$ can be analytically extended to a certain region of the complex plane can be used to bound the decay of the Hermite coefficients, which in turn bounds the error in polynomial approximation:

**Theorem 6.4** (Informal version of Theorem G.2). *$\tanh'$ is approximable on $\mathbb{R}$ in the $L^2$-norm with respect to the Gaussian measure by (Hermite) polynomials of degree $p$ with error $\exp(-\Omega(\sqrt{p}))$.*

In contrast, the $p$-th Hermite coefficients of the step function (also called threshold or sgn) which is the derivative of ReLU (whose $G$-matrix has large minimum eigenvalue), decays as $p^{-0.75}$ (this fact

underlies Theorem L.2). The $L^2$-approximation gives us a bound on the expected loss. To argue about high probability bounds, we need to resort to concentration of measure arguments. This requires the number of neurons $m$ to be large. Thus, these two notions of approximation complement each other.

Now, using these theorems for the small and large $m$ regimes, we can show that the eigenvalues of the $G$-matrix are indeed small as stated in Theorem 4.4. These results can be easily extended to swish. In fact, the above theorems hold for general functions satisfying certain regularity conditions such as having an analytic continuation onto a strip of complex plane that contains the domain of interest, e.g. an interval of $\mathbb{R}$ or all of $\mathbb{R}$ (Appendix F).

**For smoothed data not restricted to small dimension, tanh works well.** We sketch the proof of Theorem 4.7 showing that our results about the limitations of smooth activations are essentially tight when the data is smoothed. It is well-known (see Fact C.9) that the minimum singular value of a (tall) matrix $\mathbf{M}$ is lower-bounded as follows: take a column of $\mathbf{M}$ and consider its distance from the span of the rest of the columns. The minimum of this quantity over all columns gives a lower bound on the minimum singular value (up to polynomial factors in the dimensions of $\mathbf{M}$). The problem of lower bounding $\lambda_{\min}(G^{(0)})$ then reduces to the problem of lower bounding a product involving (a) the minimum singular value of $\mathbf{X}^{*p}$, the $p$-th Khatri–Rao power of the data matrix $\mathbf{X} = [\mathbf{x}_1, \ldots, \mathbf{x}_n]$, (b) the $p$-th Hermite coefficient of $\tanh'$ (see Lemma I.1 and Lemma I.2). We use $p$ to be approximately equal to $1/\gamma$. For (a) we use the above strategy to lower bound the minimum singular value of $\mathbf{X}^{*p}$. It turns out that for any given column, its distance from the span of the rest of the columns can be written as a polynomial in the noise variables. We can then use the anticoncentration inequality of Carbery–Wright (see Fact C.8) to show that this distance is unlikely to be small, and then use the union bound to show that this is unlikely to be small for every column. For (b), we invoke a result of Boyd (1984) implying that the upper bound $\exp(-\Omega(\sqrt{p}))$ on the $p$-th Hermite coefficient of $\tanh'$ used in Theorem 6.4 above is essentially tight. The choice of $p$ that gives the best lower bound depends on the activation function.

**Depth helps for tanh.** We now sketch the proof of Theorem 4.8 (for a formal statement, see Theorem J.6). For each $i \in [n]$ and $l \in \{0, ..., L-1\}$, let $\mathbf{x}_i^{(l)}$ be the output of layer $l$ on input $\mathbf{x}_i$. We track the behavior of the $\mathbf{x}_i^{(l)}$ as $l$ increases:

**Lemma 6.5** (informal version of Lemma J.1 and Lemma J.2). *As $l$ increases, the Euclidean norm of each $\mathbf{x}_i^{(l)}$ is approximately preserved. On the other hand, for every $i \neq j$, $|(\mathbf{x}_i^{(l)})^T \mathbf{x}_j^{(l)}|$ shrinks.*

This implies that for sufficiently large $L$, the Gram matrix of the output of the penultimate layer, i.e. $(\mathbf{X}^{(L-1)})^T \mathbf{X}^{(L-1)}$, where $\mathbf{X}^{(L-1)} = [\mathbf{x}_1^{(L-1)}, \ldots, \mathbf{x}_n^{(L-1)}]$ is diagonally-dominant and has large minimum eigenvalue. The rest of the proof has some overlap with the proof for smoothed data above. For each $p \geq 0$, $\lambda_{\min}(\mathbf{G}^{(0)})$ can be lower bounded by a product involving (a) the minimum eigenvalue of the $p$-th Hadamard power of the Gram matrix of $\mathbf{x}_i^{(L-1)}$, and (b) the $p$-th Hermite coefficient of $\tanh'$ (see Equation 47). For (a) we use the diagonal-dominance of the Gram matrix. For (b) we proceed as in the case of smoothed data. The choice $p = \Theta(\log n)$ turns out to give the best lower bound on $\lambda_{\min}(\mathbf{G}^{(0)})$.

## 7 EXPERIMENTS

**Synthetic data.** We consider $n$ equally spaced data points on $\mathbb{S}^1$, randomly lifted to $\mathbb{S}^9$. We randomly label the data-points from $\mathcal{U}\{-1, 1\}$. We train a 2-layer neural network in the DZPS setting with mean squared loss, containing $10^6$ neurons in the first layer with activations tanh, ReLU, swish and ELU at learning rate $10^{-3}$. The output layer is not trained during gradient descent. In Figure 1(a) and Figure 1(b) we plot the squared loss against the number of epochs trained. Results are averaged over 5 different runs. We observed that the eigenvalues and the eigenvectors stayed essentially constant throughout training, indicating overparametrized regime. ReLU converges to zero training error much faster than other activation functions, ELU is faster than tanh and swish. In Figure 1(c) and Figure 1(d) we plot the eigenvalues at initialization. Eigenvalues of ReLU and ELU are larger compared to those of tanh and swish. This is consistent with the theory.

**Real data.** We consider a random subset of $10^4$ images from CIFAR10 dataset (Krizhevsky & Hinton, 2009). We train a 2-layer network containing $10^5$ neurons in the first layer. First, we verify Assumption 2 regarding $\delta$-separation of data samples. We plot the $L^2$-distances between all pairs

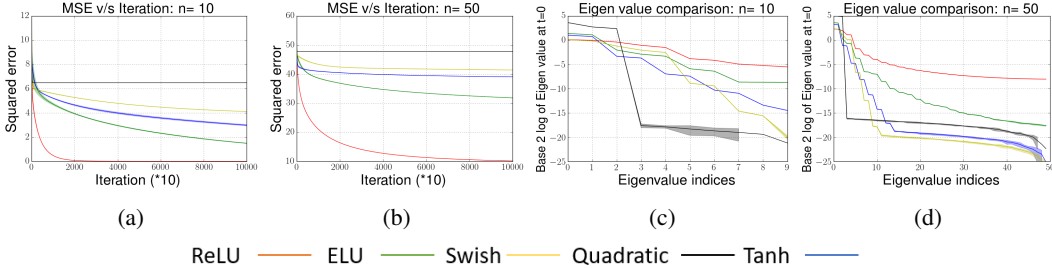

Figure 1: Experiments on synthetic dataset (From left to right) (a)Rate of convergence of 2-layer network for different activations when $n = 10$ (b) Rate of convergence of 2-layer network for different activations when $n = 50$ (c) Eigenvalues of the $G$-matrix at initialization for different activations when $n = 10$ (d) Eigenvalues of the $G$-matrix at initialization for different activations when $n = 50$

of preprocessed images (described in Section 2) in Figure 2(a). It shows that the assumptions hold for CIFAR10, with $\delta$ at least 0.1. Figure 2(b) has the plot of the cumulative sums of eigenvalues, normalized to the range $[0, 1]$, of the data covariance matrix. This figure shows that the intrinsic dimension of data is much larger than $\mathcal{O}(\log n)$, where $n$ denotes the number of samples.

Eigenvalues of the $G$-matrix for different activations at initialization are plotted in Figure 2(c). This shows that ReLU has higher eigenvalues compared to other activations. However there isn't much difference between the spectrum of ELU and tanh. This is likely due to the fact that we are in the regime of Theorem I.4.

We observed a difference in the rate of convergence while training a 2-layer network, with both layers trainable, using 256 batch sized stochastic gradient descent (SGD) with cross entropy loss on the random subset of CIFAR10 dataset at l.r. $10^{-3}$ (Figure 2(d)). Here we are not in the overparametrized regime as the eigenvalues and eigenvectors change considerably during training. Therefore, observations in Figure 2(d) can be attributed to the eigenvalue plots in Figure 2(c) only in the first few iterations of SGD.

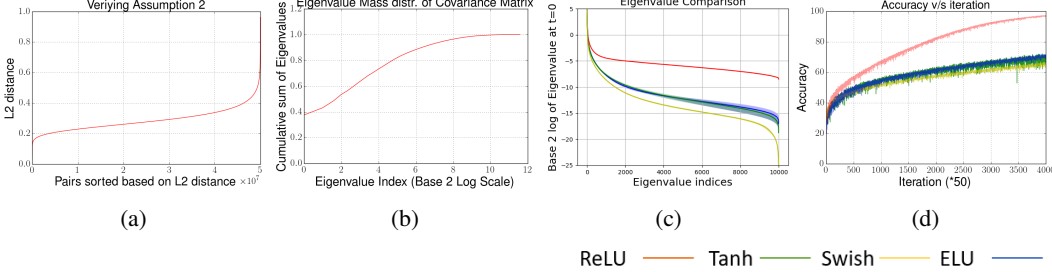

Figure 2: Experiments on a random subset of $10^4$ images from CIFAR10 dataset: (a) $L^2$-distances between all pairs of preprocessed images (b) Semilog plot of sum of squares of top $k$ singular values of data matrix (c) Eigenvalue distribution of $G$-matrix at initialization (d) Convergence speed of 2 layer networks using different activation functions.

## 8   CONCLUSION

In this paper we characterized the effect of activation function on the training of neural networks in the overparametrized regime. Our results hold for very general classes of activations and cover all the activations in use that we are aware of. Many avenues for further investigation remain: there are gaps between theory and practice because of the differences in the sizes, learning rates, optimization procedures and architectures used in practice and those analyzed in theory: compared to practice, most theoretical results in the recent literature (including the present paper) require the size of the networks to be very large and the learning rate to be very small. Bridging this gap is an exciting challenge. Fine-grained distinction between the performance of activations is also of interest. For example, Figure 2(d) shows that ReLU converges much faster compared to the other activations. But the roles can be reversed based on the architecture and the dataset etc., e.g., Ramachandran et al. (2018). In a given situation, what makes one activation more suitable than another?

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

## CONTENTS

## A    ADDITIONAL RELATED WORK

The literature is extensive; we only mention some of the related work Pennington et al. (2017); Louart et al. (2018); Hanin (2018); Hanin & Rolnick (2018); Pennington & Worah (2017; 2018); Pennington et al. (2018). Of these, perhaps the closest to our work is Louart et al. (2018): they keep the random weights in the input layer fixed and only train the output layer. The main result is determination of the spectrum of a matrix associated with the input layer with the activation function being Lipschitz. This matrix is different from one considered here. Also the scaling used is different: both the dimension of the data, number of samples, and the number of neurons grow at the same rate to infinity. The effect of the activation function on the training via the spectrum is considered but no results are provided for popular activations. Hayou et al. (2019) aim to find the best possible initialization of weights, that lead to better propagation of information in an untrained neural network, i.e. the rate of convergence of correlation among hidden layer outputs of datapoints to 1 should be of the order $\frac{1}{\text{poly}(L)}$, where $L$ is the number of layers. They show smooth activation functions have a slower rate and hence, are better to use in deep neural networks. However, this set of parameterizations, called "edge of chaos", were proven to be essential for training by Lee et al. (2017) under the framework of equivalence of infinite width deep neural networks to Gaussian processes over the weight parameters. The extent to which the approximation of stochastic gradient descent by Bayesian inference holds is still an open problem.

## B    ACTIVATIONS

We introduce some of the most popular activation functions. These activation functions are unary, i.e. of type $\phi : \mathbb{R} \to \mathbb{R}$, and act on vectors entrywise: $\phi((t_1, t_2, ...)) := (\phi(t_1), \phi(t_2), ...)$. In this paper we do not study activation functions with learnable parameters such as PReLU, or activation functions such as maxout which are not unary. Activations functions are also referred to as nonlinearities, although the case of linear activation functions has also received much attention.

- ReLU$(X) := \max(x, 0)$
- LReLU$(x) := \max(x, 0) + \alpha \min(x, 0)$, where $\alpha$ is a constant less than 1
- Linear$(x) := x$
- tanh$(x) := \frac{e^{2x}-1}{e^{2x}+1}$
- sigmoid$(x) := \frac{1}{1+e^{-x}}$
- swish$(x) := \frac{x}{1+e^{-x}}$ Ramachandran et al. (2018) (called SiLU in Elfwing et al. (2017))
- ELU$(x) := \max(x, 0) + \min(x, 0)(e^x - 1)$ Clevert et al. (2016)
- SELU$(x) := \alpha_1 \max(x, 0) + \alpha_2 \min(x, 0)(e^x - 1)$, where $\alpha_1$ and $\alpha_2$ are two different constants Klambauer et al. (2017).

## C    PRELIMINARY FACTS AND DEFINITIONS

We note some well known facts about concentration of Gaussian random variables to be used in the proofs.

**Fact C.1.** *For a Gaussian random variable* $v \sim \mathcal{N}(0, \sigma^2)$, $\forall t \in (0, \sigma)$, *we have*

$$P_v\{|v| \geq t\} \in \left(1 - \frac{4}{5}\frac{t}{\sigma}, 1 - \frac{2}{3}\frac{t}{\sigma}\right).$$

*In fact, the following holds true.* $\forall t \in (0, \sigma/2)$, *we have*

$$P_v\{|v - a| \geq t\} \in \left(1 - \frac{4}{5}\frac{t}{\sigma}, 1 - \frac{1}{4}\frac{t}{\sigma}\right).$$

*where* $0 \leq a \leq \sigma$.

**Fact C.2.** *For a Gaussian random variable* $v \sim \mathcal{N}(\mu, \sigma^2)$, $\forall t \in (0, \sigma)$, *we have* $\forall t \geq 0$

$$P_v\left\{|v - \mu| \leq t\right\} \leq 1 - 2e^{-\frac{t^2}{2\sigma^2}}.$$

**Fact C.3** (Hoeffding's inequality, see Boucheron et al. (2013)). *Let $x_1, x_2, .., x_n$ be n independent random variables, where $x_i$ lies in the interval $[a_i, b_i]$, and let $\bar{x}$ be the empirical mean, i.e., $\bar{x} = \frac{\sum_{i=1}^n x_i}{n}$. Then,*

$$\Pr\left(\left|\bar{x} - \mathbb{E}\left(\bar{x}\right)\right| \geq t\right) \leq 2e^{-\frac{2n^2 t^2}{\sum_{i=1}^n (a_i - b_i)^2}}.$$

**Fact C.4** (Multiplicative Chernoff bound, see Boucheron et al. (2013)). *Let $x_1, x_2, .., x_n$ be independent random variables taking values in $\{0, 1\}$, and let $\bar{x}$ be the empirical mean, i.e., $\bar{x} = \frac{\sum_{i=1}^n x_i}{n}$. Then*

$$\Pr\left(\bar{x} < (1-t)\,\mathbb{E}\left(\bar{x}\right)\right) \leq e^{-\frac{t^2 \mathbb{E}(\bar{x})}{2}},$$
$$\Pr\left(\bar{x} > (1+t)\,\mathbb{E}\left(\bar{x}\right)\right) \leq e^{-\frac{t^2 \mathbb{E}(\bar{x})}{2+t}}$$

*for all $t \in (0, 1)$.*

**Fact C.5** (Maximum of Gaussians, see Boucheron et al. (2013)). *Let $x_1, x_2, ..., x_n$ be n Gaussians following $\mathcal{N}\left(0, \sigma^2\right)$. Then,*

$$\Pr\left\{\max_{i \in [n]} |x_i| \leq k\sqrt{\log n}\,\sigma\right\} \geq 1 - \frac{2}{n^{\frac{k^2}{2} - 1}}.$$

*where $k > 0$ is a constant.*

**Fact C.6** (Chi square concentration bound, see lemma 1 in Laurent & Massart (2000)). *For a variable $x$ that follows chi square distribution with $k$ degrees of freedom, the following concentration bounds hold true.*

$$P(x - k \geq 2\sqrt{kt} + 2t) \leq \exp(-t),$$
$$P(k - x \geq 2\sqrt{kt}) \leq \exp(-t).$$

**Fact C.7** (Mini-max formulation of singular values). *Given a matrix $\mathbf{M} \in \mathbb{R}^{m \times n}$, assuming $n \leq m$, the singular values of $\mathbf{M}$ can be defined as follows.*

$$\sigma_k\left(\mathbf{M}\right) = \min_{\mathbf{U}}\left\{\max_{\zeta}\left\{\mathbf{M}\zeta \middle| \zeta \in \mathbf{U}, \zeta \neq 0\right\} \middle| \dim\left(\mathbf{U}\right) = k\right\} \quad \forall k \in [n].$$

**Fact C.8** (adaptation of Carbery & Wright (2001); see Anderson et al. (2014)). *Let $Q\left(x_1, \ldots, x_n\right)$ be a multilinear polynomial of degree d. If $\mathrm{Var}\left(Q\right) = 1$, when $x_i \sim \mathcal{N}\left(0, 1\right) \forall i \in [n]$. Then, $\forall t \in \mathbb{R}$ and $\epsilon > 0$, there exits $C > 0$ s.t.*

$$\Pr_{(x_1, \ldots, x_n) \sim \mathcal{N}(0, \mathbf{I}_n)}\left(\left|Q\left(x_1, \ldots, x_n\right) - t\right| \leq \epsilon\right) \leq C d \epsilon^{1/d}.$$

**Fact C.9** (Rudelson & Vershynin (2009)). *Given a matrix $\mathbf{A} \in \mathbb{R}^{n \times m}$, the following holds true for all $i \in [m]$.*

$$\frac{1}{\sqrt{m}} \min_{i \in [m]} \mathrm{dist}\left(\mathbf{a}_i, \mathbf{A}_{-i}\right) \leq \sigma_{\min}(\mathbf{A})$$

*where $\mathbf{A}_{-i} = \mathrm{span}\left(\mathbf{a}_j : j \neq i\right)$*

**Fact C.10** (Gershgorin circle theorem, see e.g. Varga (2010)). *Every eigenvalue of a matrix $\mathbf{A} \in \mathbb{C}^{n \times n}$, with entries $a_{ij}$, lies within at least one of the discs $\left\{D(a_{ii}, r_i)\right\}_{i=1}^n$, where $r_i = \sum_{j:j \neq i}|a_{ij}|$ and $D(a_{ii}, r_i) \subset \mathbb{C}$ denotes a disc centered at $a_{ii}$ and with radius $r_i$.*

**Fact C.11** (Weyl's inequalities (Weyl, 1912)). *Let $\mathbf{A}$ and $\mathbf{B}$ be two hermitian matrices. Then, the following hold true $\forall i, j \in [n]$.*

$$\lambda_{i+j-1}\left(\mathbf{A} + \mathbf{B}\right) \leq \lambda_i\left(\mathbf{A}\right) + \lambda_j\left(\mathbf{B}\right)$$

$$\lambda_{i+j-n}\left(\mathbf{A} + \mathbf{B}\right) \geq \lambda_i\left(\mathbf{A}\right) + \lambda_j\left(\mathbf{B}\right)$$

**Definition C.1** (Khatri Rao product and Hadamard product, see Khatri & Rao (1968)). Given a matrix $\mathbf{A} \in \mathbb{R}^{m \times n}$ and a matrix $\mathbf{B} \in \mathbb{R}^{p \times q}$, the Kronecker (or tensor) product $\mathbf{A} \otimes \mathbf{B}$ is defined as the $mp \times nq$ matrix given by

$$
\mathbf{A} \otimes \mathbf{B} = \begin{bmatrix} a_{11}\mathbf{B} & a_{12}\mathbf{B} & \cdots & a_{1n}\mathbf{B} \\ a_{21}\mathbf{B} & a_{22}\mathbf{B} & \cdots & a_{2n}\mathbf{B} \\ \vdots & \vdots & \ddots & \vdots \\ a_{m1}\mathbf{B} & a_{m2}\mathbf{B} & \cdots & a_{mn}\mathbf{B} \end{bmatrix}.
$$

Given a matrix $\mathbf{A} \in \mathbb{R}^{m \times n}$ and a matrix $\mathbf{B} \in \mathbb{R}^{p \times n}$, the Khatri Rao product $\mathbf{A} * \mathbf{B}$ is defined as the $mp \times n$ matrix given by

$$
\mathbf{A} * \mathbf{B} = \begin{bmatrix} \mathbf{a}_1 \otimes \mathbf{b}_1 & \mathbf{a}_2 \otimes \mathbf{b}_2 & \cdots \mathbf{a}_n \otimes \mathbf{b}_n \end{bmatrix}.
$$

Given a matrix $\mathbf{A} \in \mathbb{R}^{m \times n}$ and a matrix $\mathbf{B} \in \mathbb{R}^{m \times n}$, the Hadamard product $\mathbf{A} \odot \mathbf{B} \in \mathbb{R}^{m \times n}$ is defined as

$$
(\mathbf{A} * \mathbf{B})_{ij} = a_{ij}b_{ij}, \quad \forall i \in [m], j \in [n].
$$

**Notation:** Given a matrix $\mathbf{X} \in \mathbb{R}^{m \times n}$, we denote order-$r$ Khatri-Rao product of $\mathbf{X}$ as $\mathbf{X}^{*r} \in \mathbb{R}^{m^r \times n}$, which represents $\underbrace{\mathbf{X} * \mathbf{X} \ldots * \mathbf{X}}_{r \text{ times}}$. We denote order-$r$ Hadamard product of $\mathbf{X}$ as $\mathbf{X}^{\odot r} \in \mathbb{R}^{n \times n}$, which represents $\underbrace{\mathbf{X} \odot \mathbf{X} \ldots \odot \mathbf{X}}_{r \text{ times}}$ and can be shown to be equal to $(\mathbf{X}^{*r})^T \mathbf{X}^{*r}$.

## C.1 GENERAL APPROACH FOR BOUNDING EIGENVALUE OF $\mathbf{G}$-MATRIX

The Gram matrix $\mathbf{G}$ can be written as

$$
\mathbf{G} = \frac{1}{m} \mathbf{M}^T \mathbf{M}.
$$

where $\mathbf{M} \in \mathbb{R}^{md \times n}$ is defined by

$$
\mathbf{M}_{d(k-1)+1:\, dk, i} = a_k \phi' \left( \mathbf{w}_k^T \mathbf{x}_i \right) \mathbf{x}_i \quad \text{for } k \in [m].
$$

Denoting by $\lambda_k(\mathbf{G})$ and $\sigma_k(\mathbf{M})$ as $k$th eigenvalue and $k$th singular value of $\mathbf{G}$ and $\mathbf{M}$ respectively, we can write $\lambda_k(\mathbf{G})$ as $\frac{1}{m}\sigma_k(\mathbf{M})^2$. The minimum singular value of $\mathbf{M}$ can be defined (through the minmax theorem) as

$$
\sigma_{\min}(\mathbf{M}) = \min_{\zeta \in \mathbb{S}^{n-1}} \|\mathbf{M}\zeta\|.
$$

Hence, the general approach to show a lower bound of $\sigma_{\min}(M)$ is to show that the following quantity

$$
\left\| \sum_{i=1}^n \zeta_i \mathbf{m}_i \right\| = \sum_{k=1}^m \sqrt{\left\| \sum_{i=1}^n a_k \zeta_i \phi'(\mathbf{w}_k^T \mathbf{x}_i)\mathbf{x}_i \right\|^2} \tag{4}
$$

is lower-bounded for all $\zeta \in \mathbb{S}^{n-1}$.

To show an upper bound, we pick a $\zeta' \in \mathbb{S}^{n-1}$ and use

$$
\sigma_{\min}(\mathbf{M}) = \min_{\zeta \in \mathbb{S}^{n-1}} \|\mathbf{M}\zeta\| \leq \left\| \sum_{i=1}^n \zeta_i' \mathbf{m}_i \right\|. \tag{5}
$$

## D STANDARD PARAMETRIZATION AND INITIALIZATIONS

A 2-layer (i.e. 1-hidden layer) neural network is given by

$$
F(\mathbf{x}; \mathbf{a}, \mathbf{b}, \mathbf{W}) = \sum_{k=1}^m a_k \phi \left( \mathbf{w}_k^T x + b_k \right),
$$

where $\mathbf{a} \in \mathbb{R}^m$ is the output layer weight vector, $\mathbf{W} \in \mathbb{R}^{d \times m}$ is the hidden layer weight matrix and $\mathbf{b} \in \mathbb{R}^m$ denotes the hidden layer bias vector. This parametrization differs slightly from the choice made in Equation 1. By standard initialization techniques He et al. (2015) and Glorot & Bengio (2010), there are two ways in which the initial values of the weights $\mathbf{W}^{(0)}$ and $\mathbf{a}^{(0)}$ are chosen, depending on whether the number of neurons in the previous layer is taken into account (fanin) or the number of neurons in the current layer is taken into account (fanout).

- Init(fanin) : $\mathbf{w}_k^{(0)} \sim \mathcal{N}\left(0, \frac{1}{d}\mathbf{I}_d\right)$ and $a_k^{(0)} \sim \mathcal{N}\left(0, \frac{1}{m}\right) \quad \forall k \in [m]$,
- Init(fanout): $\mathbf{w}_k^{(0)} \sim \mathcal{N}\left(0, \frac{1}{m}\mathbf{I}_d\right)$ and $a_k^{(0)} \sim \mathcal{N}\left(0, 1\right) \quad \forall k \in [m]$.

Note that Allen-Zhu et al. (2019) use Init (fanout) initialization. The elements of $\mathbf{b}$ are initialized from $\mathcal{N}\left(0, \beta^2\right)$, where $\beta$ is a small constant. We set $\beta = 0.1$ in Init (fanin) initialization and $\beta = \frac{1}{\sqrt{m}}$ in Init (fanout) initialization.

## D.1   NOTE ON $\mathbf{G}$-MATRIX FOR STANDARD INITIALIZATIONS

We follow the argument in Section 3 to get the $\mathbf{G}$-matrix $(\mathbf{G})$ as

$$g_{ij}^{(t)} = \eta \sum_{k=1}^{m} a_k^2 \phi'\left(\mathbf{w}_k^T \mathbf{x}_i + b_k\right) \phi'\left(\mathbf{w}_k^T \mathbf{x}_j + b_k\right) \langle \mathbf{x}_i, \mathbf{x}_j \rangle,$$

where $\eta$ is the gradient flow rate needed to control the gradient during descent algorithm to stop gradient explosion, i.e. we need to control the maximum eigenvalue of the Gram matrix.

For Init (fanin) and Init (fanout), we set $\eta$ as 1 and $\frac{1}{m}$ respectively to get the same Gradient Gram matrix as in Equation 2.

## E   LOWER BOUND ON THE TRACE OF $G$-MATRIX

In this section, we lower bound the trace of the gradient matrix for a general activation function as a point of comparison for our results regarding the lowest eigenvalue.

**Theorem E.1.** *Let $\phi$ be an activation function, with Lipschitz constant $\alpha$ and let $\mathbf{G}^{(0)}$ be the $\mathbf{G}$-matrix at initialization. Let $\mathbb{E}_{\mathbf{w} \sim \mathcal{N}(0, \mathbf{I})}\left(\phi'\left(\mathbf{w}^T \mathbf{x}_i\right)^2\right) = 2c$ for a positive constant $c$ and for all $i$. Then, $\mathsf{tr}(\mathbf{G}^{(0)}) \geq cn$ with probability greater than $1 - e^{-\Omega\left(m/\alpha^2 \log^2 m\right)} - m^{-3.5}$.*

*Remark.* The constant $c$ depends on the choice of the activation function and is bounded away from $0$ for most activation functions such as ReLU or tanh.

*Proof.* The trace of the $G$-matrix is given by

$$\mathsf{tr}(\mathbf{G}^{(0)}) = \frac{1}{m} \sum_{j=1}^{m} \sum_{i=1}^{n} \left(a_j^{(0)} \phi'\left(\mathbf{w}_j^{(0)T} x_i\right)\right)^2.$$

Using Fact C.5, $\left\{a_j^{(0)}\right\}_{j=1}^{m}$ can be shown to be in the range $\left(-3\sqrt{\log m}, 3\sqrt{\log m}\right)$ with probability at-least $1 - m^{-3.5}$. Assuming this is true, we claim the following two statements. For any $j$ we have (expectation is over $\mathbf{W}^{(0)}$ and $\left\{a_k^{(0)}\right\}_{k=1}^{m}$)

$$\mathbb{E}_{\mathbf{w}_j^{(0)}, a_j^{(0)} : |a_j^{(0)}| \leq 3\sqrt{\log m}} \sum_{i=1}^{n} \left(a_j^{(0)} \phi'\left(\mathbf{w}_j^{(0)T} x_i\right)\right)^2$$

$$= \sum_{i=1}^{n} \mathbb{E}_{a_j^{(0)} : |a_j^{(0)}| \leq 3\sqrt{\log m}} \left(a_j^{(0)}\right)^2 \mathbb{E}_{\mathbf{w}_j^{(0)}}\left(\phi'\left(\mathbf{w}_j^{(0)T}\mathbf{x}_i\right)\right)^2$$

$$\geq \sum_{i=1}^{n} 2c\left(1 - \sqrt{\frac{\log m}{m}}\right)$$

$$\geq cn.$$

where we use the independence of $a_j^{(0)}$ and $\mathbf{w}_j^{(0)}$ and the variance of $a_j^{(0)}$ is at-least $\left(1 - \sqrt{\frac{\log m}{m}}\right)$, taking the bound on $a_j^{(0)}$ into account, in the intermediate steps.

Also, we get from the Hoeffding bounds,

$$\Pr\left[\frac{1}{m}\sum_{j=1}^{m}\sum_{i=1}^{n}\left(a_j^{(0)}\phi'\left(\left(\mathbf{w}_j^{(0)}\right)^T\mathbf{x}_i\right)\right)^2 \geq \frac{1}{2}cn\right] \leq 1 - e^{-\Omega\left(mn^2/\alpha^4\log^2 m\right)}$$

as required. $\qquad\square$

This shows that the trace is large with high probability. From it follows that average of eigenvalue is $\Omega(1)$. It also follows that maximum eigenvalue is $\Omega(1)$.

## F    UPPER BOUND ON LOWEST EIGENVALUE FOR tanh

For activation functions represented by polynomials of low degree, we show that the $G$-matrix is singular. In fact, the rank of this matrix is small if the degree of the polynomial is small. To upper bound the lowest eigenvalue of the $G$-matrix for the tanh activation function, we proceed by approximating $\tanh'$ with polynomials of low degree. It turns out that $\tanh'$ can be well-approximated by polynomials in senses to be described below. This allows us to use the result about polynomial activation functions to show that the minimum singular value of the $G$-matrix of tanh is small. The approximation of $\tanh'$ by polynomials turns out to be non-trivial. It does have Taylor expansion centered at $0$ but the radius of convergence is $\pi/2$ and thus cannot be used directly if the approximation is required for bigger intervals.

We consider two different notions of approximations by polynomials, depending on the initialization and the regime of the parameters. It is instructive to first consider polynomial activations before going to tanh. Next section is devoted to polynomial activations.

### F.1    POLYNOMIAL ACTIVATION FUNCTIONS

Let's begin with the linear activation function. In this case, the $G$-matrix turns out to be the Gram matrix of the data. Since each datapoint is low dimensional, we get that it is singular:

**Lemma F.1.** *(Linear activation function) If $\phi(x) = x$ and $d < n$, then the G-matrix is singular, that is*

$$\lambda_{\min}\left(\mathbf{G}^{(0)}\right) = 0.$$

*In fact, at least $(n - d)$ eigenvalues of the G-matrix are 0.*

*Proof.* Since $\phi(x) = x$, we have $\phi'(x) = 1$ and the $G$-matrix is given by

$$\mathbf{G} = \frac{1}{m}\left(\sum_{j=1}^{m}a_j^2\right)\mathbf{X}^T\mathbf{X}$$

where $\mathbf{X} = [\mathbf{x}_1, \mathbf{x}_2, ..., \mathbf{x}_n]$ is the matrix containing the $\mathbf{x}_i$'s as its columns. Since the $\mathbf{x}_i$'s are $d$-dimensional vectors, $\text{rank}(\mathbf{X}) \leq d$ and so, $\mathbf{X}$ can have at most $d$ non-zero singular values, which leads to at most $d$ non-zero eigenvalues for $G$. $\qquad\square$

We next show that activation functions represented by low degree polynomial must have singular Gradient Gram matrices:

**Theorem F.2.** *Let $\phi'(x) = \sum_{\ell=0}^{p}c_\ell x^\ell$ and $d' = \dim\left(\text{Span}\{x_1 \ldots x_n\}\right)$. Then, the G-matrix is singular, that is*

$$\lambda_{\min}\left(\mathbf{G}^{(0)}\right) = 0,$$

*assuming that the following condition holds,*

$$\binom{d' + p}{p} < n + 1. \tag{6}$$

*Proof.* Referring to Equation 5, it suffices to find one $\zeta' \in \mathbb{S}^{n-1}$ such that

$$\left\| \sum_{i=1}^{n} \zeta_i' \mathbf{m}_i \right\|_2 = \sqrt{\sum_{k=1}^{m} \left\| \sum_{i=1}^{n} \zeta_i' a_k \phi'(\mathbf{w}_k^T \mathbf{x}_i) \mathbf{x}_i \right\|_2^2} = 0.$$

For any $k \in [m]$ and $\zeta \in \mathbb{R}^n$ consider

$$\sum_{i=1}^{n} \zeta_i a_k \phi'(\mathbf{w}_k^T \mathbf{x}_i) \mathbf{x}_i = \sum_{i=1}^{n} \zeta_i a_k \sum_{\ell=1}^{p} c_{\ell-1} \left( \mathbf{w}_k^T \mathbf{x}_i \right)^{\ell-1} \mathbf{x}_i$$

which can be written as

$$a_k \sum_{l=1}^{p} c_{\ell-1} \sum_{\beta \in \mathbb{Z}_+^d, \|\beta\|_1 = \ell-1} \underbrace{\left( \sum_{i=1}^{n} \zeta_i \mathbf{x}_i^{\beta} \mathbf{x}_i \right)}_{\dagger} \mathbf{w}_k^{\beta} \tag{7}$$

Here $\mathbb{Z}_+$ denotes the set of non-negative integers, and the notation $\mathbf{x}^{\beta}$ is shorthand for $\prod_{j=1}^{d} x_j^{\beta_j}$. Note that the term denoted by $\dagger$ is a $d$-dimensional vector, since $\mathbf{x}_i \in \mathbb{R}^d$. The actual number of unique equations in $\dagger$ depends on $d'$, since the $\mathbf{x}_i$'s can have only upto $d'$ order unique moments. Hence, if we want to make the above zero, it suffices to make the term denoted by $\dagger$ zero for each of the summands. This is a system of linear equations in variables $\{\zeta_i\}_{i=1}^n$. Counting the number of constraints for each summand and summing over all the indices gives us that the number of constraints is given by

$$\sum_{\ell=1}^{p} \binom{d' + \ell - 1}{\ell}$$

which is equal to $\binom{d'+p}{p} - 1$. Note that this can also be seen by counting the number of non-trivial monomials of degree at most $p$ in $d'$ variables. Hence, making the number of constraints less then number of variables, leads to existence of at least one non-zero vector $\zeta''$ satisfying the set of constraints. Since, the set of linear equations is independent of the choice of $k$, the claim holds true for all $k$. Thus, using a unit normalized $\zeta''$ in Equation 5, we get $\sigma_{\min}(\mathbf{M}) = 0$. $\square$

**Corollary F.2.1.** *If $d' = \dim(\text{span}\{\mathbf{x}_1 \dots \mathbf{x}_n\}) \leq \mathcal{O}\left(\log^{0.75} n\right)$, $\phi'(x) = \sum_{\ell=0}^{p} c_\ell x^\ell$ and $p \leq \mathcal{O}(n^{\frac{1}{d'}})$ then the minimum eigenvalue of the G-matrix satisfies*

$$\lambda_{\min}\left(\mathbf{G}^{(0)}\right) = 0.$$

*Proof.* If $d' = \mathcal{O}\left(\log^{0.75} n\right)$, Condition 6 can be simplified to get

$$p = \mathcal{O}(n^{\frac{1}{d'}})$$

Applying Theorem F.2 with the above condition, we get the desired solution. $\square$

By slightly modifying the proof of the above theorem, we can actually show not only that the matrix is singular, but also that the kernel must be high dimensional.

**Theorem F.3.** *If $d' = \dim(\text{span}\{\mathbf{x}_1 \dots \mathbf{x}_n\}) \leq \mathcal{O}\left(\log^{0.75} n\right)$, $\phi'(x) = \sum_{\ell=0}^{p} c_\ell x^\ell$ and $p \leq \mathcal{O}(n^{\frac{1}{d'}})$, then $\left(1 - \frac{1}{d'}\right) n$ low-order eigenvalues of the G-matrix satisfy*

$$\lambda_k\left(\mathbf{G}^{(0)}\right) = 0, \quad \forall k \geq \left\lceil \frac{n}{d'} \right\rceil.$$

*Proof.* We follow the proof of Theorem F.2. Referring to Fact C.7, it suffices to find one $n\left(1 - \frac{1}{d'}\right)$ dimensional subspace $\mathbf{U}$ such that

$$\left\| \sum_{i=1}^{n} \zeta_i \mathbf{m}_i \right\|_2 = \sqrt{ \sum_{k=1}^{m} \left\| \sum_{i=1}^{n} \zeta_i a_k \phi'(\mathbf{w}_k^T \mathbf{x}_i) \mathbf{x}_i \right\|_2^2 } = 0.$$

holds true $\forall \zeta \in \mathbf{U}$.

For a weight row vector $\mathbf{w}_k \in \{\mathbf{w}_1, \dots, \mathbf{w}_m\}$, its corresponding output weight $a_k$ and an arbitrary $\zeta \in \mathbb{R}^n$, we can simplify the following quantity

$$\sum_{i=1}^{n} \zeta_i a_k \phi'(\mathbf{w}_k^T \mathbf{x}_i) \mathbf{x}_i = \sum_{i=1}^{n} a_k \zeta_i \sum_{\ell=1}^{p} c_{\ell-1} \left( \sum_{j=1}^{d} w_{k,j} \, x_{i,j} \right)^{\ell-1} \mathbf{x}_i$$

to get the same set of constraints on variable $\zeta$, as in Equation 7. Making the number of constraints less than $\left\lceil \frac{n}{d'} \right\rceil$ leads to the existence of the desired subspace $\mathbf{U}$, whose dimension is $n\left(1 - \frac{1}{d'}\right)$, satisfying the constraints. This can be restated as

$$\binom{d' + p}{p} - 1 \leq \frac{n}{d'}$$

which can be further simplified using the fact that $d' \leq \mathcal{O}\left(\log^{0.75} n\right)$ to get

$$p \leq \mathcal{O}\left( \left(\frac{n}{d'}\right)^{\frac{1}{d'}} \right) = \mathcal{O}\left(n^{1/d'}\right).$$

In the above inequality, we use the fact that $1 \leq d'^{1/d'} \leq \sqrt{2}$. Since, the set of linear equation is independent of the choice of $\mathbf{w}$, the claim holds true for all $\mathbf{w} \in \{\mathbf{w}_1, ..., \mathbf{w}_m\}$, from which the result follows. $\square$

Now, if a function is well approximated by a low degree polynomial then we can use the idea from the above theorem to kill all but the small error term leaving us with a small eigenvalue. But, for different regimes of the parameters, we need to consider different polynomial approximations.

## F.2   $L^\infty$ Approximation using the Chebyshev Polynomials

Let $f : [-k, k] \to \mathbb{R}$ be a function for some $k > 0$. We would like to approximate $f$ with polynomials in the $L^\infty$ norm. That is, we would like a polynomial $g_p$ of degree $p$ such that

$$\sup_{x \in [-k,k]} \left| f(x) - g_p(x) \right| \leq \epsilon.$$

First, we reduce the above problem to that of approximating functions on $[-1, 1]$. The idea is to consider the scaled function $f_k(x) = f(kx)$. Note that $f_k : [-1, 1] \to \mathbb{R}$. Let $h$ be a polynomial approximating $f_k$ i.e. $\sup_x \left| f_k(x) - h(x) \right| \leq \epsilon$. Then, consider the function $g_p(x) = h\left(x/k\right)$. Then, $\left| f(x) - g_p(x) \right| = \left| f_k\left(x/k\right) - h(x/k) \right| \leq \epsilon$. Thus, we can consider approximation of functions on $[-1, 1]$.

We are interested in approximating the derivative of $\tanh$ on $(-\tau, \tau)$. Denote by $\sigma$ the sigmoid function given by

$$\sigma(x) = \frac{1}{1 + e^{-x}}.$$

It follows from the definition that $\tanh(x) = 2\sigma(2x) - 1$.

The approach is to consider a series in the Chebyshev polynomials $T_n$. That is, we consider

$$\sum_{i=0}^{N} a_n T_n$$

The coefficients $a_n$ corresponding to $\sigma$ can be computed using the orthogonality relations for the Chebyshev polynomials.

$$a_n = \frac{2}{\pi} \int_{-1}^{1} \frac{\sigma(x) T_n(x)}{\sqrt{1 - x^2}} dx.$$

Using this, one can show the following theorem about the polynomial approximations of the sigmoid function.

**Theorem F.4** (Equation B.7 in Shalev-Shwartz et al. (2011)). *For each $k \geq 0$ and $\epsilon \in (0, 1]$, there is a polynomial $g_p$ of degree $p$ with*

$$p = \left\lceil \frac{\log\left(\frac{4\pi + 2k}{\pi^2 \epsilon}\right)}{\log\left(1 + \pi k^{-1}\right)} \right\rceil$$

*such that*

$$\sup_{x \in [-1,1]} \left| g_p(x) - \sigma(kx) \right| \leq \epsilon.$$

The proof of the claim follows by bounding $a_n$ using contour integration. From the above discussion, in order to approximate tanh in the interval $(-\tau, \tau)$, we need to approximate $2\sigma(2\tau x) - 1$. From Theorem F.4, we require a polynomial of degree

$$p = \left\lceil \frac{\log\left(\frac{4\pi + 2\tau}{\pi^2 \epsilon}\right)}{\log\left(1 + \pi \tau^{-1}\right)} \right\rceil. \tag{8}$$

Recall that we actually need to approximate the derivative of tanh. But this can be achieved easily from the fact that $\tanh'(x) = \left(1 + \tanh(x)\right)\left(1 - \tanh(x)\right)$ and the following lemma.

**Lemma F.5.** *Let $I$ be an interval and let $f_i, g_i : I \to \mathbb{R}$ for $i \in \{1, 2\}$ be functions such that $\sup_{x \in I} \left| f_i(x) - g_i(x) \right| \leq \epsilon$ for all $i$ and $\left| f_i(x) \right| \leq 1$ for $x \in I$ and all $i$. Then,*

$$\sup_{x \in I} \left| f_1(x) f_2(x) - g_1(x) g_2(x) \right| \leq 3\epsilon.$$

*Proof.* For any $x \in I$ we have

$$
\begin{aligned}
\left| f_1(x) f_2(x) - g_1(x) g_2(x) \right| &\leq \left| f_1(x) f_2(x) - f_1(x) g_2(x) + f_1(x) g_2(x) - g_1(x) g_2(x) \right| \\
&\leq \left| f_1(x) f_2(x) - f_1(x) g_2(x) \right| + \left| f_1(x) g_2(x) - g_1(x) g_2(x) \right| \\
&\leq \left| f_1(x) \right| \left| f_2(x) - g_2(x) \right| + \left| g_2(x) \right| \left| f_1(x) - g_1(x) \right| \\
&\leq \epsilon + (1 + \epsilon)\epsilon \\
&\leq 2\epsilon + \epsilon^2 \\
&\leq 3\epsilon. \qquad \qquad \square
\end{aligned}
$$

### F.3 $L^2$ APPROXIMATION USING HERMITE POLYNOMIALS

Next we consider approximating $f$ in the 2-norm i.e. we would like to find a polynomial $h_p$ of degree $p$ such that

$$\int_{-\infty}^{\infty} \left| f(x) - h_p(x) \right|^2 d\mu(x)$$

is minimized. $\mu$ denotes the Gaussian measure on the real line. Note that this problem can be solved using the technique of orthogonal polynomials since $L^2(\mathbb{R}, \mu)$ is a Hilbert space. The study of orthogonal polynomial is a rich and well-developed area in mathematics (see Szegő (1975); Lebedev (1972)). Our main focus in this section will be the case where $f$ is the derivative of tanh and in later sections we extend this analysis to other activation functions.

Let $H_k$ denote the $k$-th normalized (physicists') Hermite polynomial given by

$$H_k(x) = \left[\sqrt{\pi}2^k k!\right]^{-1/2} (-1)^k e^{x^2} \frac{d^k}{dx^k} e^{-x^2}, \tag{9}$$

and the corresponding normalized (probabilists') Hermite polynomial is given by

$$He_k(x) = \left[\sqrt{\pi}k!\right]^{-1/2} (-1)^k e^{x^2/2} \frac{d^k}{dx^k} e^{-x^2/2}. \tag{10}$$

The Hermite polynomials are usually written without the normalization. The normalization terms ensure that the polynomials form orthonormal systems with respect to their measures. Recall that $\mu(x;\sigma)$ denotes the density function of a Gaussian variable with mean $0$ and standard deviation $\sigma$. The series of polynomials $H_k$ are orthonormal with respect to Gaussian measure $\mu\left(x;\frac{1}{\sqrt{2}}\right)$ and the series of polynomials $He_k$ are orthonormal with respect to the standard Gaussian measure $\mu(x;1)$ i.e.

$$\int_{-\infty}^{\infty} H_m(x) H_n(x) \, d\mu\left(x;\frac{1}{\sqrt{2}}\right) = \delta_{m,n}, \tag{11}$$

$$\int_{-\infty}^{\infty} He_m(x) He_n(x) \, d\mu(x;1) = \delta_{m,n}. \tag{12}$$

The two versions of the Hermite polynomials are related by

$$H_m(x) = He_m\left(\sqrt{2}x\right). \tag{13}$$

For any function $f \in L^2(\mu)$, we can define the Hermite expansion of the function as

$$f = \sum_{i=0}^{\infty} f_i He_i.$$

From the orthogonality of the Hermite polynomials, we can compute the coefficients as

$$f_i = \int f(x) He_i(x) \, d\mu(x;1).$$

Since $L^2$ is a Hilbert space, we can use the Pythagoras theorem to bound the error of the projection onto the space of degree $k$ polynomials as

$$\sum_{i=k+1}^{\infty} |f_i|^2.$$

This leads us to consider the Hermite coefficients of the functions we would like to study.

We defined the Hermite expansion in terms of probabilists' Hermite polynomials. For physicists' version we can define an expansion similarly. In this paper, we will use probabilists' version in our proofs. Since the literature we draw on comes from both conventions, we will need to talk about physicists' version also.

In the following we will be using complex numbers. For $z \in \mathbb{C}$, the imaginary part of $z$ is denoted by $\Im(z)$.

The following theorem provides conditions under which the Hermite coefficients decay rapidly. It says that if a function extends analytically to a strip around the real axis and the function decays sufficiently rapidly as the real part goes to infinity, the Hermite series converges uniformly over compact sets in the strip and consequently has rapidly decaying Hermite coefficients. The extension to the complex plane provide the function with strong regularity conditions.

**Theorem F.6.** *(Theorem 1 in Hille (1940)) Let $f(z)$ be an analytic function. A necessary and sufficient condition in order that the Fourier-Hermite series*

$$\sum_{k=0}^{\infty} c_k H_k(z) e^{-\frac{z^2}{2}}, \quad c_k = \int_{-\infty}^{\infty} f(t) H_k(t) e^{-\frac{t^2}{2}} \, dt \tag{14}$$

*shall exist and converge to the sum $f(z)$ in the strip $S_\tau = \left\{ z \in \mathbb{C} : \left| \Im(z) \right| < \tau \right\}$, is that $f(z)$ is holomorphic in $S_\tau$ and that to every $\beta$, $0 \leq \beta < \tau$, there exits a finite positive $B(\beta)$ such that*

$$\left| f(x + iy) \right| \leq B(\beta) e^{-|x|(\beta^2 - y^2)^{1/2}}$$

*where $x \in (-\infty, \infty)$, $y \in (-\beta, \beta)$. Moreover, whenever the condition is satisfied, we have*

$$|c_k| \leq M(\epsilon) e^{-(\tau - \epsilon)\sqrt{2k+1}} \tag{15}$$

*for all positive $\epsilon$. Here, $M$ denotes a function that depends only on $\epsilon$.*

The function tanh can be naturally extended to the complex plane using its definition in terms of the exponential function. From this definition, it follows that tanh has a simple pole at every point such that $e^{2z} + 1 = 0$. The set of solutions to this are given by $2z = (2n + 1)\pi$. Thus, tanh is holomorphic in any region not containing these singularities. In particular, tanh is holomorphic in the strip $S_{\pi/2} = \left\{ z \in \mathbb{C} : \left| \Im(z) \right| < \pi/2 \right\}$. The same holds for $\tanh'$.

Using the above theorem, we bound the size of the Hermite coefficients of the derivative of tanh and thus bound the error of approximation by low degree polynomials.

**Theorem F.7.** *Let $\phi_1(z) = \tanh'\left(z/\sqrt{2}\right) e^{-\frac{z^2}{2}}$. Consider the Hermite expansion of $\phi_1$ in terms of $\{H_k\}_{k=0}^\infty$, as*

$$\phi_1(z) = \sum_{k=0}^\infty c_k H_k(z) e^{-\frac{z^2}{2}}. \tag{16}$$

*Then,*

$$|c_k| \leq \mathcal{O}\left( e^{-\frac{\pi\sqrt{k}}{4}} \right).$$

*Proof.* As before consider the strip $S_\tau = \left\{ z \in \mathbb{C} : \left| \Im(z) \right| < \tau \right\}$. Note that $\phi_1(z)$ is holomorphic in $S_\tau$ for $\tau < \sqrt{2}\pi/2$. For every $\beta \in [0, \sqrt{2}\pi/4]$, consider $z = x + iy \in S_\beta$ and set $\sqrt{2}x' = x$ and $\sqrt{2}y' = y$. Also note that $\tanh'(z) = 1/\cosh^2(z)$. Then

$$\begin{aligned}
\left| \phi_1(z) \right| &= \left| \frac{1}{\cosh^2(x' + iy')} \right| \left| e^{-(x'+iy')^2} \right| \\
&= \left| \frac{1}{\cosh(x' + iy')} \right|^2 \left| e^{-x^2 + y^2 - 2ixy} \right| \\
&= \left| \frac{1}{\cos y' \cosh x' + i \sin y' \sinh x'} \right|^2 \left| e^{-x^2 + y^2 - 2ixy} \right| \\
&= \frac{1}{\cos^2 y' \cosh^2 x' + \sin^2 y' \sinh^2 x'} \left| e^{-x^2 + y^2 - 2ixy} \right| \\
&\leq \frac{e^{-x^2} e^{y^2} \left| e^{2ixy} \right|}{\cos^2 y' \cosh^2 x'} \\
&\leq \frac{e^{-x^2} e^{\beta^2}}{(\cos^2 \beta) \cosh^2 x'} \\
&\leq e^{-x^2} e^{\beta^2} (\sec^2 \beta) \operatorname{sech}^2 x' \\
&\leq 4 e^{-x^2} e^{\beta^2} (\sec^2 \beta) \left( e^{x'} + e^{-x'} \right)^{-2} \\
&\leq 4 e^{-x^2} e^{\beta^2} (\sec^2 \beta) e^{-\sqrt{2}|x|} \\
&\leq 40\, e^{\beta^2} (\sec^2 \beta) e^{-\sqrt{\beta^2 - y^2}|x|}.
\end{aligned}$$

The last inequality follows by noting that $\sqrt{\beta^2 - y^2} \le \beta \le \sqrt{2}\pi/4 \le \sqrt{2}$. This satisfies the required condition with $B(\beta) = 40\,e^{\beta^2}(\sec^2\beta)$. Hence, from Theorem F.6, we have that Equation 16 is convergent in the strip $S_\tau$, for $\tau \le \frac{\sqrt{2}\pi}{4}$. Thus, from Equation 15, using $\epsilon = \frac{\sqrt{2}\pi}{8}$ we have

$$|c_k| \le Ce^{-\frac{\pi}{4\sqrt{2}}\sqrt{2k+1}}$$

for some constant $C$ independent of $k$. $\hfill\square$

**Corollary F.7.1.** *Let* $\phi_2(x) = \tanh'(x)$. *Consider the Hermite expansion of* $\phi_2$:

$$\phi_2(x) = \sum_{k=0}^{\infty} \bar{c}_k He_k(x). \tag{17}$$

*Then,*

$$|\bar{c}_k| \le \mathcal{O}\left(e^{-\frac{\pi\sqrt{k}}{4}}\right).$$

*Proof.* Using orthonormality of $He_k$ with respect to the standard Gaussian measure, we have

$$\bar{c}_k = \int_{-\infty}^{\infty} \phi_2(x)\, He_k(x)\, d\mu(x; 1) = \frac{1}{\sqrt{2\pi}} \int_{-\infty}^{\infty} \phi_2(x)\, He_k(x) e^{-\frac{x^2}{2}}\, dx$$

$$= \sqrt{2}\frac{1}{\sqrt{2\pi}} \int_{-\infty}^{\infty} \phi_2\left(\sqrt{2}x\right) He_k\left(\sqrt{2}x\right) e^{-x^2}\, dx$$

Using $\phi_2\left(\sqrt{2}x\right) = \phi_1(x)\,e^{\frac{x^2}{2}}$, defined in Theorem F.7 and $He_k\left(\sqrt{2}x\right) = H_k(x)$, as given by Equation 13, we have

$$\bar{c}_k = \frac{1}{\sqrt{\pi}} \int_{-\infty}^{\infty} \phi_1(x)\, H_k(x)\, e^{-\frac{x^2}{2}}\, dx$$

Applying Theorem F.7, we get the required bound. $\hfill\square$

**Corollary F.7.2.** *Let* $\phi_2(x) = \tanh'(x)$ *and let* $\phi_2$ *be approximated by Hermite polynomials* $\{He_k\}_{k=0}^{\infty}$ *of degree up to* $p$ *in Equation 17, denoted by*

$$h_p(x) := \sum_{k=1}^{p} \bar{c}_k He_k(x).$$

*Let*

$$E_p(x) := \phi_2(x) - h_p(x).$$

*Then,*

$$\int_{-\infty}^{\infty} E_p(x)^2\, d\mu(x; 1) \le O\left(\sqrt{p}\,e^{-\frac{\pi}{4\sqrt{2}}\sqrt{p}}\right).$$

*Proof.*

$$E_p(x) = \phi_2(x) - h_p(x) = \sum_{k=p+1}^{\infty} \bar{c}_k He_k(x).$$

Using orthonormality of normalized Hermite polynomials with respect standard Gaussian measure, we have

$$\int_{-\infty}^{\infty} E_p(x)^2\, d\mu(x; 1) = \sum_{k=p+1}^{\infty} \int_{-\infty}^{\infty} \bar{c}_k^2 He_k(x) He_k(x)\, dx = \sum_{k=p+1}^{\infty} \bar{c}_k^2.$$

Substituting the bounds for $\{\bar{c}_k\}_{k=p+1}^{\infty}$ from Corollary F.7.1, we have

$$\sum_{k=p+1}^{\infty} \bar{c}_k^2 \le \sum_{k=p+1}^{\infty} e^{-\frac{\pi}{4\sqrt{2}}\sqrt{2k+1}}$$

$$\le \int_{p}^{\infty} e^{-\frac{\pi}{4\sqrt{2}}\sqrt{2x+1}}dx$$

$$= \frac{32}{\pi^2}\left(\frac{\pi}{4\sqrt{2}}\sqrt{2p+1} + 1\right)e^{-\frac{\pi}{4\sqrt{2}}\sqrt{2p+1}},$$

as required. $\hfill\square$

For comparison, consider the Hermite expansion of the derivative of ReLU, the threshold function. It can be shown (see (Lebedev, 1972, page 75)) that

$$
\begin{aligned}
\mathsf{ReLU}'(x) &= \frac{1}{2\sqrt{\pi}} \sum_{k=0}^{\infty} \frac{(-1)^k \sqrt{\sqrt{\pi} 2^{2k+1}(2k+1)!}}{2^{2k}(2k+1)\,k!} H_{2k+1}(x) + \frac{1}{2} \\
&= \frac{1}{\sqrt[4]{4\pi}} \sum_{k=0}^{\infty} \frac{(-1)^k \sqrt{(2k+1)!}}{2^k(2k+1)\,k!} H_{2k+1}(x) + \frac{1}{2} \\
&\approx \sum_{k=0}^{\infty} \frac{(-1)^k \sqrt{2^{2k}k^{2k}\sqrt{2k}e^{-2k}}}{2^k\sqrt{(2k+1)}k^k\sqrt{k}e^{-k}} H_{2k+1}(x) + \frac{1}{2} \\
&\approx \sum_{k=0}^{\infty} \frac{(-1)^k}{\sqrt{(2k+1)}\sqrt[4]{k}} H_{2k+1}(x) + \frac{1}{2} \\
&\approx \sum_{k=0}^{\infty} \frac{(-1)^k}{k^{0.75}} H_{2k+1}(x) + \frac{1}{2}.
\end{aligned}
\tag{18}
$$

We can also expand the threshold function, in terms of probabilist's Hermite polynomials in the following manner. If the expansion of threshold function is written as $\sum_{k=0}^{\infty} \bar{c}_k H_{e_k}(x)$, then

$$
\begin{aligned}
\bar{c}_k &= \frac{1}{\sqrt{2\pi}} \int_{-\infty}^{\infty} \mathsf{ReLU}'(x) H_{e_k}(x) e^{-\frac{x^2}{2}} dx \\
&= \frac{1}{\sqrt{\pi}} \int_{-\infty}^{\infty} \mathsf{ReLU}'(\sqrt{2}y) H_{e_k}(\sqrt{2}y) e^{-y^2} dy \\
&= \frac{1}{\sqrt{\pi}} \int_{-\infty}^{\infty} \mathsf{ReLU}'(y) H_k(y) e^{-y^2} dy
\end{aligned}
\tag{19}
$$

$$
= c_k \approx \frac{(-1)^k}{k^{0.75}}
\tag{20}
$$

where we use Equation 13 and the fact that $\mathsf{ReLU}'(y) = \mathsf{ReLU}'(\sqrt{2}y)$ in Equation 19.

It can now be seen that the Hermite coefficients do not decay rapidly for this function.

### F.3.1 DZPS SETTING

For this choice of initialization, defined in section 2, we need to consider two different regimes depending on the number of neurons $m$. When $m$ is small, we use Chebyshev approximation in the $L^\infty$ norm, while we use the $L^2$ approximation by Hermite polynomials when $m$ is large. First consider the Chebyshev approximation.

**Theorem F.8.** *Assuming $\phi(x) = \mathsf{tanh}(x)$ and weights $\mathbf{w}_k^{(0)} \sim \mathcal{N}(0, \mathbf{I}_d), a_k^{(0)} \sim \mathcal{N}(0,1)\ \forall k \in [m]$, the minimum eigenvalue of the $\mathbf{G}$-matrix is*

$$
\lambda_{\min}\left(\mathbf{G}^{(0)}\right) \le \mathcal{O}\left( n \left( \frac{4\pi + 6\sqrt{\log nm}}{\left(1 + \frac{\pi/3}{\sqrt{\log nm}}\right)^p} \right)^2 \right)
$$

*with probability at least $1 - \frac{2}{(mn)^{3.5}}$ with respect to $\left\{\mathbf{w}_k^{(0)}\right\}_{k=1}^m$ and $\left\{a_k^{(0)}\right\}_{k=1}^m$, where $p$ is the largest integer that satisfies Condition 6.*

*Proof.* In the following, for typographical reasons we will write $\mathbf{w}_k$ instead of $\mathbf{w}_k^{(0)}$ and $a_k$ instead of $a_k^{(0)}$. Referring to Equation 5, it suffices to find a vector $\zeta^g \in \mathbb{S}^{n-1}$ s.t. $\left\|\sum_{i=1}^n \zeta_i^g \mathbf{m}_i\right\|$ is small.

For each $k \in [m]$ and $i \in [n]$, $\mathbf{w}_k^T \mathbf{x}_i$ is a Gaussian random variable following $\mathcal{N}(0,1)$. Thus, there are $mn$ Gaussian random variables and with probability at least $\left(1 - \frac{2}{(mn)^{3.5}}\right)$ with respect to $\{\mathbf{w}_k\}_{k=1}^m$, $\max_{i\in[n],k\in[m]}\left|\mathbf{w}_k^T \mathbf{x}_i\right| \le 3\sqrt{\log nm}$. Hence, we restrict ourselves to the

range $\left(-3\sqrt{\log nm}, 3\sqrt{\log nm}\right)$, when we analyze $\phi(x)$. Now, from Equation 8 and Lemma F.5, we have that there exists a polynomial $g(x)$ of degree $p$ that can approximate $\phi'$ in the interval $\left(-3\sqrt{\log nm}, 3\sqrt{\log nm}\right)$ with the error of approximation in $L^\infty$ norm $\epsilon$ given by

$$\epsilon \leq 3\left(\frac{4\pi + 6\sqrt{\log nm}}{\left(1 + \frac{\pi/3}{\sqrt{\log nm}}\right)^p}\right).$$

From Theorem F.2, there exists $\zeta^g \in \mathbb{S}^{n-1}$ s.t.

$$\sum_{k=1}^{m}\left\|\sum_{i=1}^{n}\zeta_i^g a_k\, g\left(\mathbf{w}_k^T\mathbf{x}_i\right)\mathbf{x}_i\right\|_2^2 = 0$$

provided Condition 6 holds true. For any weight vector $\mathbf{w}_k$, it's corresponding output weight $a_k$ and any $\zeta \in \mathbb{S}^{n-1}$, we have

$$\left\|\sum_{i=1}^{n}\zeta_i\, a_k\phi'(\mathbf{w}_k^T\mathbf{x}_i)\mathbf{x}_i - \sum_{i=1}^{n}\zeta_i\, a_k g(\mathbf{w}_k^T\mathbf{x}_i)\mathbf{x}_i\right\|_2 \leq \sqrt{a_k^2\sum_{i=1}^{n}\zeta_i^2\left(\phi'(\mathbf{w}_k^T\mathbf{x}_i) - g(\mathbf{w}_k^T\mathbf{x}_i)\right)^2}\sqrt{\sum_{i=1}^{n}\|\mathbf{x}_i\|^2}$$
$$(21)$$
$$\leq \sqrt{n}\epsilon|a_k| \qquad (22)$$

where we use triangle inequality and the Cauchy-Schwartz inequality in Inequality 21, $\|\zeta\| = 1$, $\|\mathbf{x}_i\| = 1$ and that the maximum error of approximation is $\epsilon$ in Inequality 22. Hence, for $\zeta = \zeta^g$, we have

$$\left\|\sum_{i=1}^{n}\zeta_i^g a_k\phi'\left(\mathbf{w}_k^T\mathbf{x}_i\right)\mathbf{x}_i\right\|_2 \leq \left\|\sum_{i=1}^{n}\zeta_i^g a_k\phi'\left(\mathbf{w}_k^T\mathbf{x}_i\right)\mathbf{x}_i - \sum_{i=1}^{n}\zeta_i^g a_k g(\mathbf{w}_k^T\mathbf{x}_i)\mathbf{x}_i\right\|_2 + \left\|\sum_{i=1}^{n}\zeta_i^g a_k g(\mathbf{w}_k^T\mathbf{x}_i)\mathbf{x}_i\right\|_2$$
$$\leq \sqrt{n}|a_k|\,\epsilon \qquad (23)$$

Thus,

$$\left\|\sum_{i=1}^{n}\zeta_i^g\mathbf{m}_i\right\| = \sqrt{\sum_{k=1}^{m}\left\|\sum_{i=1}^{n}\zeta_i^g a_k\phi'\left(\mathbf{w}_k^T\mathbf{x}_i\right)\mathbf{x}_i\right\|^2} \leq \sqrt{\sum_{k=1}^{m}a_k^2}\sqrt{n}\epsilon \leq \sqrt{5m}\sqrt{n}\epsilon$$

where in the final step, we use chi-square concentration bounds (Fact C.6) to show that $\sqrt{\sum_{k=1}^{m}a_k^2}$ is at-most $\sqrt{5m}$ with probability at-least $1 - e^{-m}$. Using Equation 5, $\sigma_{\min}(\mathbf{M}) \leq \sqrt{5m}\sqrt{n}\epsilon$. That implies, $\lambda_{\min}(\mathbf{G}) = \frac{1}{m}\lambda_{\min}(\mathbf{M})^2 \leq 5n\epsilon^2$. Since, $\epsilon$ decreases with increasing $p$, we substitute the value of $\epsilon$ at maximum value of $p$ possible in order to get the desired result. $\qquad\square$

Note that since the upper bound on the eigenvalue depends on $m$, the bound becomes increasingly worse as we increase $m$. This is because as we increase $m$, the interval $\left(-3\sqrt{\log nm}, 3\sqrt{\log nm}\right)$ in which we need the polynomial approximation to hold increases in length and thus the degree needed to approximate grows with $m$. To remedy this, we relax the approximation guarantee required from the $L^\infty$ norm to the $L^2$ norm under the Gaussian measure. This naturally leads to approximation by Hermite polynomials as discussed in subsection F.3.

**Theorem F.9.** *Assuming $\phi(x) = \tanh(x)$ and weights $\mathbf{w}_k^{(0)} \sim \mathcal{N}(0, \mathbf{I}_d)$, $a_k^{(0)} \sim \mathcal{N}(0, 1)\ \forall k \in [m]$, with probability at least $1 - e^{-\Omega\left(\frac{mc^2}{n^2\log m}\right)} - m^{-3.5}$ over the choice of $\left\{\mathbf{w}_k^{(0)}\right\}_{k=1}^{m}$ and $\left\{a_k^{(0)}\right\}_{k=1}^{m}$ the minimum eigenvalue of the G-matrix is bounded by c, i.e.*

$$\lambda_{\min}(\mathbf{G}^{(0)}) \leq c, \text{ where}$$

$$c = \max\left(\mathcal{O}\left(\frac{n\log n\log m}{\sqrt{m}}\right), \mathcal{O}\left(n^2\sqrt{p}e^{-\frac{\pi}{4\sqrt{2}}\sqrt{p}}\right)\right)$$

*and $p$ is the largest integer that satisfies Condition 6.*

*Proof.* In the following, for typographical reasons we will write $\mathbf{w}_k$ instead of $\mathbf{w}_k^{(0)}$. Referring to Equation 5, it suffices to find a vector $\zeta^h \in \mathbb{S}^{n-1}$ s.t. $\left\|\sum_{i=1}^{n} \zeta_i^h \mathbf{m}_i\right\|$ is small.

Theorem G.2 gives the error of approximating $\phi'$ by a polynomial $h$, consisting of Hermite polynomials of degree $\leq p$, in the $L^2$ norm. Let $E_p$ denote the error function of approximation, given by $E_p(x) = \phi'(x) - h(x)$.

From Theorem F.2, there exists $\zeta^h \in \mathbb{S}^{n-1}$ s.t. for all $\mathbf{w}$ and $\tilde{a}$ we have

$$\sum_{i=1}^{n} \zeta_i^h \tilde{a} h(\mathbf{w}^T \mathbf{x}_i) \mathbf{x}_i = 0,$$

provided $p$ satisfies Condition 6.

We can use Fact C.5 to confine the maximum magnitude of $a_k$ to $3\sqrt{\log m}$. Thus, assuming that this condition holds true, we claim the following. Using $\zeta^h$, we get

$$\mathbb{E}_{\mathbf{w}\sim\mathcal{N}(0,\mathbf{I}_d),\tilde{a}\sim\mathcal{N}(0,1)}\left\|\sum_{i=1}^{n} \zeta_i^h \tilde{a}\phi'(\mathbf{w}^T \mathbf{x}_i)\mathbf{x}_i\right\|^2$$

$$= \mathbb{E}_{\mathbf{w}\sim\mathcal{N}(0,\mathbf{I}_d),\tilde{a}\sim\mathcal{N}(0,1)}\left\|\sum_{i=1}^{n} \zeta_i^h \tilde{a} h_p(\mathbf{w}^T \mathbf{x}_i)\mathbf{x}_i + \sum_{i=1}^{n} \tilde{a}\zeta_i^h E_p(\mathbf{w}^T \mathbf{x}_i)\mathbf{x}_i\right\|^2 \tag{24}$$

$$= \mathbb{E}_{\mathbf{w}\sim\mathcal{N}(0,\mathbf{I}_d),\tilde{a}\sim\mathcal{N}(0,1)}\left\|\sum_{i=1}^{n} \zeta_i^h \tilde{a} E_p(\mathbf{w}^T \mathbf{x}_i)\mathbf{x}_i\right\|^2 \tag{25}$$

$$\leq \mathbb{E}_{\mathbf{w}\sim\mathcal{N}(0,\mathbf{I}_d),\tilde{a}\sim\mathcal{N}(0,1)}\tilde{a}^2\left\|\sum_{i=1}^{n} \zeta_i^h \mathbf{x}_i\right\|^2\left(\sum_{i=1}^{n}\left(E_p\left(\mathbf{w}^T \mathbf{x}_i\right)\right)^2\right) \tag{26}$$

$$\leq n\sum_{i=1}^{n}\left\{\mathbb{E}_{\mathbf{w}\sim\mathcal{N}(0,\mathbf{I}_d)}\left(E_p\left(\mathbf{w}^T \mathbf{x}_i\right)\right)^2\right\} \tag{27}$$

$$= n^2\,\mathbb{E}_{\mathbf{w}\sim\mathcal{N}(0,\mathbf{I}_d)}\left(E_p\left(\mathbf{w}^T \mathbf{x}_1\right)\right)^2. \tag{28}$$

We approximate $\phi'$ by $h$ and use the definition of $\zeta^h$ in Equation 24, apply Cauchy-Schwartz in Equation 25, $\|\mathbf{x}_i\| = 1 \;\forall i \in [n], \|\zeta^h\| = 1$, the linearity of expectation in Equation 26 and maximum variance of $\tilde{a}$, given the constraint on it's magnitude, as upperbounded by 1 in Equation 27. $\mathbf{w}^T \mathbf{x}_1$ follows a Gaussian distribution $\mathcal{N}(0,1)$. Hence, denoting $\epsilon_h$ as the error of approximation from Corollary F.7.2, we get

$$\mathbb{E}_{\mathbf{w}\sim\mathcal{N}(0,\mathbf{I}_d),\tilde{a}\sim\mathcal{N}(0,1)}\left\|\sum_{i=1}^{n} \zeta_i^h \tilde{a}\phi'(\mathbf{w}^T \mathbf{x}_i)\mathbf{x}_i\right\|^2 \leq n^2\epsilon_h$$

where

$$\epsilon_h \leq \mathcal{O}(\sqrt{p}e^{-\frac{\pi}{4\sqrt{2}}\sqrt{p}}).$$

Applying Hoeffding's inequality for $m$ weight vectors $\mathbf{w}_k \sim \mathcal{N}(0,\mathbf{I}_d)$ and $a_k \sim \mathcal{N}(0,1)$, we get

$$\Pr_{\{\mathbf{w}_k\}_{k=1}^m,\{a_k\}_{k=1}^m}\left\{\frac{1}{m}\sum_{k=1}^{m}\left\|\sum_{i=1}^{n} \zeta_i^h a_k\phi'(\mathbf{w}_k^T \mathbf{x}_i)\mathbf{x}_i\right\|^2 \leq \left(n^2\epsilon_h + t\right)\right\} \geq 1 - e^{-\frac{2mt^2}{9n^2\log^2 m}} - m^{-3.5}.$$

In the above inequality, we use the restriction of the maximum magnitude of $a_k$ to $3\sqrt{\log m}$ and hence, $\forall k, \left\|\sum_{i=1}^{n} \zeta_i^h a_k\phi'\left(\mathbf{w}_k^T \mathbf{x}_i\right)\mathbf{x}_i\right\| \in \left(0, 3\sqrt{n}\sqrt{\log m}\right)$. Using $t = \max\left(n^2\epsilon_h, \frac{n\log n\log m}{\sqrt{m}}\right)$, c

being a constant and substituting the value of $\epsilon$, we get the final upper bound. Thus,

$$\sum_{k=1}^{m}\left\|\sum_{i=1}^{n}\zeta_i^h\phi'(\mathbf{w}_k^T\mathbf{x}_i)\mathbf{x}_i\right\|_2^2 \leq m\max\left(\mathcal{O}\left(n^2\epsilon_h\right),\mathcal{O}\left(\frac{n\log n\log m}{\sqrt{m}}\right)\right) \tag{29}$$

Using Equation 5 and the fact $\lambda_{\min}(\mathbf{G}) = \frac{1}{m}\sigma_{\min}(\mathbf{M})^2$, we get the final bound. $\qquad\square$

The upper bound on minimum eigenvalue from Theorem F.8 can be rewritten as

$$\lambda_{\min}(\mathbf{G}^{(0)}) \leq \mathcal{O}\left(n\log(nm)e^{-p\log\left(1+\frac{\pi}{3}\sqrt{\log(nm)}\right)}\right)$$

for a small constant $C$ and $p$ denotes the largest integer that satisfies Condition 6. Let us assume that $d' \leq \mathcal{O}\left(\log^{0.75}n\right)$, where $d' = \dim\left(\text{Span}\{\mathbf{x}_1\ldots\mathbf{x}_n\}\right)$. Then, we use the value of $p$ from Corollary F.2.1 for the next set of arguments. Substituting the value of $p$, we get

$$\lambda_{\min}(\mathbf{G}^{(0)}) \leq \mathcal{O}\left(n\log(nm)e^{-n^{1/d'}\log\left(1+\frac{\pi}{3}\sqrt{\log(nm)}\right)}\right)$$

Assuming $m < e^{\mathcal{O}\left(n^{1/d'}\right)}$, we have

$$\lambda_{\min}(\mathbf{G}^{(0)}) \leq \mathcal{O}\left(n^2 e^{-\Omega\left(n^{1/2d'}\right)}\right) = e^{-\Omega\left(n^{1/2d'}\right)}.$$

By Theorem F.9, when $m > e^{\Omega\left(n^{1/2d'}\right)}$,

$$\lambda_{\min}(\mathbf{G}^{(0)}) \leq \max\left(n^{1.5}\log(n)e^{-\Omega\left(n^{1/2d'}\right)},\mathcal{O}\left(n^2 e^{-\frac{\pi}{4\sqrt{2}}n^{1/2d'}}\right)\right) = e^{-\Omega\left(n^{1/2d'}\right)}$$

with high probability with respect to $\{\mathbf{w}_k\}_{k=1}^m$ and $\{a_k\}_{k=1}^m$. Hence, the final bounds of minimum singular value of Gram matrix in case of tanh activation has been summarized below.

**Theorem F.10.** *Let* $d' = \dim\left(\text{span}\{\mathbf{x}_1\ldots\mathbf{x}_n\}\right) \leq \mathcal{O}\left(\log^{0.75}n\right)$. *Assuming* $\phi(x) = \tanh(x)$ *and weights* $\mathbf{w}_k^{(0)} \sim \mathcal{N}(0,\mathbf{I}_d)$, $a_k^{(0)} \sim \mathcal{N}(0,1)$ $\forall k \in [m]$, *the minimum eigenvalue of the G-matrix satisfies*

$$\lambda_{\min}\left(\mathbf{G}^{(0)}\right) \leq e^{-\Omega(n^{1/2d'})}$$

*with probability at least* $1 - \frac{1}{n^{3.5}}$ *with respect to the weight vectors* $\left\{\mathbf{w}_k^{(0)}\right\}_{k=1}^m$ *and* $\left\{a_k^{(0)}\right\}_{k=1}^m$.

In fact, as in Theorem F.3, we can show that the smallest $\left(1 - \frac{1}{d'}\right)n$ eigenvalues of the matrix are small. This is captured in the following theorem.

**Corollary F.10.1.** *Let* $d' = \dim\left(\text{span}\{\mathbf{x}_1\ldots\mathbf{x}_n\}\right) = \mathcal{O}\left(\log^{0.75}n\right)$. *Assuming* $\phi(x) = \tanh(x)$ *and weights* $\mathbf{w}_k^{(0)} \sim \mathcal{N}(0,\mathbf{I}_d)$, $a_k^{(0)} \sim \mathcal{N}(0,1)$ $\forall k \in [m]$, *then eigenvalues of the G-matrix satisfy*

$$\lambda_k\left(\mathbf{G}^{(0)}\right) \leq e^{-\Omega\left(n^{1/2d'}\right)} \quad \forall k \geq \left\lceil\frac{n}{d'}\right\rceil$$

*with probability at least* $1 - \frac{1}{n^{2.5}}$ *with respect to the weight vectors* $\left\{\mathbf{w}_k^{(0)}\right\}_{k=1}^m$ *and* $\left\{a_k^{(0)}\right\}_{k=1}^m$.

*Proof.* In the following, for typographical reasons we will write $\mathbf{w}_k$ instead of $\mathbf{w}_k^{(0)}$ and $a_k$ instead of $a_k^{(0)}$. We give a proof outline. We will approximate $\phi'$ by a $p$-degree polynomial $h$ as in Theorem F.8

and Theorem F.9, where $p \leq \mathcal{O}\left(n^{1/d'}\right)$. From Theorem F.3, we get that for polynomial $h$, there exits a $n(1 - \frac{1}{d'})$ dimensional subspace $\mathbf{U}$ for which the following quantity

$$\sqrt{\sum_{k=1}^{m} \left\| \sum_{i=1}^{n} \zeta_i a_k h\left(\mathbf{w}_k^T \mathbf{x}_i\right) \mathbf{x}_i \right\|^2} = 0$$

is 0, $\forall \zeta \in \mathbf{U}$. Now, we can take an orthonormal basis $\zeta^{(\mathbf{U})} = \left[\zeta^{(1)}, \zeta^{(2)}, ..., \zeta^{\left(n\left(1 - \frac{1}{d'}\right)\right)}\right]$ of $\mathbf{U}$ and for each $\zeta^{(j)}$, we can follow the same proof structure in Theorem F.8, Theorem F.9 and Theorem F.10 to get

$$\frac{1}{m}\sum_{k=1}^{m} \left\| \sum_{i=1}^{n} \zeta_i^{(j)} a_k \phi'\left(\mathbf{w}_k^T \mathbf{x}_i\right) \mathbf{x}_i \right\|^2 \leq e^{-\Omega\left(n^{1/2d'}\right)}$$

with probability at-least $1 - \frac{1}{n^{2.5}}$ with respect to $\{\mathbf{w}_k\}_{k=1}^m$ and $\{a_k\}_{k=1}^m$. Now, for bounding singular value $\sigma_k(M)$, for $k \geq \left\lceil \frac{n}{d'} \right\rceil$, we use the following argument. We choose a subset $\mathcal{S}_{(n-k)}$ of size $n - k$ from $\zeta^{(\mathbf{U})}$. This subset is a $n - k$ dimensional subspace $\mathbf{U}'$ of $\mathbb{R}^n$. Each $\zeta \in \mathbf{U}'$ can be written in the form

$$\zeta = \sum_{j \in [n] : \zeta^{(j)} \in \mathbf{U}'} \alpha_j \zeta^{(j)}$$

with $\sum_{j \in [n] : \zeta^{(j)} \in \mathbf{U}'} \alpha_j^2 = 1$. Then, for each $\zeta \in \mathbf{U}'$,

$$\frac{1}{m}\sum_{k=1}^{m} \left\| \sum_{i=1}^{n} \zeta_i a_k \phi'\left(\mathbf{w}_k^T \mathbf{x}_i\right) \mathbf{x}_i \right\|^2 = \frac{1}{m}\sum_{k=1}^{m} \left\| \sum_{i=1}^{n} \sum_{j \in [n] : \zeta^{(j)} \in \mathbf{U}'} \alpha_j \zeta_i^{(j)} a_k \phi'\left(\mathbf{w}_k^T \mathbf{x}_i\right) \mathbf{x}_i \right\|^2$$

$$= \frac{1}{m}\sum_{k=1}^{m} \left\| \sum_{j \in [n] : \zeta^{(j)} \in \mathbf{U}'} \alpha_j \left( \sum_{i=1}^{n} \zeta_i^{(j)} a_k \phi'\left(\mathbf{w}_k^T \mathbf{x}_i\right) \mathbf{x}_i \right) \right\|^2$$

$$\leq \frac{1}{m}\sum_{k=1}^{m} \left( \sum_{j \in [n] : \zeta^{(j)} \in \mathbf{U}'} \alpha_j^2 \right) \left( \sum_{j \in [n] : \zeta^{(j)} \in \mathbf{U}'} \left\| \left( \sum_{i=1}^{n} \zeta_i^{(j)} a_k \phi'\left(\mathbf{w}_k^T \mathbf{x}_i\right) \mathbf{x}_i \right) \right\|^2 \right)$$

$$\leq \left( n\left(1 - \frac{1}{d'}\right)\right) e^{-\Omega\left(n^{1/2d'}\right)} = e^{-\Omega\left(n^{1/2d'}\right)} \tag{30}$$

Thus, it follows from the definition of $\sigma_k(M)$ from Fact C.7 that

$$\sigma_k(\mathbf{M}) \leq \sqrt{m} e^{-\Omega\left(n^{1/4d'}\right)}$$

Using $\lambda_k\left(\mathbf{G}^{(0)}\right) = \frac{1}{m}\sigma_k(\mathbf{M})^2$, we get the final upper bound. $\qquad \square$

### F.3.2 STANDARD SETTING

Now, we consider upper bounding the eigenvalue of the $G$-matrix for the standard initialization, defined in Appendix D.

**Theorem F.11** (Init(fanout) setting). *Assuming* $\phi(x) = \tanh(x)$ *and weights* $\mathbf{w}_k^{(0)} \sim \mathcal{N}\left(0, \frac{1}{m}\mathbf{I}_d\right), a_k^{(0)} \sim \mathcal{N}(0,1)$ *and* $b_k^{(0)} \sim \mathcal{N}\left(0, \frac{1}{m}\right) \forall k \in [m]$, *the minimum eigenvalue of the* $G$-*matrix is*

$$\mathcal{O}\left( n \left( \frac{4\pi + 6\sqrt{\frac{\log nm}{m}}}{\left(1 + \frac{\pi/3}{\sqrt{\frac{\log nm}{m}}}\right)^p} \right)^2 \right)$$

with probability at least $1 - \frac{2}{(mn)^{3.5}}$ with respect to $\left\{ \mathbf{w}_k^{(0)} \right\}_{k=1}^m$, $\left\{ a_k^{(0)} \right\}_{k=1}^m$ and $\left\{ b_k^{(0)} \right\}_{k=1}^m$, where $p$ is the largest integer that satisfies Condition 6.

*Proof.* For each $k \in [m]$ and $i \in [n]$, $\mathbf{w}_k^T \mathbf{x}_i$ is a Gaussian random variable following $\mathcal{N}\left(0, \frac{1}{m}\right)$. Thus, there are $mn$ Gaussian random variables and it follows from Fact C.5 that with probability at least $\left(1 - \frac{2}{(mn)^{3.5}}\right)$ with respect to $\{\mathbf{w}_k\}_{k=1}^m$, $\max_{i \in [n], k \in [m]} \left| \mathbf{w}_k^T \mathbf{x}_i \right| \leq 3\sqrt{\frac{\log nm}{m}}$. Now, we follow the same proof as F.8 but restricted to the range $\left(-3\sqrt{\frac{\log nm}{m}}, 3\sqrt{\frac{\log nm}{m}}\right)$ to get the desired result. $\qquad\square$

**Corollary F.11.1** (Init(fanout) setting). *If $d' = \dim\left(\mathrm{Span}\{\mathbf{x}_1 \ldots \mathbf{x}_n\}\right) \leq \mathcal{O}\left(\log^{0.75} n\right)$. Assuming $\phi(x) = \tanh(x)$ and weights $\mathbf{w}_k^{(0)} \sim \mathcal{N}\left(0, \frac{1}{m}\mathbf{I}_d\right), a_k^{(0)} \sim \mathcal{N}(0,1)$ and $b_k^{(0)} \sim \mathcal{N}\left(0, \frac{1}{m}\right) \forall k \in [m]$, the minimum eigenvalue of the $\mathbf{G}$-matrix satisfies*

$$\lambda_{\min}\left(\mathbf{G}^{(0)}\right) \leq e^{-\Omega(n^{1/2d'})}$$

*with probability at least $1 - \frac{1}{n^3}$ with respect to the weight vectors $\left\{ \mathbf{w}_k^{(0)} \right\}_{k=1}^m$, $\left\{ a_k^{(0)} \right\}_{k=1}^m$ and $\left\{ b_k^{(0)} \right\}_{k=1}^m$.*

*Proof.* It follows from the same proof as Theorem F.3. $\qquad\square$

**Theorem F.12** (Init(fanin) setting). *Assuming $\phi(x) = \tanh(x)$ and weights $\mathbf{w}_k^{(0)} \sim \mathcal{N}\left(0, \frac{1}{d}\mathbf{I}_d\right)$, $a_k^{(0)} \sim \mathcal{N}\left(0, \frac{1}{m}\right)$ and $b_k^{(0)} \sim \mathcal{N}(0, 0.01) \forall k \in [m]$, the minimum eigenvalue of the $\mathbf{G}$-matrix is*

$$\lambda_{\min}\left(\mathbf{G}^{(0)}\right) \leq \mathcal{O}\left( n^2 \left( \frac{4\pi + 6\sqrt{\frac{\log nm}{d}}}{\left(1 + \frac{\pi/3}{\sqrt{\frac{\log nm}{d}}}\right)^p} \right)^2 \right)$$

*with probability at least $1 - \frac{2}{(mn)^{3.5}}$ with respect to $\left\{ \mathbf{w}_k^{(0)} \right\}_{k=1}^m$, $\left\{ a_k^{(0)} \right\}_{k=1}^m$ and $\left\{ b_k^{(0)} \right\}_{k=1}^m$, where $p$ is the largest integer that satisfies Condition 6.*

*Proof.* The proof follows from the proofs of Theorem F.8 and Theorem F.9, with the region of approximation reduced to $\left(-3\sqrt{\frac{\log nm}{d}}, 3\sqrt{\frac{\log nm}{d}}\right)$. $\qquad\square$

**Corollary F.12.1** (Init(fanin) setting). *If $d' = \dim\left(\mathrm{Span}\{\mathbf{x}_1 \ldots \mathbf{x}_n\}\right) \leq \mathcal{O}\left(\log^{0.75} n\right)$. Assuming $\phi(x) = \tanh(x)$ and weights $\mathbf{w}_k^{(0)} \sim \mathcal{N}\left(0, \frac{1}{d}\mathbf{I}_d\right)$, $a_k^{(0)} \sim \mathcal{N}\left(0, \frac{1}{m}\right)$ and $b_k^{(0)} \sim \mathcal{N}(0, 0.01) \forall k \in [m]$, the minimum eigenvalue of the $\mathbf{G}$-matrix satisfies*

$$\lambda_{\min}\left(\mathbf{G}^{(0)}\right) \leq e^{-\Omega(\sqrt{d}n^{1/2d'})}$$

*with probability at least $1 - \frac{1}{n^3}$ with respect to the weight vectors $\left\{ \mathbf{w}_k^{(0)} \right\}_{k=1}^m$, $\left\{ a_k^{(0)} \right\}_{k=1}^m$ and $\left\{ b_k^{(0)} \right\}_{k=1}^m$.*

## G   UPPER BOUND ON LOWEST EIGENVALUE FOR SWISH

In this section, we show upper bounds on the eigenvalues for the $G$-matrix for the Swish activation function using techniques largely similar to the techniques uses for the $\tanh$ activation function. This is not too surprising since they satisfy the following functional identity

$$\mathsf{swish}(x) = \frac{x}{2}\left[\tanh\left(\frac{x}{2}\right) + 1\right].$$

Hence

$$\mathsf{swish}'(x) = \frac{1}{2}\left[1 + \frac{x}{2}\tanh'\left(\frac{x}{2}\right) + \tanh\left(\frac{x}{2}\right)\right].$$

**Theorem G.1.** $\mathsf{swish}'(t)$ *is approximated by a degree $p$ polynomial $g_p(t)$ within error $\epsilon$ in the interval $[-k, k]$ in the $L^\infty$ norm:*

$$\sup_{t\in[-k,k]} \left|\mathsf{swish}'(t) - g_p(t)\right| \le \epsilon$$

*where*

$$p = \left\lceil \frac{\log\left(\frac{4\pi k + 2k^2}{\pi^2 \epsilon}\right)}{\log\left(1 + \pi k^{-1}\right)} \right\rceil.$$

Similarly, for the $L^2$ approximation for swish, we proceed using the same technique as for $\tanh$.

**Theorem G.2.** *Let $\phi_2(x) = \mathsf{swish}'(x)$ and let $\phi_2$ be approximated by Hermite polynomials $\{He_k\}_{k=0}^\infty$ of degree up to $p$ in Equation 17, denoted by*

$$h_p(x) = \sum_{k=1}^{p} \bar{c}_k He_k(x).$$

*Let*

$$E_p(x) = \phi_2(x) - h_p(x).$$

*Then,*

$$\int_{-\infty}^{\infty} E_p(x)^2 \, d\mu(x; 1) \le O\left(\sqrt{p}\, e^{-\frac{\pi}{4\sqrt{2}}\sqrt{p}}\right).$$

Using the above theorems and the techniques from the previous sections, we can upper bound the eigenvalues of the $G$-matrix with the swish activation f0unction. We summarize this in the following theorems.

**Theorem G.3.** *Consider the setting of Du et al. (2019a). If $d' = \dim\left(\mathrm{Span}\left\{\mathbf{x}_1 \ldots \mathbf{x}_n\right\}\right) \le \mathcal{O}\left(\log^{0.75} n\right)$. Assuming $\phi(x) = \mathsf{swish}(x)$ and weights $\mathbf{w}_k^{(0)} \sim \mathcal{N}(0, \mathbf{I}_d) \forall k \in [m]$, then eigenvalues of the $\mathbf{G}$-matrix satisfy*

$$\lambda_k\left(\mathbf{G}^{(0)}\right) \le e^{-\Omega\left(n^{1/2d'}\right)} \quad \forall k \ge \left\lceil\frac{n}{d'}\right\rceil$$

*with probability at least $1 - \frac{1}{n^{2.5}}$ with respect to the weight vectors $\left\{\mathbf{w}_k^{(0)}\right\}_{k=1}^{m}$ and $\left\{a_k^{(0)}\right\}_{k=1}^{m}$.*

## H   A DISCUSSION ON UPPER BOUND OF LOWEST EIGENVALUE FOR GENERAL ACTIVATION FUNCTIONS

In this section, we generalize the results of the previous sections upper bounding the eigenvalues of the $G$-matrix to a more general class of activation functions. To this end we note that the only property of the $\tanh$ and swish we used was that these functions are well-approximated by polynomials of low degree. The approximation theorems used in the previous sections can be stated under fairly general conditions on the activation functions.

For the Chebyshev approximation, it can be shown that a function with $k$ derivatives with bounded norms can be approximated by Chebyshev polynomials of degree $N$ with error that decays like $N^{-k}$. This shows that for smooth functions the error decays faster than any inverse polynomial. Under the assumption of analyticity, this can be further improved to get exponential decay of error. We summarize this in the following theorem.

**Theorem H.1** (see Section 5.7 in Mason & Handscomb (2002)). *Let $f : [-1, 1] \to \mathbb{R}$ be a function with $k + 1$ continuous derivatives. Let $S_N f$ be the Chebyshev approximation of $f$ to degree $N$. Then, we have*

$$\sup_{x \in [-1,1]} \left| f(x) - (S_N f)(x) \right| \leq \mathcal{O}\left( N^{-k} \right).$$

*Furthermore, if $f$ can be extended analytically to the ellipse*

$$E_r = \left\{ z \in \mathbb{C} : z = \frac{(w + w^{-1})}{2} \quad |w| \leq r \right\},$$

*then*

$$\sup_{x \in [-1,1]} \left| f(x) - (S_N f)(x) \right| \leq \mathcal{O}\left( r^{-N} \right).$$

Similarly, for Hermite approximation one can state the decay of the Hermite coefficients in terms of the regularity of the derivatives, expressed in terms of inclusion of the function in certain Sobolev spaces. Also, Theorem F.6 indicates that extending the function on to the complex plane gives better convergence properties. See Thangavelu (1993) for further details.

With these general approximation theorems and techniques from the previous sections, we can extend the upper bound on the eigenvalues on activation functions satisfying sufficient regularity conditions.

## I  LOWER BOUND ON EIGENVALUES FOR tanh WHEN THE DIMENSION OF THE DATA IS NOT TOO SMALL

For the following proof, we assume the data generation process as follows.

**Assumption 3.** *The data is mildly generic as in smoothed analysis: e.g., $\mathbf{x}_i$ is obtained by adding small multiple $\left( \sigma = \mathcal{O}\left( \frac{\delta}{2\sqrt{n}} \right) \right)$ of IID standard Gaussian noise within the subspace $\mathbf{V}'$ of arbitrary initial samples $\mathbf{x}'_i$, with $\mathbf{V}' := \mathrm{Span}\{\mathbf{x}'_1, \ldots, \mathbf{x}'_n\}$ and renormalizing to 1. Denoting the initial set of samples as $\mathbf{X}' = \{\mathbf{x}'_1, \ldots \mathbf{x}'_n\}$ and the noise matrix $\mathbf{N}$, that has each entry coming iid from $\mathcal{N}\left(0, \sigma^2\right)$, we have the data matrix $\mathbf{X}$ defined by $\{\mathbf{x}_1, \ldots \mathbf{x}_n\}$, where*

$$\mathbf{x}_i = \frac{\mathbf{n}_i + \mathbf{x}'_i}{\left\| \mathbf{n}_i + \mathbf{x}'_i \right\|}$$

*Remark.* Assuming that the initial samples $\mathbf{x}'_i$ are one-normalized, w.h.p. the norm of the noisy vectors $\mathbf{x}'_i + \mathbf{n}_i$ are in the range $(1 - \delta, 1 + \delta)$ and thus, renormalization involves division by a constant in the range $(\frac{1}{1+\delta}, \frac{1}{1-\delta})$. Assuming that the initial samples $\mathbf{x}'_i$ are $2\delta$ separated, w.h.p. the separation between $\mathbf{x}_i$ can be shown to be at least $\delta$, thus satisfying Assumption 2.

**Assumption 4.** *Let $d' = \mathrm{span}\{\mathbf{x}_1, \ldots, \mathbf{x}_n\}$. For simplicity, we assume $d = d'$ i.e. $\mathbf{x}_1, ..., \mathbf{x}_n$ lie in $d'$-dimensional space (otherwise we project them to $d'$ dimensional space using SVD) and $d' \geq 2$.*

While our result here builds upon the smoothed analysis of Anderson et al. (2014), the following lemma provides a more modular approach though no new essential technical ingredient.

**Lemma I.1** (cf. Lemma H.1 in Oymak & Soltanolkotabi (2019))**.** *For an activation function $\phi$ and a data matrix $\mathbf{X} \in \mathbb{R}^{d \times n}$ with unit Euclidean norm columns, the minimum eigenvalue of the Gram matrix $\mathbf{G}^\infty$, satisfies the following inequality*

$$\lambda_{\min}\left(\mathbf{G}^\infty\right) \geq \bar{c}_r^2\left(\phi'\right) \lambda_{\min}\left(\left(\mathbf{X}^T\mathbf{X}\right)^{\odot(r+1)}\right), \quad \forall r \geq 0$$

*where $\left(\mathbf{X}^T\mathbf{X}\right)^{\odot(r+1)}$ is given by $\left(\mathbf{X}^{*r}\right)^T \mathbf{X}^{*r}$, $\mathbf{X}^{*r} \in \mathbb{R}^{n \times d^r}$ denotes the Khatri-Rao product of matrix $\mathbf{X}$ and $\bar{c}_r(\phi')$ denotes the $r$-th coefficient in the probabilists' Hermite expansion of $\phi'$.*

*Proof.* Each element of $\mathbf{G}^\infty$ can be expressed in the following manner.

$$g_{ij}^\infty = E_{\mathbf{w} \sim \mathcal{N}(0,1), \tilde{a} \sim \mathcal{N}(0,1)} \tilde{a}^2 \phi'\left(\mathbf{w}^T\mathbf{x}_i\right) \phi'\left(\mathbf{w}^T\mathbf{x}_j\right) \mathbf{x}_i^T\mathbf{x}_j = \sum_{a=0}^\infty \bar{c}_a^2\left(\phi'\right) \left(\mathbf{x}_i^T\mathbf{x}_j\right)^{a+1}$$

where we use a) unit variance of $\tilde{a}$ and independence of $\tilde{a}$ and $\mathbf{w}$ and b) the fact that $\mathbf{w}^T\mathbf{x}_i$ and $\mathbf{w}^T\mathbf{x}_j$ are $\mathbf{x}_i^T\mathbf{x}_j$ correlated for a normally distributed vector $\mathbf{w}$ and hence, use Lemma N.4. Thus,

$$G^\infty = \sum_{a=0}^\infty \bar{c}_a^2\left(\phi'\right) \left(\mathbf{X}^T\mathbf{X}\right)^{\odot(a+1)}$$

Using Weyl's inequality (Fact C.11) for the sum of PSD matrices $\left\{ \left(\mathbf{X}^T\mathbf{X}\right)^{\odot(a)} \right\}_{a=1}^\infty$, we get

$$\mathbf{G}^\infty \succeq \bar{c}_r^2\left(\phi'\right) \left(\mathbf{X}^T\mathbf{X}\right)^{\odot(r+1)}, \forall r \geq 0$$

from which, the assertion follows.  □

**Lemma I.2.** *Let $d' = \dim \mathrm{span}\{\mathbf{x}_1, \ldots, \mathbf{x}_n\}$. Denote by $p$ be an integer that satisfies*

$$\binom{d'}{p} \geq n. \tag{31}$$

*Then for any $\kappa \in (0, 1)$, with probability at least $1 - \kappa$ with respect to the noise matrix $\mathbf{N}$,*

$$\sigma_n\left(\mathbf{X}^{*p}\right) \geq \Omega\left( \frac{1}{\sqrt{n}} \left( \frac{\sigma\kappa}{np} \right)^p \right).$$

*Proof.* $\mathbf{X}^{*p}$ has $d'^p$ rows and $n$ columns. However, the number of distinct rows is given by $\binom{d'+p-1}{p}$. This follows by counting the number of distinct terms in the polynomial $\left(\sum_{k=1}^{d'} v_k\right)^p$, where $\{v_k\}_{k=1}^{d'}$ is a set of $d'$ variables. Since, we assume that $\binom{d'}{p}$, which is lesser than this quantity, is greater than $n$, the number of distinct rows is greater than the number of columns for the matrix $\mathbf{X}^{*p}$. Let $\hat{\mathbf{X}}^{*p}$ denotes a $n \times n$ sized square block of $\mathbf{X}^{*p}$, that contains any random subset of size $n$ from the set of distinct rows of $\mathbf{X}^{*p}$ as rows, such that each row represents elements of Khatri-Rao product of the form $\prod_{j\in[d']} x_j^{b_j}$, $0 \le b_j \le 1 \,\forall j \in [d']$, for a vector $\mathbf{x} \in \mathbb{R}^{d'}$. We can see that $\lambda_n(\hat{\mathbf{X}}^{*p}) \le \sigma_n(\mathbf{X}^{*p})$. Hence, we will focus on the minimum eigenvalue $\lambda_n(\hat{\mathbf{X}}^{*p})$.

Fix $k \in [n]$ and let $\mathbf{u}$ be the vector orthogonal to the subspace spanned by the columns of $\hat{\mathbf{X}}^{*p}$, except the $k^{th}$ column. Vector $\mathbf{u}$ is well-defined with probability 1. Then the distance between $\hat{\mathbf{x}}_k^{*p}$ and the span of the rest of the columns, denoted $\mathrm{dist}(\hat{\mathbf{x}}_k^{*p}, \hat{\mathbf{X}}_{-k}^{*p})$, is given by

$$\mathbf{u}^T \hat{\mathbf{x}}_k^{*p} = \sum_{s\in[n]} u_s g_s\left(\left\{cx'_{kj} + cn_{kj}\right\}_{j=1}^{d'}\right) \tag{32}$$

$$=: P\left(\left\{cn_{kj}\right\}_{j=1}^{d'}\right), \tag{33}$$

where $g_s$ denotes a degree-$p$ polynomial and is given by

$$g_s\left(\left\{cx'_{kj} + cn_{kj}\right\}_{j=1}^{d'}\right) = \prod_{j\in[d']}\left(cx'_{kj} + cn_{kj}\right)^{b_j^s}, \quad 0 \le b_j^s \le 1 \,\forall j \in [d'], \sum_{j\in[d']} b_j^s = p.$$

$c$ denotes a constant in the range $(\frac{1}{1+\delta}, \frac{1}{1-\delta})$, which is the normalization factor used in Assumption 3.

Hence, Equation 32 is a degree $p$ polynomial in variables $n_{kj}$. We will apply the anticoncentration inequality of Carbery-Wright to show that the distance between any column and the span of the rest of the columns is large with high probability. The variance of the polynomial is given by

$$\mathrm{Var}\left(P\left(\{n_{kj}\}_{j=1}^{d'}\right)\right) \ge \sum_{s\in[n]} |u_s|^2 \prod_{j\in[d']} \mathbb{E}\left(cx'_{kj} + cn_{kj}\right)^{2b_j^s} \ge \left(\frac{\sigma}{1+\delta}\right)^{2p} \ge \left(\frac{\sigma}{2}\right)^{2p}$$

where we use the fact that $\|u\| = 1$ and $n_{kj}$ are Gaussian variables of variance $\frac{\delta^2}{4n}$. Using a minor adjustment of Fact C.8, which takes into consideration the fact that our gaussian variables are of variance $\frac{\delta^2}{4n}$ and the variance of the polynomial is not 1, we have

$$\Pr\left\{\left|P\left(\{n_{kj}\}_{j=1}^{d'}\right)\right| \le \epsilon\right\} \le Cp\frac{\epsilon^{1/p}}{\frac{\sigma}{2}}, \quad C > 0 \text{ is a constant.}$$

Using a union bound over the choice of $k$, we get

$$\Pr\left\{\mathrm{dist}(\hat{\mathbf{x}}_k^{*p}, \hat{\mathbf{X}}_{-k}^{*p}) \le \epsilon, \quad \forall k \in [n]\right\} \le Cpn\frac{\epsilon^{1/p}}{\frac{\sigma}{2}}$$

Using $\epsilon = \left(\frac{\sigma\kappa}{2Cpn}\right)^p$, we get

$$\sigma_n\left(\hat{\mathbf{X}}^{*p}\right) = \frac{1}{\sqrt{n}} \min_{k\in[n]} \mathrm{dist}(\hat{\mathbf{x}}_k^{*p}, \hat{\mathbf{X}}_{-k}^{*p}) \ge \epsilon/\sqrt{n}.$$

with probability at least $1 - \kappa$. We use Fact C.9 in the above inequality. $\square$

**Theorem I.3.** *Let $\phi(x)$ be a constant degree $p$ polynomial, with leading coefficient 1, and $d' = \dim \mathrm{span}\{\mathbf{x}_1, \ldots, \mathbf{x}_n\} \ge \Omega(n^{1/p})$. Then for any $\kappa \in (0, 1)$ we have*

$$\lambda_{\min}\left(\mathbf{G}^{(0)}\right) \ge \Omega\left(\frac{1}{n}\left(\frac{\kappa\sigma}{np}\right)^{2p}\right)$$

with probability at least $1 - \kappa$ w.r.t. the noise matrix $\mathbf{N}$ and $\{\mathbf{w}_k^{(0)}\}_{k=1}^m$, provided

$$m \geq \Omega\left(\frac{p^{4p}n^{4p+4}\log\left(n/\kappa\right)\log^{2p+3}m}{\sigma^{4p}\kappa^{4p}}\right).$$

*Proof.* For a degree-$p$ polynomial with leading coefficient 1, the $(p-1)$-th coefficient in Hermite expansion of $\phi'$ is given by 1. Also, note that Equation 31 is satisfied by $p$, given that $d' \geq \Omega\left(n^{\frac{1}{p}}\right)$ for a constant $p$. Thus, using Lemma I.1, Lemma I.2 to find minimum eigenvalue of $\mathbf{G}^{\infty}$ and then applying Hoeffding's inequality (Fact C.3) to bound the deviation of minimum eigenvalue of $\mathbf{G}^{(0)}$ from $\mathbf{G}^{\infty}$, we get the desired bound. $\square$

**Theorem I.4.** *Let the activation function $\phi$ be $\tanh$ and $d' = \dim\text{span}\{\mathbf{x}_1, ..., \mathbf{x}_n\} \geq \Omega\left(\log n\right)$. Then for any $\kappa \in (0, 1)$ we have*

$$\lambda_{\min}\left(\mathbf{G}^{(0)}\right) \geq \Omega\left(\frac{\bar{c}_{p-1}^2\left(\tanh'\right)}{n}\left(\frac{\kappa\sigma}{np}\right)^{2p}\right)$$

*with probability at least $1 - \kappa$ w.r.t. the noise matrix $\mathbf{N}$ and $\{\mathbf{w}_k^{(0)}\}_{k=1}^m$, provided*

$$m \geq \Omega\left(\frac{p^{4p}n^{4p+4}\log\left(n/\kappa\right)\log^2 m}{\sigma^{4p}\kappa^{4p}}\right),$$

*where $p$ denotes the smallest odd integer satisfying*

$$\binom{d'}{p} \geq n,$$

*and $\bar{c}_p\left(\tanh'\right)$ denotes the $p$-th coefficient in the probabilists' Hermite expansion of $\tanh'$.*

*Proof.* $p$ is chosen such that Equation 31 is satisfied. We use Lemma I.1, Lemma I.2 to find minimum eigenvalue of $\mathbf{G}^{\infty}$ and then applying Hoeffding's inequality (Fact C.3) to bound the deviation of minimum eigenvalue of $\mathbf{G}^{(0)}$ from $\mathbf{G}^{\infty}$, we get the desired bound. $\square$

Now, we specify the behavior of the probabilists' hermite expansion coefficients $c_{p-1}\left(\phi'\right)$ for $\phi = \tanh$. Let $\beta$, the exponent of real axis convergence of $\tanh'$, be the least upper bound on $\gamma$ for which

$$\tanh'(x) = \mathcal{O}\left(e^{-\nu|x|^{\gamma}}\right), \quad x \in \mathbb{R},$$

for some constant $\nu > 0$ as $|x| \to \infty$. We have $\beta = 1$, as $\tanh'(x) \sim e^{-4|x|}$ for large $|x|$.

Hence, using Eq. 5.15 in Boyd (1984) for the coefficients $\bar{c}_k$ in the probabilists' Hermite expansion of $\tanh'$ we have

$$\bar{c}_k = \frac{2}{(2k+1)^{1/4}}\Theta\left(e^{-\frac{\pi}{4}(2k+1)^{\frac{1}{2}}}\right), \quad \text{as } k \to \infty. \tag{34}$$

We remark that Boyd (1984) uses physicists' Hermite expansion. Following similar technique as in Corollary F.7.1 and Theorem F.7, we can get the exact similar form of probabilists' Hermite expansion coefficients $\bar{c}_k\left(\tanh'\right)$.

Thus for $p = \mathcal{O}(\log n)$, we have $\bar{c}_p = \tilde{\Omega}\left(e^{-c'\sqrt{\log n}}\right)$.

**Corollary I.4.1.** *Let $\phi(x)$ be the activation function $\tanh$ and $d' = \text{span}\{\mathbf{x}_1, ..., \mathbf{x}_n\} = \Theta\left(\log n\right)$. Then,*

$$\lambda_{\min}\left(\mathbf{G}^{(0)}\right) \geq \left(\frac{\sigma}{n}\right)^{\mathcal{O}(\log n)}$$

*with probability at least $1 - 1/\text{poly}(n) - e^{-\Omega\left(\frac{m}{\log^2 m}\left(\frac{\sigma}{n}\right)^{\mathcal{O}(\log n)}\right)}$ w.r.t. the noise matrix $\mathbf{N}$ and $\{\mathbf{w}_k^{(0)}\}_{k=1}^m$.*

**Corollary I.4.2.** *Let the activation be* tanh *and* $d' = \dim \text{span}\{\mathbf{x}_1, ..., \mathbf{x}_n\} = \Theta(n^\gamma)$, *for a constant* $\gamma \geq \Omega\left(\frac{\log \log n}{\log n}\right)$. *Then for any* $\kappa \in (0, 1)$ *we have*

$$\lambda_{\min}\left(\mathbf{G}^{(0)}\right) \geq \Omega\left(\frac{e^{-c'\sqrt{\log n}}}{n}\left(\frac{\kappa\sigma}{np}\right)^{2p}\right)$$

*with probability at least* $1 - \kappa$ *w.r.t. the noise matrix* $\mathbf{N}$ *and* $\{\mathbf{w}_k^{(0)}\}_{k=1}^m$, *provided*

$$m \geq \Omega\left(\frac{p^{4p}n^{4p+4}\log\left(n/\kappa\right)\log^2 m}{\sigma^{4p}\kappa^{4p}}\right),$$

*where* $p$ *is the smallest "odd" integer satisfying*

$$\binom{d'}{p} \geq n,$$

*and* $c'$ *is a constant.* $p$ *can be shown to lie in the range,*

$$\frac{1}{\gamma} \leq p \leq \frac{2}{\gamma - \frac{2}{\log n}}.$$

## J    DEPTH HELPS FOR tanh

Let the neural network under consideration be

$$F\left(\mathbf{x}; \mathbf{a}, \left\{\mathbf{W}^{(l)}\right\}_{l=1}^{L}\right) = \frac{c_\phi}{\sqrt{m}} \sum_{k=1}^{m} a_k \phi\left(\left(\mathbf{w}_k^{(L)}\right)^T \mathbf{x}^{(L-1)}\right)$$

where $\mathbf{x}^{(l)} \in \mathbb{R}^m \quad \forall l \geq 1$ and $\mathbf{x}^{(l)} \in \mathbb{R}^d$ for $l = 0$, is defined recursively by its components as follows.

$$x_k^{(l)} = \frac{c_\phi}{\sqrt{m}} \phi\left(\left(\mathbf{w}_k^{(l)}\right)^T \mathbf{x}^{(l-1)}\right) \quad \forall k \in [m], \forall l \geq 1$$

$$x_k^{(0)} = x_k \quad \forall k \in [d].$$

$c_\phi = \left(\mathbb{E}_{z \sim \mathcal{N}(0,1)} \phi(z)^2\right)^{-\frac{1}{2}}$, with $\phi$ following the following three properties.

- $\phi(0) = 0$.
- $\phi$ is $\alpha$-Lipschitz.
- $\mathbb{E}_{z \sim \mathcal{N}(0,1)} \phi(z) = 0$

The weight matrices and the output weight vector are given by $\left\{\mathbf{W}^{(l)}\right\}_{i=1}^{L}$ and $\mathbf{a}$ respectively, where $\mathbf{a} \in \mathbb{R}^m$, $\mathbf{W}^{(l)} \in \mathbb{R}^{m \times m}$ for $l \geq 2$ and $\mathbf{W}^{(l)} \in \mathbb{R}^{m \times d}$ for $l = 1$.

Now, we define the Gram matrix $\mathbf{G}^{(0)}$ as follows (cf. Eq. 13 in Du et al. (2019b)).

$$g_{ij}^{(0)} = \frac{1}{m} \sum_{k \in [m]} a_k^2 \phi'\left(\left(\mathbf{w}_k^{(L)}\right)^T \mathbf{x}_i^{(L-1)}\right) \phi'\left(\left(\mathbf{w}_k^{(L)}\right)^T \mathbf{x}_j^{(L-1)}\right)$$

with its counterpart $\mathbf{G}^\infty$, when $m \to \infty$, given by

$$g_{ij}^\infty = \mathbb{E}_{\mathbf{w} \sim \mathcal{N}(0,\mathbf{I}), a \sim \mathcal{N}(0,1)} a^2 \phi'\left(\mathbf{w}^T \mathbf{x}_i^{(L-1)}\right) \phi'\left(\mathbf{w}^T \mathbf{x}_j^{(L-1)}\right)$$

**Lemma J.1.** *For a small constant $\epsilon > 0$, if $m \geq \Omega\left(\max\left(\frac{2^L \log^2 m}{\epsilon^2} \log nL, (nL)^{2/7}\right)\right)$, then with probability at least $1 - e^{-\Omega\left(m\epsilon^2/2^L \log^2 m\right)} - \frac{nL}{m^{3.5}}$ we have*

$$\left\|\mathbf{x}_i^{(l)}\right\| \in (1 - \epsilon, 1 + \epsilon) \quad \forall i \in [n], l \in \{0, \ldots, L - 1\}. \tag{35}$$

*Proof.* We will use induction on $l$ to show that for any given $i$ with appropraite probability we have

$$\left\|\mathbf{x}_i^{(l)}\right\| \in \left(1 - \left(4c_\phi^2 \alpha^2\right)^{l-L} \epsilon, 1 + \left(4c_\phi^2 \alpha^2\right)^{l-L} \epsilon\right) \quad \forall l \in [L].$$

We will apply union bound over the choice of $i$ to derive the result for all $i \in [n]$. The result holds true for $l = 0$ by Assumption 1. Let's assume that the result holds true for $l = t$. For a randomly picked vector $\mathbf{w} \sim \mathcal{N}(0, \mathbf{I})$, $\mathbf{w}^T \mathbf{x}_i^{(t)}$ follows a normal distribution with mean 0 and standard deviation $\left\|\mathbf{x}_i^{(t)}\right\| \in (1 - \epsilon_t, 1 + \epsilon_t)$, where $\epsilon_t = \left(4c_\phi^2 \alpha^2\right)^{t-L} \epsilon$. Denote unit normalized form of $\mathbf{x}_i^{(t)}$ as $\overline{\mathbf{x}}_i^{(t)}$. Since, there are $m$ random Gaussian vectors in the matrix $\mathbf{W}^{(t+1)}$, leading to formation of $m$ Gaussians along the dimension of $\mathbf{W}^{(t+1)T} \mathbf{x}_i^{(t)}$, we can apply Fact C.5 to confine each dimension of $\mathbf{W}^{(t+1)T} \mathbf{x}_i^{(t)}$ to the range $\left(-3\sqrt{\log m}\left\|\mathbf{x}_i^{(t)}\right\|, 3\sqrt{\log m}\left\|\mathbf{x}_i^{(t)}\right\|\right)$ with probability at least $1 - \frac{1}{m^{3.5}}$. Assuming that this holds true, we can claim the following: First,

$$\Pr_{\mathbf{W}^{(t+1)}}\left(\left|\frac{1}{m}\sum_{k=1}^{m} c_\phi^2 \phi\left(\mathbf{w}_k^{(t+1)T} \mathbf{x}_i^{(t)}\right)^2 - E_{\mathbf{w} \sim \mathcal{N}(0,\mathbf{I})} c_\phi^2 \phi\left(\mathbf{w}^T \mathbf{x}_i^{(t)}\right)^2\right| \geq \alpha^2 c_\phi^2 \epsilon_t\right) \leq 2e^{-\Omega\left(\frac{m\epsilon_t^2}{\log^2 m}\right)}$$

$$\tag{36}$$

where we use Hoeffding's inequality (Fact C.3) and bound on $\mathbf{w}_k^{(t)T}\mathbf{x}_i^{(t)}$ and the $\alpha$-Lipschitzness of the activation $\phi$ to put a bound on $\left|\phi(\mathbf{w}_k^{(t+1)T}\mathbf{x}_i^{(t)})\right|$ to be used in Hoeffding's inequality.

Second, using Taylor expansion of $\phi$, the deviation of $E_{\mathbf{w}\sim\mathcal{N}(0,\mathbf{I})}c_\phi^2\phi(\mathbf{w}^T\mathbf{x}_i^{(t)})^2$ from $\mathbb{E}_{z\sim\mathcal{N}(0,1)}c_\phi^2\phi(z)^2$ can be bounded in the following manner.

$$\mathbb{E}_{\mathbf{w}\sim\mathcal{N}(0,\mathbf{I})}\,c_\phi^2\,\phi(\mathbf{w}^T\mathbf{x}_i^{(t)})^2 = \mathbb{E}_{\mathbf{w}\sim\mathcal{N}(0,\mathbf{I})}\,c_\phi^2\,\phi\left(\mathbf{w}^T\overline{\mathbf{x}}_i^{(t)}\right)^2 + \epsilon' = \mathbb{E}_{z\sim\mathcal{N}(0,1)}\,c_\phi^2\,\phi(z)^2 + \epsilon' \quad (37)$$

where $\epsilon' \in \left(-2c_\phi^2\alpha^2\epsilon_t, 2c_\phi^2\alpha^2\epsilon_t\right)$. This follows from the following set of equations.

$$\phi(\mathbf{w}^T\mathbf{x}_i^{(t)})^2 = \phi\left(\mathbf{w}^T\overline{\mathbf{x}}_i^{(t)}\right)^2$$
$$+ 2\int_{s=0}^1 \phi\left((1-s)\mathbf{w}^T\overline{\mathbf{x}}_i^{(t)} + s\mathbf{w}^T\mathbf{x}_i^{(t)}\right)\phi'\left((1-s)\mathbf{w}^T\overline{\mathbf{x}}_i^{(t)} + s\mathbf{w}^T\mathbf{x}_i^{(t)}\right)\left(\mathbf{w}^T\mathbf{x}_i^{(t)} - \mathbf{w}^T\overline{\mathbf{x}}_i^{(t)}\right)ds$$
$$\leq \phi\left(\mathbf{w}^T\overline{\mathbf{x}}_i^{(t)}\right)^2 + 2\epsilon_t(1+\epsilon_t)\alpha^2\left(\mathbf{w}^T\mathbf{x}_i^{(t)}\right)^2. \tag{38}$$

In the inequality above we used the facts that $\phi$ is $\alpha$-Lipschitz, and since $\phi(0) = 0$ by assumption, $\phi(z) \leq \alpha|z|$.

This gives

$$\mathbb{E}_{\mathbf{w}\sim\mathcal{N}(0,\mathbf{I})}\,c_\phi^2\,\phi(\mathbf{w}^T\mathbf{x}_i^{(t)})^2 \leq \mathbb{E}_{\mathbf{w}\sim\mathcal{N}(0,\mathbf{I})}\,c_\phi^2\,\phi\left(\mathbf{w}^T\overline{\mathbf{x}}_i^{(t)}\right)^2 + 2\epsilon_t(1+\epsilon_t)\alpha^2 c_\phi^2\,\mathbb{E}_{\mathbf{w}\sim\mathcal{N}(0,\mathbf{I})}\left(\mathbf{w}^T\mathbf{x}_i^{(t)}\right)^2$$
$$\leq \mathbb{E}_{\mathbf{w}\sim\mathcal{N}(0,\mathbf{I})}\,c_\phi^2\phi\left(\mathbf{w}^T\overline{\mathbf{x}}_i^{(t)}\right)^2 + 2\epsilon_t(1+\epsilon_t)\alpha^2 c_\phi^2.$$

Third, we use the definition of $c_\phi$ and the $\alpha$-Lipschitzness of $\phi$, to bound the error due to restricting the maximum magnitude of $\mathbf{w}^T\mathbf{x}_i^{(t)}$ to $3\sqrt{\log m}\left\|\mathbf{x}_i^{(t)}\right\|$, to get

$$\mathbb{E}_{\mathbf{w}\sim\mathcal{N}(0,\mathbf{I}):\left|\mathbf{w}^T\overline{\mathbf{x}}_i^{(t)}\right|\leq 3\sqrt{\log m}}c_\phi^2\phi\left(\mathbf{w}^T\overline{\mathbf{x}}_i^{(t)}\right)^2 = E_{\mathbf{w}\sim\mathcal{N}(0,\mathbf{I})}c_\phi^2\phi\left(\mathbf{w}^T\overline{\mathbf{x}}_i^{(t)}\right)^2 + \mathcal{O}\left(\sqrt{\frac{\log m}{m}}\right)$$
$$= 1 + \mathcal{O}\left(\sqrt{\frac{\log m}{m}}\right) \tag{39}$$

This can be shown by the following equation.

$$\left|E_{\mathbf{w}\sim\mathcal{N}(0,\mathbf{I}):\left|\mathbf{w}^T\overline{\mathbf{x}}_i^{(t)}\right|\leq 3\sqrt{\log m}}c_\phi^2\,\phi\left(\mathbf{w}^T\overline{\mathbf{x}}_i^{(t)}\right)^2 - E_{\mathbf{w}\sim\mathcal{N}(0,\mathbf{I})}c_\phi^2\,\phi\left(\mathbf{w}^T\overline{\mathbf{x}}_i^{(t)}\right)^2\right| \leq 2\frac{1}{\sqrt{2\pi}}\int_{3\sqrt{\log m}}^\infty \phi(x)^2 e^{-x^2/2}dx$$
$$\leq \alpha^2 2\frac{1}{\sqrt{2\pi}}\int_{3\sqrt{\log m}}^\infty x^2 e^{-x^2/2}dx$$
$$\leq \alpha^2\sqrt{\frac{\log m}{m}} \quad (40)$$

Combining Equation 36, Equation 37 and Equation 39, we get that with probability at least $1 - 2e^{-\Omega\left(\frac{m\epsilon_t^2}{\log^2 m}\right)} - m^{-7/2}$,

$$\left\|\mathbf{x}_i^{(t+1)}\right\| = \left(\frac{1}{m}\sum_{k=1}^m c_\phi^2\phi(\mathbf{w}_k^{(t)T}\mathbf{x}_i)^2\right)^{1/2} \in \left(1 - 4\alpha^2 c_\phi^2\epsilon_t, 1 + 4\alpha^2 c_\phi^2\epsilon_t\right).$$

We use the union bound for all the induction steps and examples to get the final desired bounds. $\quad\square$

**Lemma J.2.** *If for a pair $i, j \in [n]$, $\overline{\mathbf{x}}_i^{(l-1)T}\overline{\mathbf{x}}_j^{(l-1)} = \rho$ s.t. $|\rho| \leq 1 - \delta$ and if Equation 35 holds, then for all small $\epsilon > 0$,*

$$\left|\overline{\mathbf{x}}_i^{(l)T}\overline{\mathbf{x}}_j^{(l)}\right| \leq 3\left(1 + 2\epsilon\right)\epsilon\alpha^2 c_\phi^2 + (1 + 2\epsilon)\max\left(\frac{1+c}{2}\left(\frac{\delta}{n^2}\right) + \frac{1-c}{2}\left(\frac{\delta}{n^2}\right)^2, \frac{1+c}{2}|\rho| + \frac{1-c}{2}|\rho|^2\right)$$

*with probability at least $1 - m^{-7/2} - e^{-\Omega\left(m\delta^2/\alpha^4 n^4 \log^2 m\right)}$ w.r.t. initialization, where $c$ denotes the ratio $\frac{\bar{c}_1^2(\phi)}{\sum_{a=1}^\infty \bar{c}_a^2(\phi)}$.*

*Proof.* From Equation 35, for a small constant $\epsilon$, $\left\|\mathbf{x}_i^{(l-1)}\right\| \in (1 - \epsilon, 1 + \epsilon), \forall i \in [n]$ with high probability. For a vector $\mathbf{x}$ set $\overline{\mathbf{x}} := \mathbf{x}/\|\mathbf{x}\|$; thus we will use $\overline{\mathbf{x}}_i^{(l-1)}$ for $\mathbf{x}_i^{(l-1)}/\left\|\mathbf{x}_i^{(l-1)}\right\|$. We use Fact C.5 to restrict the maximum magnitude of the $2m$ Gaussians $\mathbf{w}_k^T\mathbf{x}_i^{(l-1)}$ and $\mathbf{w}_k^T\mathbf{x}_j^{(l-1)}$ for $k \in [m]$ to $3\sqrt{\log m}\left\|\mathbf{x}_i^{(l-1)}\right\|$ and $3\sqrt{\log m}\left\|\mathbf{x}_j^{(l-1)}\right\|$ respectively. Assuming that this condition holds, we have

$$\mathbf{x}_i^{(l)T}\mathbf{x}_j^{(l)} = c_\phi^2 \frac{1}{m} \sum_{k \in [m]} \phi\left(\mathbf{w}_k^{(l)T}\mathbf{x}_i^{(l-1)}\right)\phi\left(\mathbf{w}_k^{(l)T}\mathbf{x}_j^{(l-1)}\right)$$

$$= c_\phi^2 \frac{1}{m} \sum_{k \in [m]} \phi\left(\mathbf{w}_k^{(l)T}\overline{\mathbf{x}}_i^{(l-1)}\right)\phi\left(\mathbf{w}_k^{(l)T}\overline{\mathbf{x}}_j^{(l-1)}\right) + \epsilon' \tag{41}$$

$$= c_\phi^2 E_{\mathbf{w} \sim \mathcal{N}(0,\mathbf{I})}\phi\left(\mathbf{w}_k^{(l)T}\overline{\mathbf{x}}_i^{(l-1)}\right)\phi\left(\mathbf{w}_k^{(l)T}\overline{\mathbf{x}}_j^{(l-1)}\right) + \epsilon' + \epsilon'' \tag{42}$$

$$= c_\phi^2 \sum_{a=0}^\infty \bar{c}_a^2(\phi)\rho^a + \epsilon' + \epsilon'' + \epsilon'''. \tag{43}$$

We get Equation 41 along the lines of Equation 38: we use 1-st order Taylor expansion of $\phi\left(\mathbf{w}_k^T\mathbf{x}_i^{(l-1)}\right)$ around $\mathbf{w}_k^T\overline{\mathbf{x}}_i^{(l-1)}$ and $\phi\left(\mathbf{w}_k^T\mathbf{x}_j^{(l-1)}\right)$ around $\mathbf{w}_k^T\overline{\mathbf{x}}_j^{(l-1)}$, $\alpha$-Lipschitzness of $\phi$ and upper and lower bounds of $\left\|\mathbf{x}_i^{(l-1)}\right\|$ and $\left\|\mathbf{x}_j^{(l-1)}\right\|$ from Lemma J.1 to get

$$\phi(\mathbf{w}_k^T\mathbf{x}_i^{(l-1)})\phi(\mathbf{w}_k^T\mathbf{x}_j^{(l-1)}) = \phi\left(\mathbf{w}_k^T\overline{\mathbf{x}}_i^{(l-1)}\right)\phi(\mathbf{w}_k^T\overline{\mathbf{x}}_j^{(l-1)})$$

$$+ \int_{s=0}^1 \phi(\mathbf{w}_k^T\overline{\mathbf{x}}_j^{(l-1)})\phi'\left((1-s)\mathbf{w}_k^T\overline{\mathbf{x}}_i^{(l-1)} + s\mathbf{w}_k^T\mathbf{x}_i^{(l-1)}\right)\left(\mathbf{w}_k^T\mathbf{x}_i^{(l-1)} - \mathbf{w}_k^T\overline{\mathbf{x}}_i^{(l-1)}\right)ds$$

$$+ \int_{t=0}^1 \phi(\mathbf{w}_k^T\overline{\mathbf{x}}_i^{(l-1)})\phi'\left((1-t)\mathbf{w}_k^T\overline{\mathbf{x}}_j^{(l-1)} + t\mathbf{w}_k^T\mathbf{x}_j^{(l-1)}\right)\left(\mathbf{w}_k^T\mathbf{x}_j^{(l-1)} - \mathbf{w}_k^T\overline{\mathbf{x}}_j^{(l-1)}\right)dt$$

$$+ \int_{s=0}^1\int_{t=0}^1 \phi'\left((1-s)\mathbf{w}_k^T\overline{\mathbf{x}}_i^{(l-1)} + s\mathbf{w}_k^T\mathbf{x}_i^{(l-1)}\right)\left(\mathbf{w}_k^T\mathbf{x}_i^{(l-1)} - \mathbf{w}_k^T\overline{\mathbf{x}}_i^{(l-1)}\right)$$

$$\phi'\left((1-t)\mathbf{w}_k^T\overline{\mathbf{x}}_j^{(l-1)} + t\mathbf{w}_k^T\mathbf{x}_j^{(l-1)}\right)\left(\mathbf{w}_k^T\mathbf{x}_j^{(l-1)} - \mathbf{w}_k^T\overline{\mathbf{x}}_j^{(l-1)}\right)ds dt$$

$$\leq \phi\left(\mathbf{w}_k^T\overline{\mathbf{x}}_i^{(l-1)}\right)\phi\left(\mathbf{w}_k^T\overline{\mathbf{x}}_j^{(l-1)}\right) + \alpha^2\left(2\epsilon(1+\epsilon) + \epsilon^2(1+\epsilon)^2\right)\left|\mathbf{w}_k^T\overline{\mathbf{x}}_i^{(l-1)}\right|\left|\mathbf{w}_k^T\overline{\mathbf{x}}_j^{(l-1)}\right|. \tag{44}$$

In the inequality above we used the facts that $\phi$ is $\alpha$-Lipschitz, and since $\phi(0) = 0$ by assumption, $\phi(z) \leq \alpha|z|$. This gives

$$\frac{1}{m}\sum_{k=1}^m c_\phi^2\phi(\mathbf{w}_k^T\mathbf{x}_i^{(l-1)})\phi(\mathbf{w}_k^T\mathbf{x}_j^{(l-1)}) \leq \frac{1}{m}\sum_{k=1}^m c_\phi^2\phi(\mathbf{w}_k^T\overline{\mathbf{x}}_i^{(l-1)})\phi(\mathbf{w}_k^T\overline{\mathbf{x}}_i^{(l-1)})$$

$$+ \alpha^2\left(2\epsilon(1+\epsilon) + \epsilon^2(1+\epsilon)^2\right)\frac{1}{m}\sum_{k=1}^m c_\phi^2\left|\mathbf{w}_k^T\overline{\mathbf{x}}_i^{(l-1)}\right|\left|\mathbf{w}_k^T\overline{\mathbf{x}}_j^{(l-1)}\right|$$

$$\leq \frac{1}{m}\sum_{k=1}^m c_\phi^2\phi(\mathbf{w}_k^T\overline{\mathbf{x}}_i^{(l-1)})\phi(\mathbf{w}_k^T\overline{\mathbf{x}}_i^{(l-1)}) + 4\alpha^2 c_\phi^2\left(2\epsilon(1+\epsilon) + \epsilon^2(1+\epsilon)^2\right).$$

In the last step, we use Hoeffding's inequality to bound the deviation of $\frac{1}{m}\sum_{k=1}^{m} c_\phi^2 \left|\mathbf{w}_k^T \overline{\mathbf{x}}_i^{(l-1)}\right| \left|\mathbf{w}_k^T \overline{\mathbf{x}}_j^{(l-1)}\right|$ from $\mathbb{E}_{\mathbf{w}\sim\mathcal{N}(0,\mathbf{I})} c_\phi^2 \left|\mathbf{w}^T \overline{\mathbf{x}}_i^{(l-1)}\right| \left|\mathbf{w}^T \overline{\mathbf{x}}_j^{(l-1)}\right|$ and using standard gaussian moments, we can show that that $\mathbb{E}_{\mathbf{w}\sim\mathcal{N}(0,\mathbf{I})} \left|\mathbf{w}^T \overline{\mathbf{x}}_i^{(l-1)}\right| \left|\mathbf{w}^T \overline{\mathbf{x}}_j^{(l-1)}\right|$ is at-most 4. Thus, combining everything,we get

$$\epsilon' \in \left(-20\epsilon\alpha^2 c_\phi^2, 20\epsilon\alpha^2 c_\phi^2\right)$$

with probability at least $1 - e^{-\Omega\left(m/\log^2 m\right)}$. In Equation 42, we use Hoeffding's inequality (Fact C.3), $\alpha$-Lipschitzness of $\phi$ and bound for the magnitude of the $2m$ gaussians $\mathbf{w}_k^T \overline{\mathbf{x}}_i^{(l-1)}$ and $\mathbf{w}_k^T \overline{\mathbf{x}}_j^{(l-1)}$ to get $\epsilon'' \in (-\tau, \tau)$ where

$$\tau = \begin{cases} \frac{1-c}{2}|\rho|\left(1-|\rho|\right), & \text{if}|\rho| \geq \frac{\delta}{n^2}. \\ \frac{1-c}{2}\frac{\delta}{n^2}\left(1-\frac{\delta}{n^2}\right), & \text{otherwise.} \end{cases}$$

with probability at least $1 - e^{-\left(\Omega\left(m\delta^2/\alpha^4 n^4 \log^2 m\right)\right)}$. Note that, the minimum value of $\tau$ is $\frac{1-c}{2}\frac{\delta}{n^2}\left(1-\frac{\delta}{n^2}\right)$. The reason of using this form of $\tau$ will be discussed below. We use Lemma N.4 in Equation 43, with $\rho_{00} = \left\|\overline{\mathbf{x}}_i^{(l-1)}\right\|^2$, $\rho_{11} = \left\|\overline{\mathbf{x}}_j^{(l-1)}\right\|^2$ and $\rho_{01} = \rho\left\|\overline{\mathbf{x}}_i^{(l-1)}\right\|\left\|\overline{\mathbf{x}}_j^{(l-1)}\right\|$. This follows from the fact that for a random normal vector $\mathbf{w} \sim \mathcal{N}(0,\mathbf{I})$, $\mathbf{w}^T\mathbf{x}$ follows a normal distribution with mean 0 and variance $\|\mathbf{x}\|^2$ and the covariance of $\mathbf{w}^T\mathbf{x}$ and $\mathbf{w}^T\mathbf{y}$ is $\mathbf{x}^T\mathbf{y}$ for two vectors $\mathbf{x}$ and $\mathbf{y}$. We have an additional error term $\epsilon''' \in \left(-\sqrt{\frac{\log m}{m}}, \sqrt{\frac{\log m}{m}}\right)$ owing to the condition that $\left|\mathbf{w}^T \overline{\mathbf{x}}_i^{(l-1)}\right|$ must be at most $3\sqrt{\log m}$ (proof will follow exactly along the lines of Equation 40).

Let us now focus on the quantity $R(\rho)$ for the activation function $c_\phi\phi(.)$, defined in Fact N.2. From the definition of $c_\phi$, it follows that $c_\phi^2 = \frac{1}{\sum_{a=1}^{\infty} \bar{c}_a^2(\phi)}$. Thus, we have $\left|R(\rho)\right| \leq R(|\rho|) \leq R(1) = 1$, which comes from the properties of $R$, as given in Fact N.2. Again, we have

$$\left|R(\rho)\right| \leq R(|\rho|) \leq \left(c_\phi^2 \sum_{a=1}^{\infty} \bar{c}_a^2(\phi)|\rho|^2 + c_\phi^2 c_1^2(\phi)\left(|\rho| - |\rho|^2\right)\right)$$

$$= \left(|\rho| + \frac{\bar{c}_1^2(\phi)}{\sum_{a=1}^{\infty} \bar{c}_a^2(\phi)}\left(1-|\rho|\right)\right)|\rho|$$

Thus,

$$\left|\mathbf{x}_i^{(l)T}\mathbf{x}_j^{(l)}\right| \leq |\epsilon''| + |\epsilon'| + |\epsilon'''| + |R(\rho)| \leq 6\epsilon\alpha^2 c_\phi^2 + \max\left(\frac{1+c}{2}\left(\frac{\delta}{n^2}\right) + \frac{1-c}{2}\left(\frac{\delta}{n^2}\right)^2, \frac{1+c}{2}|\rho| + \frac{1-c}{2}|\rho|^2\right)$$

(45)

where $c$ denotes the ratio $\frac{\bar{c}_1^2(\phi)}{\sum_{a=1}^{\infty} \bar{c}_a^2(\phi)}$. In the equation above, we can see that the form of $\tau = \max|\epsilon'|$ has been chosen so that $R(|\rho|) + |\epsilon'| \leq \frac{1}{2}\left(R(|\rho|) + |\rho|\right)$ Thus,

$$\left|\overline{\mathbf{x}}_i^{(l)T}\overline{\mathbf{x}}_j^{(l)}\right| \leq 6(1+2\epsilon)\epsilon\alpha^2 c_\phi^2 + (1+2\epsilon)\max\left(\frac{1+c}{2}\left(\frac{\delta}{n^2}\right) + \frac{1-c}{2}\left(\frac{\delta}{n^2}\right)^2, \frac{1+c}{2}|\rho| + \frac{1-c}{2}|\rho|^2\right)$$

(46)

where we use the bound on $\left\|\mathbf{x}_i^{(l)}\right\|$ from Equation 35. $\qquad\square$

**Lemma J.3.** $\forall i,j \in [n], i \neq j$, if $\overline{\mathbf{x}}_i^T\overline{\mathbf{x}}_j \leq 1 - \delta$, then $\overline{\mathbf{x}}_i^{(L-1)T}\overline{\mathbf{x}}_j^{(L-1)} \leq \epsilon$, where

$$\Omega\left(\frac{\delta}{n^2(1-c)}\right) \leq \epsilon \leq 1 - \Omega\left(\frac{\delta}{n^2(1-c)}\right),$$

*and*

$$L \geq \frac{2}{1-c} \max\left(\Omega\left(\log\frac{1}{\delta}\right), \Omega\left(\log\frac{1}{\epsilon}\right)\right),$$

*with probability at least* $1 - \Omega\left(\frac{1}{m^3}\right)$, *provided* $m \geq \Omega\left(\frac{2^L n^2 \alpha^4 c_\phi^4 \log(n^2 L) \log^2 m}{\delta^2}\right)$, *, where c denotes the ratio* $\frac{\bar{c}_1^2(\phi)}{\sum_{a=1}^\infty \bar{c}_a^2(\phi)}$.

*Proof.* Applying Lemma J.2 with $\epsilon = \frac{\delta}{20n^2\alpha^2 c_\phi^2}$, we have for each layer $l \in [L]$ and $i, j \in [n]; i \neq j$,

$$\left|\overline{\mathbf{x}}_i^{(l)T}\overline{\mathbf{x}}_j^{(l)}\right| \leq f\left(\left|\overline{\mathbf{x}}_i^{(l-1)T}\overline{\mathbf{x}}_j^{(l-1)}\right|\right)$$

holds with probability $1 - \Omega\left(\frac{1}{m^3}\right)$, provided $m \geq \Omega\left(\frac{2^L n^4 \alpha^4 c_\phi^4 \log(n^2 L) \log^2 m}{\delta^2}\right)$. Here function $f : \mathbb{R} \to \mathbb{R}$ is s.t. for $\rho \in \mathbb{R}$,

$$f(\rho) = \begin{cases} \hat{f}(\rho), & \text{if } |\rho| \geq \frac{\delta}{n^2}. \\ \frac{1+c}{2}\left(\frac{\delta}{n^2}\right) + \frac{1-c}{2}\left(\frac{\delta}{n^2}\right)^2 + \frac{\delta}{n^2}, & \text{otherwise.} \end{cases}$$

where function $\hat{f}$ is defined as

$$\hat{f}(\rho) = \frac{1+c}{2}\rho + \frac{1-c}{2}\rho^2 + \frac{\delta}{n^2}$$

Thus,

$$\left|\overline{\mathbf{x}}_i^{(L)T}\overline{\mathbf{x}}_j^{(L)}\right| \leq \underbrace{f \circ f \dots \circ f}_{L \text{ times}}\left(\left|\overline{\mathbf{x}}_i^{(0)T}\overline{\mathbf{x}}_j^{(0)}\right|\right)$$

Let's now focus on the function $f$. It can be seen that for the function $\hat{f}$, $\frac{2\delta}{n^2(1-c)}$ and $1 - \frac{2\delta}{n^2(1-c)}$ are two fixed points, and starting from any positive $\rho^{(0)}$ strictly less than $1 - \frac{2\delta}{n^2(1-c)}$ and following fixed point algorithm leads to convergence to the point $\frac{2\delta}{n^2(1-c)}$. Since, $\hat{f}$ equals the function $f$ till the convergence point, the rate of convergence of fixed point algorithm for the function $f$ is well approximated by the rate of convergence for the function $\hat{f}$. Also, the rate of convergence of $\hat{f}$ is equal to the rate of convergence of the function $\tilde{f} : \mathbb{R} \to \mathbb{R}$ defined for each $\rho \in \mathbb{R}$ as

$$\tilde{f}(\rho) = \frac{1+c}{2}\rho + \frac{1-c}{2}\rho^2.$$

Lemma J.4 shows that starting at $\rho^{(0)} = 1 - \delta$, the number of fixed point iteration steps to reach $\epsilon$ for function $\tilde{f}$ is given by $\frac{2}{1-c}\max\left(\Omega\log\left(\frac{1}{\delta}\right), \Omega\log\left(\frac{1}{\epsilon}\right)\right)$. Hence, from this argument, if $\left|\overline{\mathbf{x}}_i^{(0)T}\overline{\mathbf{x}}_j^{(0)}\right| \leq 1 - \delta$ and $L \geq \frac{2}{1-c}\max\left(\Omega\left(\log\frac{1}{\delta}\right), \Omega\left(\log\frac{1}{\epsilon}\right)\right)$, the quantity $\left|\overline{\mathbf{x}}_i^{(L)T}\overline{\mathbf{x}}_j^{(L)}\right|$ becomes less than $\epsilon$. $\qquad\square$

**Lemma J.4.** *For $\rho \in \mathbb{R}$, define the function $\tilde{f} : \mathbb{R} \to \mathbb{R}$ by*

$$\tilde{f}(\rho) = a\rho + (1-a)\rho^2$$

*where $a$ is a constant in $(0, \frac{1}{2})$. Starting at $\rho^{(0)} = 1 - \delta$, the number of fixed point iteration steps to reach $\epsilon$ is given by $\frac{1}{1-a}\max\left(\Omega\left(\log\frac{1}{\delta}\right), \Omega\left(\log\frac{1}{\epsilon}\right)\right)$.*

*Proof.* The function $\tilde{f}$ has fixed points 0 and 1, but it's easy to see that starting at any point below 1, fixed point iteration converges to 0; we want to understand the speed of convergence. We will divide the fixed point iterate's path into two sub-paths (a) movement from $1 - \delta$ to $1 - b$ ($b$ is a small constant) and (b) movement from $1 - b$ to $\epsilon$.

- **Movement from** $1 - \delta$ **to** $1 - b$ **:**. We will divide the path into subpaths $(1 - 2^t\delta, 1 - 2^{t-1}\delta)$, where $t$ is an integer in $(0, \log \frac{b}{\delta})$. We show that $\forall t \leq \mathcal{O}\left(\log \frac{b}{\delta}\right)$, the number of iterations to reach from $1 - 2^{t-1}\delta$ to $1 - 2^t\delta$ can be upper bounded by $\frac{1}{1-a}$. The number of iterations of function $\tilde{f}$ to go from $(1 - 2^{t-1}\delta, 1 - 2^t\delta)$ is at most the number of iterations of the linear function $\hat{f} : \mathbb{R} \to \mathbb{R}$ defined by

$$\hat{f}(\rho) = \left(1 - (1-a)2^{t-1}\delta\right)\rho.$$

  The number of fixed point iterations of $\hat{f}$ to go from $\left(1 - 2^{t-1}\delta\right)$ to $\left(1 - 2^t\delta\right)$ is given by $\frac{\log \frac{1 - 2^{t-1}\delta}{1 - 2^t\delta}}{\log\left(1 - (1-a)2^{t-1}\delta\right)}$, which can be shown to be less than $2\frac{2^t\delta - 2^{t-1}\delta}{(1-a)2^{t-1}\delta} = \frac{2}{1-a}$ using Taylor expansion of log function. Thus, the total number of iterations involved in the entire path is upper bounded by $\frac{2}{1-a} \log \frac{b}{\delta}$.

- **Movement from** $1 - b$ **to** $\epsilon$ **:** The number of iterations of $\tilde{f}$ is t most that of a linear function $\hat{f}$ in the domain $(0, 1 - b)$ defined by

$$\hat{f}(\rho) = \left(1 - (1-a)b\right)\rho.$$

  The number of iterations of $\hat{f}$ to go from $(1 - b)$ to $\epsilon$ is given by $\frac{\log \frac{1-b}{\epsilon}}{\log \frac{1}{1-b(1-a)}}$, which upper bounded by $\frac{1}{b(1-a)} \log \frac{1-b}{\epsilon}$, for small enough constant $b$.

Thus, summing the number of steps needed in the two subpaths leads to the desired quantity. $\qquad \square$

**Theorem J.5.** *If*

$$L \geq \frac{2}{1-c} \max\left(\Omega\left(\log \frac{1}{\delta}\right), \Omega\left(\frac{\log 2n}{\log n}\right)\right),$$

*then*

$$\lambda_{\min}\left(\mathbf{G}^{(0)}\right) \geq \Omega\left(\bar{c}^2_{\Theta(\log n)}(\phi')\right) - \mathcal{O}\left(\frac{\delta}{n}\right)$$

*with probability at least* $1 - \Omega\left(\frac{1}{m^3}\right)$, *provided* $m \geq \Omega\left(\frac{2^L n^4 \alpha^4 c_\phi^4 \log(n^2 L) \log^2 m}{\delta^2}\right)$, *where* $\bar{c}_k(\phi')$ *denotes the* $k^{\text{th}}$ *order coefficient in the probabilists' Hermite expansion of* $\phi'$ *and* $c$ *denotes the ratio* $\frac{\bar{c}_1^2(\phi)}{\sum_{a=1}^{\infty} \bar{c}_a^2(\phi)}$.

*Proof.* Assuming Equation 35 with $\epsilon = \frac{\delta}{n^2 c_\phi^2 \alpha^2}$, using Lemma I.1 we get

$$\lambda_{\min}\left(\mathbf{G}^\infty\right) \geq \bar{c}^2_{\Theta(\log n)}(\phi') \lambda_{\min}\left(\left(\left(\overline{\mathbf{X}}^{(L)}\right)^{*\log n}\right)^T \left(\overline{\mathbf{X}}^{(L)}\right)^{*\log n}\right) + \epsilon' + \epsilon'' \qquad (47)$$

where $\overline{\mathbf{X}}^{(L)}$ denotes a $m \times n$ matrix, with its $i^{\text{th}}$ column containing $\overline{\mathbf{x}}_i^{(L)}$, which is the unit normalized form of $\mathbf{x}_i^{(L)}$, $\left(\overline{\mathbf{X}}^{(L)}\right)^{*r}$ denotes its order $r^{\text{th}}$ Khatri–Rao power. There are two error terms $\epsilon'$ and $\epsilon''$ because of two reasons (a) norm of $\mathbf{x}_i^{(L)}$ is $\epsilon$ away from 1 (b) magnitude of $\mathbf{w}^T \mathbf{x}_i^{(l)}$ is restricted to $3\sqrt{\log m}$. Following the line of proof of Equation 38, magnitude of $\epsilon'$ can be bounded to $\mathcal{O}\left(nc_\phi^2 \alpha^2 \epsilon\right)$. Also, following the line of proof of Equation 40, magnitude of $\epsilon''$ can be bounded to $n\sqrt{\frac{\log m}{m}}$. Now, we make the following claims.

**First**, $\mathbf{x}_i^T \mathbf{x}_j$, for any $i, j \in [n]$ with $i \neq j$, can be shown to be at most to $1 - \delta$, using Assumption 2.

**Second**, if $\overline{\mathbf{x}}_i^T \overline{\mathbf{x}}_j = \rho$ s.t. $|\rho| \leq 1 - \delta$, applying Lemma J.3 shows that $\left| \left( \overline{\mathbf{x}}_i^{(L)} \right)^T \overline{\mathbf{x}}_j^{(L)} \right| \leq \tilde{\epsilon}$, for a small constant $\tilde{\epsilon}$, provided $L \geq \frac{2}{1-c} \max \left( \Omega \left( \log \frac{1}{\delta} \right), \Omega \left( \log \frac{1}{\tilde{\epsilon}} \right) \right)$, with high probability.

**Third**, if for any $i, j \in [n]$, $\left( \overline{\mathbf{x}}_i^{(L)} \right)^T \overline{\mathbf{x}}_j^{(L)} = \rho' < 1$, then we can see that $\left( \left( \overline{\mathbf{x}}_i^{(L)} \right)^{*r} \right)^T \left( \overline{\mathbf{x}}_j^{(L)} \right)^{*r} = \rho'^r$, where $\mathbf{x}^{*r}$ denotes the order-$r$ Khatri–Rao product of a vector $\mathbf{x}$.

Combining these claims, we get the following.

First, The diagonal elements of the matrix $\left( \left( \overline{\mathbf{X}}^{(L)} \right)^{* \log n} \right)^T \left( \overline{\mathbf{X}}^{(L)} \right)^{* \log n}$ are equal to 1. Second,

the non diagonal element at row $i$ and column $j$ of the matrix $\left( \left( \overline{\mathbf{X}}^{(L)} \right)^{* \log n} \right)^T \left( \overline{\mathbf{X}}^{(L)} \right)^{* \log n}$ is

given by $\left( \left( \overline{\mathbf{x}}_i^{(L)} \right)^{* \log n} \right)^T \left( \overline{\mathbf{x}}_i^{(L)} \right)^{* \log n}$, whose magnitude is bounded by $\tilde{\epsilon}^{\log n}$. Hence, if

$$ L \geq \frac{2}{1-c} \max \left( \Omega \left( \log \frac{1}{\delta} \right), \Omega \left( \frac{\log 2n}{\log n} \right) \right), $$

each non diagonal element's absolute value becomes less than $\frac{1}{2n}$ and so the absolute sum of non diagonal elements is at least $\frac{1}{2}$ away from the absolute value of the diagonal element, for each row of the matrix $\left( \left( \mathbf{X}^{(L)} \right)^{* \log n} \right)^T \left( \mathbf{X}^{(L)} \right)^{* \log n}$. Using Fact C.10, we get for r $= \log n$,

$$ \lambda_{\min} \left( \mathbf{X}^{(L)*rT} \mathbf{X}^{(L)*r} \right) \geq \frac{1}{2} \tag{48} $$

Combining the value of $\epsilon'$, value of $\epsilon''$ and Equation 48 gives us the minimum eigenvalue of $\mathbf{G}^\infty$. Since,

$$ \lambda_{\min} \left( G^{(0)} \right) \geq \lambda_{\min} \left( G^\infty \right) - \left\| \left( G^\infty - G^{(0)} \right) \right\|_F, $$

we use Hoeffding's inequality to bound the magnitude of each element of $\left( G^\infty - G^{(0)} \right)$ to $\lambda_{\min} \left( G^\infty \right) / 2n$ and hence, get a bound on $\lambda_{\min} \left( G^{(0)} \right)$. $\square$

**Theorem J.6.** *If $\phi = \tanh$ and*

$$ L \geq \frac{2}{1-c} \max \left( \Omega \left( \log \frac{1}{\delta} \right), \left( \frac{\log 2n}{\log n} \right) \right), $$

*then*

$$ \lambda_{\min} \left( G^{(0)} \right) \geq e^{-\Omega \left( \log^{\frac{1}{2}} n \right)} \gg 1/\mathsf{poly}(n) $$

*with probability at least $1 - \Omega \left( \frac{1}{m^3} \right)$, provided $m \geq \Omega \left( \frac{2^L n^4 \log \left( n^2 L \right) \log^2 m}{\delta^2} \right)$, for an arbitrary constant $\epsilon > 0$. The constant $c$ denotes the ratio $\frac{\bar{c}_1^2 (\tanh)}{\sum_{a=1}^\infty \bar{c}_a^2 (\tanh)}$.*

*Proof.* The proof follows from Theorem J.5, with the bound on Hermite coefficients for $\tanh'$ from Eqn. (34). $\square$

We re-state the rate of convergence theorem from Du et al. (2019b) and use our bounds on minimum eigenvalue of the gram matrix to give a more finer version of the theorem.

**Theorem J.7** (Thm 5.1 in Du et al. (2019b)). *Assume that Assumption 1 and Assumption 2 hold true, $|y_i| = \mathcal{O}(1) \ \forall i \in [n]$ and the number of neurons per layer satisfy*

$$m \geq \Omega \left( 2^{O(L)} \max \left\{ \frac{n^4}{\lambda_{\min}^4 \left( \mathbf{G}^{(0)} \right)}, \frac{n}{\kappa}, \frac{n^2 \log \left( \frac{Ln}{\kappa} \right)}{\lambda_{\min}^2 \left( \mathbf{G}^{(0)} \right)} \right\} \right).$$

*Then, if we follow a Gradient Descent algorithm with step size*

$$\eta = O \left( \frac{\lambda_{\min} \left( \mathbf{G}^{(0)} \right)}{n^2 2^{O(L)}} \right),$$

*with probability at least $1 - \kappa$ over the random initialization, the following holds true $\forall t \geq 1$.*

$$\left\| \mathbf{y} - \mathbf{u}^{(t)} \right\|^2 \leq \left( 1 - \frac{\eta \lambda_{\min} \left( \mathbf{G}^{(0)} \right)}{2} \right)^t \left\| \mathbf{y} - \mathbf{u}^{(0)} \right\|^2.$$

Thus, refining the above theorem with our computed bounds for $\lambda_{\min} \left( \mathbf{G}^{(0)} \right) > \mathcal{O}(\frac{1}{n})$, we get the following.

**Theorem J.8.** *If $\phi = \tanh$,*

$$L \geq \frac{2}{1 - c} \max \left( \Omega \left( \log \frac{1}{\delta} \right), \left( \frac{\log 2n}{\log n} \right) \right),$$

*and the number of neurons per layer satisfy*

$$m \geq \Omega \left( 2^{O(L)} \max \left\{ n^8, \frac{n}{\kappa}, n^4 \log \left( \frac{Ln}{\kappa} \right) \right\} \right),$$

*then, if we follow a Gradient Descent algorithm with step size*

$$\eta \leq O \left( \frac{1}{n^3 2^{O(L)}} \right),$$

*with probability at least $1 - \kappa$ over the random initialization, the following holds true $\forall t \geq 1$.*

$$\left\| \mathbf{y} - \mathbf{u}^{(t)} \right\|^2 \leq \left( 1 - \frac{\eta}{2n} \right)^t \left\| \mathbf{y} - \mathbf{u}^{(0)} \right\|^2.$$

*The constant $c$ denotes the ratio $\frac{\bar{c}_1^2(\tanh)}{\sum_{a=1}^{\infty} \bar{c}_a^2(\tanh)}$.*

# K  LOWER BOUND ON LOWEST EIGENVALUE FOR NON-SMOOTH FUNCTIONS

For $\alpha \in (-1, 1)$ define the activation function $\phi$ by

$$\phi(x) = \phi_1(x)\mathcal{I}_{x<\alpha} + \phi_2(x)\mathcal{I}_{x\geq\alpha}.$$

We show that the minimum singular value of the $G$-matrix is at least inverse polynomially large in $n$ and $\delta$, provided $\phi_1$ and $\phi_2$ satisfy the following properties for some positive integer $r$. We denote this condition by $\mathbf{J_r}$.

- $\phi_1, \phi_2 \in C^{r+1}$ in the domains $(-\infty, \alpha]$ and $[\alpha, \infty)$, respectively.
- The first $(r+1)$ derivatives of $\phi_1$ and $\phi_2$ are upper bounded in magnitude by 1 in $(-\infty, \alpha]$ and $[\alpha, \infty)$ respectively.
- For $0 \leq i < r$, we have $\phi_1^{(i)}(\alpha) = \phi_2^{(i)}(\alpha)$.
- $\left|\phi_1^{(r)}(\alpha) - \phi_2^{(r)}(\alpha)\right| = 1$, i.e. the $r$-th derivative has a jump discontinuity at $\alpha$.

In the following we consider $\mathbf{J_1}$ and $\mathbf{J_2}$. The results can be easily generalized to higher $r$ but with lower bound degrading as $n^{-2^r}$.

## K.1  $\mathbf{J_1}$: THE FIRST DERIVATIVE IS DISCONTINUOUS AT A POINT

### K.1.1  DZPS SETTING

Recall that this setting was defined in section 2. ReLU, SELU and LReLU satisfy the conditions for the following theorem. The data set $\{(\mathbf{x}_i, y_i)\}_{i=1}^n$, for $\mathbf{x}_i \in \mathbb{R}^d$ and $y_i \in \mathbb{R}$ is implicitly understood in the theorem statements below.

**Theorem K.1.** *Let the condition on $\phi$ be satisfied for $r = 1$. Assume that $\mathbf{w}_k^{(0)} \sim \mathcal{N}(0, \mathbf{I}_d)$ and $\mathbf{a}_k^{(0)} \sim \mathcal{N}(0,1) \; \forall k \in [m]$. Then, the minimum singular value of the $\mathbf{G}$-matrix satisfies*

$$\lambda_{\min}\left(\mathbf{G}^{(0)}\right) \geq \Omega\left(\frac{\delta}{n^3}\right),$$

*with probability at least $1 - e^{-\Omega\left(\delta m/n^2\right)}$ with respect to $\{\mathbf{w}_k^{(0)}\}_{k=1}^m$ and $\{a_k^{(0)}\}_{k=1}^m$, given that $m$ satisfies*

$$m \geq \Omega\left(\frac{n^3}{\delta} \log \frac{n}{\delta^{1/4}}\right).$$

*Proof.* In the following, we will write $\mathbf{w}_k$ instead of $\mathbf{w}_k^{(0)}$ and $a_k$ instead of $a_k^{(0)}$. Consider the following sum for an arbitrary unit vector $\zeta$ and a random standard normal vector $\mathbf{w}$:

$$\sum_{i=1}^n \zeta_i \, \phi'\left(\mathbf{w}^T\mathbf{x}_i\right) \mathbf{x}_i.$$

To lower bound the lowest eigenvalue of the $\mathbf{G}$-matrix, we will give a lower bound on the norm of this vector. In order to do this, we use the following claim whose proof is deferred to later in the section.

**Claim K.2.** *For $\zeta \in \mathbb{S}^{n-1}$, let*

$$f(\mathbf{w}) = \sum_{i=1}^n \zeta_i \, \phi'\left(\mathbf{w}^T\mathbf{x}_i\right) \mathbf{x}_i.$$

*Then we have*

$$\Pr_{\mathbf{w}\sim\mathcal{N}(0,\mathbf{I}_d)}\left(\|f(\mathbf{w})\|_2 \geq \frac{0.1}{\sqrt{n}}\right) \geq \Omega\left(\frac{\delta}{n^2}\right).$$

From this claim, we have

$$\Pr_{\mathbf{w} \sim \mathcal{N}(0, \mathbf{I}_d)} \left( \left\| \sum_{i=1}^{n} \zeta_i \, \phi' \left( \mathbf{w}^T \mathbf{x}_i \right) \mathbf{x}_i \right\|_2 \geq \frac{0.1}{\sqrt{n}} \right) \geq \Omega \left( \frac{\delta}{n^2} \right).$$

Hence,

$$\Pr_{\mathbf{w} \sim \mathcal{N}(0, \mathbf{I}_d), \tilde{a} \sim \mathcal{N}(0,1)} \left( \left\| \sum_{i=1}^{n} \tilde{a} \zeta_i \, \phi' \left( \mathbf{w}^T \mathbf{x}_i \right) \mathbf{x}_i \right\|_2 \geq \frac{0.1}{\sqrt{n}} \right) \geq \Omega \left( \frac{\delta}{n^2} \right).$$

owing to the fact that for a standard normal variate $\tilde{a}, |\tilde{a}|$ is at least 1 with probability at least 0.2 using Fact C.1. Applying the Chernoff bounds, we have

$$\Pr_{\{W_k\}_{k=1}^m, \{a_k\}_{k=1}^m} \left( \frac{1}{m} \sum_{k=1}^{m} \left\| \sum_{i=1}^{n} \zeta_i a_k \phi' \left( \mathbf{w}_k^T \mathbf{x}_i \right) \mathbf{x}_i \right\|_2^2 \geq \frac{10^{-3} \delta}{n^3} \right) \geq 1 - e^{-\Omega \left( \frac{\delta m}{n^2} \right)}.$$

To get the bound for all $\zeta \in \mathbb{S}^{n-1}$, we use an $\epsilon$-net argument with $\epsilon = \Theta \left( \frac{\sqrt{\delta}}{n^2} \right)$ and $\epsilon$-net size $\left( \frac{1}{\epsilon} \right)^n$. This gives that

$$\frac{1}{m} \sum_{k=1}^{m} \left\| \sum_{i=1}^{n} \zeta_i \phi' \left( \mathbf{w}_k^T \mathbf{x}_i \right) \mathbf{x}_i \right\|_2^2 \geq \Omega \left( \frac{\delta}{n^3} \right)$$

holds for all $\zeta \in \mathbb{S}^{n-1}$ with probability at least

$$1 - \left( \frac{n^2}{\sqrt{\delta}} \right)^n e^{-\Omega \left( \frac{\delta m}{c n^2} \right)} \geq 1 - e^{-\Omega \left( \frac{\delta m}{n^2} \right)}$$

with respect to $\{\mathbf{w}_k\}_{k=1}^m$ and $\{a_k\}_{k=1}^m$, assuming that $m \geq \Omega \left( \frac{n^3}{\delta} \log \frac{n^4}{\delta} \right)$. Thus, using the fact that $\lambda_{\min} \left( \mathbf{G}^{(0)} \right) = \frac{1}{m} \sigma_{\min} \left( \mathbf{M} \right)^2$, we get the final bound. $\qquad \square$

**Corollary K.2.1.** *Let the activation be* ReLU*, then*

$$\lambda_{\min} \left( \mathbf{G}^{(0)} \right) \geq \Omega \left( \frac{\delta}{n^3} \right)$$

*with probability at least* $1 - e^{-\Omega \left( \frac{\delta m}{n^2} \right)}$ *with respect to* $\left\{ \mathbf{w}_k^{(0)} \right\}_{k=1}^m$ *and* $\left\{ a_k^{(0)} \right\}_{k=1}^m$*, given that* $m$ *satisfies*

$$m \geq \Omega \left( \frac{n^3}{\delta} \log \frac{n}{\delta^{1/4}} \right).$$

We now move on to showing the main claim required in the Theorem K.1. Claim K.2 is an adaptation of (Allen-Zhu et al., 2019, Claim 6.4) with a slightly different choice of parameters and exposition.

*Proof of Claim K.2* . Let $i^*$ denote $\arg \max_{i \in [n]} \zeta_i$. We split vector $\mathbf{w}$ into two independent weight vectors, as follows

$$\mathbf{w} = \mathbf{w}' + \mathbf{w}'',$$
$$\mathbf{w}' = \left( \mathbf{I}_d - \mathbf{x}_{i^*} \mathbf{x}_{i^*}^T \right) \mathbf{w} - \sqrt{1 - \theta^2} g_1 \mathbf{x}_{i^*},$$
$$\mathbf{w}'' = \theta g_2 \mathbf{x}_{i^*}, \tag{49}$$

where $g_1$ and $g_2$ are two independent Gaussian random variables following $\mathcal{N} \left( 0, 1 - \theta^2 \right)$ and $\mathcal{N} \left( 0, \theta^2 \right)$ respectively and we set $\theta = \frac{\delta}{n^2}$.

Let $\mathcal{E}$ denote the following event.

$$\mathcal{E} = \left\{ \left| \mathbf{w}'^T \mathbf{x}_{i^*} - \alpha \right| < \frac{\delta}{10n^2} \text{ and } \left| \mathbf{w}'^T \mathbf{x}_i - \alpha \right| > \frac{\delta}{4n^2} \ \forall i \in [n] \setminus \{i^*\} \text{ and } |\theta g_2| \in \left( \frac{\delta}{9n^2}, \frac{\delta}{5n^2} \right) \right\}.$$

Assuming event $\mathcal{E}$ occurs, for $i = i^*$ we have

$$\left| \mathbf{w}'^T \mathbf{x}_{i^*} - \alpha \right| \leq \frac{\delta}{10n^2}, \quad \left| \mathbf{w}''^T \mathbf{x}_{i^*} \right| = \left| \theta g_2 \mathbf{x}_{i^*}^T \mathbf{x}_{i^*} \right| \geq \frac{\delta}{9n^2},$$

and for $\forall i \neq i^*$ we have

$$\left| \mathbf{w}'^T \mathbf{x}_i - \alpha \right| > \frac{\delta}{4n^2}, \quad \left| \mathbf{w}''^T \mathbf{x}_i \right| = \left| \theta g_2 \mathbf{x}_{i^*}^T \mathbf{x}_i \right| \leq \frac{\delta}{5n^2}.$$

Hence, conditioned on $\mathcal{E}$, for $i \neq i^*$ we have $\mathcal{I}_{\mathbf{w}^T \mathbf{x}_i \geq \alpha} = \mathcal{I}_{\mathbf{w}'^T \mathbf{x}_i \geq \alpha}$ always and $\mathcal{I}_{\mathbf{w}^T \mathbf{x}_{i^*} \geq \alpha} \neq \mathcal{I}_{\mathbf{w}'^T \mathbf{x}_{i^*} \geq \alpha}$ with probability 1/2.

Conditioned on $\mathcal{E}$ and using triangle and Cauchy–Schwartz inequalities we get

$$\| \sum_{i \in [n], i \neq i^*} \zeta_i \, \phi'(\mathbf{w}^T \mathbf{x}_i) \mathbf{x}_i - \sum_{[n], i \neq i^*} \zeta_i \, \phi'(\mathbf{w}'^T \mathbf{x}_i) \mathbf{x}_i \|_2$$

$$\leq \sum_{i \in [n], i \neq i^*} \| \zeta_i \mathbf{x}_i \|_2 \left| \phi'(\mathbf{w}^T \mathbf{x}_i) - \phi'(\mathbf{w}'^T \mathbf{x}_i) \right|$$

$$\leq \left( \sum_{i \in [n], i \neq i^*} \| \zeta_i \mathbf{x}_i \|_2^2 \right)^{1/2} \left( \sum_{i \in [n], i \neq i^*} (\phi'(\mathbf{w}^T \mathbf{x}_i) - \phi'(\mathbf{w}'^T \mathbf{x}_i))^2 \right)^{1/2}$$

$$= \left( \sum_{i \in [n], i \neq i^*} (\phi'(\mathbf{w}^T \mathbf{x}_i) - \phi'(\mathbf{w}'^T \mathbf{x}_i))^2 \right)^{1/2}$$

$$\leq \left( \sum_{i \in [n], i \neq i^*} \left| \mathbf{w}^T \mathbf{x}_i - \mathbf{w}'^T \mathbf{x}_i \right|^2 \right)^{1/2} \tag{50}$$

$$\leq \mathcal{O}\left( \sqrt{n} \right) \frac{\delta}{5n^2} = \mathcal{O}\left( \frac{\delta}{5n^{1.5}} \right),$$

where we use our assumption that $\left| \phi''(x) \right| \leq 1$ for $x \in \mathbb{R} \setminus \{\alpha\}$ in Inequality 50. Conditioning on $\mathcal{E}$ was used in concluding that either $\phi'(\mathbf{w}^T \mathbf{x}_i) - \phi'(\mathbf{w}'^T \mathbf{x}_i) = \phi_1'(\mathbf{w}^T \mathbf{x}_i) - \phi_1'(\mathbf{w}'^T \mathbf{x}_i)$ or $\phi'(\mathbf{w}^T \mathbf{x}_i) - \phi'(\mathbf{w}'^T \mathbf{x}_i) = \phi_2'(\mathbf{w}^T \mathbf{x}_i) - \phi_2'(\mathbf{w}'^T \mathbf{x}_i)$. In other words, $\phi_1'$ and $\phi_2'$ "don't mix". Since $\left| \lim_{z \to \alpha^-} \phi'(z) - \lim_{z \to \alpha^+} \phi'(z) \right| = 1$ and $|\zeta_{i^*}| \geq \frac{1}{\sqrt{n}}$, we have

$$\Pr_{g_2} \left( \left\| \zeta_{i^*} \, \phi'\left( \mathbf{w}^T \mathbf{x}_{i^*} \right) \mathbf{x}_{i^*} - \zeta_{i^*} \, \phi'\left( \mathbf{w}'^T \mathbf{x}_{i^*} \right) \mathbf{x}_{i^*} \right\|_2 \geq \frac{1}{\sqrt{n}} \left( 1 - \frac{\delta}{5n^2} \right) \ \Big| \ \mathcal{E} \right) = \frac{1}{2}.$$

To see this, note that given conditioned on $\mathcal{E}$, with probability 0.5 with respect to $g_2$, $\mathbf{w}^T \mathbf{x}_i$ is going to cross the jump discontinuity at $\alpha$ and thus, $\phi'$ is going to change by at least 1, minus the maximum movement on either side of $\alpha$, which is bounded. Thus,

$$\Pr_{\mathbf{w}} \left( \left\| \sum_{i=1}^n \zeta_i \phi'\left( \mathbf{w}^T \mathbf{x}_i \right) \mathbf{x}_i \right\|_2 \geq \frac{0.1}{\sqrt{n}} \ \Big| \ \mathcal{E} \right) \geq 0.5.$$

We now need to show that $\mathcal{E}$ occurs with high probability. To do this, we state the following claim that we prove later.

**Claim K.3.** *Let all the variables be as in Claim K.3. Then,*

$$\Pr_{\mathbf{w}'} \left( \left| \mathbf{w}'^T \mathbf{x}_{i^*} - \alpha \right| \leq \frac{\delta}{10n^2} \text{ and } \left| \mathbf{w}'^T \mathbf{x}_i - \alpha \right| \geq \frac{\delta}{4n^2} \ \forall i \neq i^* \right) \geq \Omega\left( \frac{\delta}{n^2 \sqrt{1 - \theta^2}} \right).$$

From Claim K.3, we have

$$\Pr_{\mathbf{w}'}\left(\left|\mathbf{w}'^T\mathbf{x}_{i^*} - \alpha\right| \leq \frac{\delta}{10n^2} \text{ and } \left|\mathbf{w}'^T\mathbf{x}_i - \alpha\right| \geq \frac{\delta}{4n^2} \ \forall i \neq i^*\right) \geq \Omega\left(\frac{\delta}{\sqrt{1-\theta^2}n^2}\right).$$

Also, using the fact that $\theta g_2 \sim \mathcal{N}\left(0, \theta^2\right)$ and Fact C.1, we have

$$\Pr_{g_2}\left(|\theta g_2| \in \left(\frac{\delta}{9n^2}, \frac{\delta}{5n^2}\right)\right) \geq \frac{\frac{\delta}{5n^2}}{\frac{\delta}{n^2}} - \frac{2}{3}\frac{\frac{\delta}{9n^2}}{\frac{\delta}{n^2}} \geq 0.08.$$

Independence of $\mathbf{w}'$ and $\mathbf{w}''$ implies

$$\Pr_{\mathbf{w}}[\mathcal{E}] \geq \Omega\left(\frac{\delta}{n^2}\right).$$

Thus,

$$\Pr_{\mathbf{w}\sim\mathcal{N}(0,\mathbf{I}_d)}\left(\|f(\mathbf{w})\|_2 \geq \frac{0.1}{\sqrt{n}}\right) \geq \Pr_{\mathbf{w}\sim\mathcal{N}(0,\mathbf{I}_d)}\left(\|f(\mathbf{w})\|_2 \geq \frac{0.1}{\sqrt{n}}\bigg|\mathcal{E}\right)\Pr_{\mathbf{w}\sim\mathcal{N}(0,\mathbf{I}_d)}[\mathcal{E}]$$

$$\geq \Omega\left(\frac{\delta}{n^2}\right). \qquad \square$$

*Proof of K.3.* By the definition of $\mathbf{w}'$, $\mathbf{w}'^T\mathbf{x}_{i^*}$ is equal to $\sqrt{1-\theta^2}g_1$, which is distributed according to $\mathcal{N}\left(0, 1-\theta^2\right)$. Hence, applying concentration bounds from Fact C.1, we get that

$$\Pr_{\mathbf{w}'}\left(\left|\mathbf{w}'^T\mathbf{x}_{i^*} - \alpha\right| \leq \frac{\delta}{10n^2}\right) \geq \frac{\delta}{40n^2\sqrt{1-\theta^2}}. \qquad (51)$$

We can divide $\mathbf{w}' \ \forall i \in [n]$ into two parts:

- Component orthogonal to $\mathbf{x}_{i^*}$ given by $\mathbf{w}'^T\left(\mathbf{I}_d - \mathbf{x}_{i^*}\mathbf{x}_{i^*}^T\right)\mathbf{x}_i$

- Component parallel to $\mathbf{x}_{i^*}$ given by $\mathbf{w}'^T\left(\mathbf{x}_{i^*}\mathbf{x}_{i^*}^T\right)\mathbf{x}_i$.

This gives us

$$\mathbf{w}'^T\mathbf{x}_i = \underbrace{\mathbf{w}'^T\left(\mathbf{I}_d - \mathbf{x}_{i^*}\mathbf{x}_{i^*}^T\right)\mathbf{x}_i}_{\diamond} + \mathbf{w}'^T\left(\mathbf{x}_{i^*}\mathbf{x}_{i^*}^T\right)\mathbf{x}_i.$$

Conditioning on $\mathbf{w}'^T\mathbf{x}_{i^*}$ such that Equation 51 is satisfied, we get that $\mathbf{w}'^T\mathbf{x}_i$ is distributed according to

$$\mathbf{w}'^T\mathbf{x}_i \sim \mathcal{N}\left(\mathbf{w}'^T\left(\mathbf{x}_{i^*}\mathbf{x}_{i^*}^T\right)\mathbf{x}_i, \left(1-\theta^2\right)\left\|\left(\mathbf{I}_d - \mathbf{x}_{i^*}\mathbf{x}_{i^*}^T\right)\mathbf{x}_i\right\|_2^2\right).$$

By our assumption, $\left\|\left(\mathbf{I}_d - \mathbf{x}_{i^*}\mathbf{x}_{i^*}^T\right)\mathbf{x}_i\right\|_2 \geq \delta$. Also,

$$0 \leq \left|\mathbf{w}'^T\left(\mathbf{x}_{i^*}\mathbf{x}_{i^*}^T\right)\mathbf{x}_i\right| = \left|\mathbf{w}'^T\mathbf{x}_{i^*}\left(\mathbf{x}_{i^*}^T\mathbf{x}_i\right)\right| \leq \left|\alpha + \frac{\delta}{10n^2}\right| \leq 1 \qquad (52)$$

Hence, again applying concentration bounds from Fact C.1, we get for a fixed $i \neq i^*$

$$\Pr_{\mathbf{w}'}\left(\left|\mathbf{w}'^T\mathbf{x}_i - \alpha\right| \geq \frac{\delta}{4n^2}\right) \geq 1 - \frac{\delta}{5n^2\sqrt{1-\theta^2}\delta}.$$

Taking a union bound, we get that $\forall i \in [n]$ and $i \neq i^*$

$$\Pr_{\mathbf{w}'}\left(\left|\mathbf{w}'^T\mathbf{x}_i - \alpha\right| \geq \frac{\delta}{4n^2}\right) \geq 1 - \frac{1}{5n\sqrt{1-\theta^2}} \geq \frac{4}{5},$$

as required. $\qquad \square$

### K.1.2 $\mathbf{J_1}$ FOR STANDARD SETTING

The above theorems can be easily adapted to the standard settings, defined in Appendix D. We capture this with the following corollaries.

**Corollary K.3.1** (Adapting Theorem K.1 for Init (fanin) setting)**.** *Let the condition on $\phi$ be satisfied for $r = 1$. Assume, $\mathbf{w}_k^{(0)} \sim \mathcal{N}\left(0, \frac{1}{d}\mathbf{I}_d\right), b_k^{(0)} \sim \mathcal{N}(0, 0.01)$ and $a_k^{(0)} \sim \mathcal{N}\left(0, \frac{1}{m}\right) \quad \forall k \in [m]$. Then, the minimum singular value of the G-matrix satisfies*

$$\lambda_{\min}\left(\mathbf{G}^{(0)}\right) \geq \Omega\left(\frac{\delta}{n^3}\right),$$

*with probability at least $1 - e^{-\Omega\left(\delta m/n^2\right)}$ with respect to $\{\mathbf{w}_k^{(0)}\}_{k=1}^m$, $\{b_k^{(0)}\}_{k=1}^m$ and $\{a_k^{(0)}\}_{k=1}^m$, given that $m$ satisfies*

$$m \geq \Omega\left(\frac{n^3}{\delta}\log\frac{n}{\delta^{1/4}}\right).$$

**Corollary K.3.2** (Adapting Theorem K.1 for Init (fanout) setting)**.** *Let the condition on $\phi$ be satisfied for $r = 1$. Assume, $\mathbf{w}_k^{(0)} \sim \mathcal{N}\left(0, \frac{1}{m}\mathbf{I}_d\right), a_k^{(0)} \sim \mathcal{N}(0, 1)$ and $b_k^{(0)} \sim \mathcal{N}\left(0, \frac{1}{m}\right) \quad \forall k \in [m]$. Then, the minimum singular value of the G-matrix satisfies*

$$\lambda_{\min}\left(\mathbf{G}^{(0)}\right) \geq \Omega\left(\frac{\delta}{n^2}\right),$$

*with probability at least $1 - e^{-\Omega\left(\delta m/n\right)}$ with respect to $\{\mathbf{w}_k^{(0)}\}_{k=1}^m$, $\{b_k^{(0)}\}_{k=1}^m$ and $\{a_k^{(0)}\}_{k=1}^m$, given that $m$ satisfies*

$$m \geq \Omega\left(\frac{n^2}{\delta}\log\frac{n}{\delta^{1/2}}\right).$$

### K.2 $\mathbf{J_2}$ : THE SECOND DERIVATIVE HAS JUMP DISCONTINUITY AT A POINT

#### K.2.1 DZPS SETTING

Recall that this setting was defined in Section 2.

**Theorem K.4.** *Let $\phi$ satisfy the condition for $r = 2$. Assume that $\mathbf{w}_k^{(0)} \sim \mathcal{N}(0, \mathbf{I}_d)$ and $a_k^{(0)} \sim \mathcal{N}(0, 1) \; \forall k \in [m]$. Then, the minimum singular value of the $\mathbf{G}$-matrix satisfies*

$$\lambda_{\min}\left(\mathbf{G}^{(0)}\right) \geq \Omega\left(\frac{\delta^3}{n^7\log n}\right),$$

*with probability at least $1 - e^{-\Omega\left(\delta m/n^2\right)}$ with respect to $\{\mathbf{w}_k^{(0)}\}_{k=1}^m$ and $\{a_k^{(0)}\}_{k=1}^m$, given that $m$ satisfies*

$$m > \max\left(\Omega\left(\frac{n^3}{\delta}\log\left(\frac{n}{\delta^{1/3}}\right)\right), \Omega\left(\frac{n^2\log(d)}{\delta}\right)\right).$$

*Proof.* In the following, we will use $\mathbf{w}_k$ instead of $\mathbf{w}_k^{(0)}$ and $a_k$ instead of $a_k^{(0)}$. Referring to Equation 4, it suffices to show that

$$\left\|\sum_{i=1}^n \zeta_i\mathbf{m}_i\right\|_2^2 = \sum_{k=1}^m \left\|\sum_{i=1}^n \zeta_i a_k \phi'\left(\mathbf{w}_k^T\mathbf{x}_i\right)\mathbf{x}_i\right\|_2^2.$$

is lower bounded for all vectors $\zeta \in \mathbb{S}^{n-1}$ with high probability. Fix a particular $\zeta \in \mathbb{S}^{n-1}$. For each $k \in [m]$ and each $i \in [n]$, we have $\mathbf{w}_k^T\mathbf{x}_i \sim \mathcal{N}(0, 1)$. First, we analyze the sum for a fixed $k$, i.e. we consider $\sum_{i=1}^n \zeta_i \; \phi'\left(\mathbf{w}_k^T\mathbf{x}_i\right)\mathbf{x}_i$. We split vector $\mathbf{w}_k$ as

$$\mathbf{w}_k = \hat{\mathbf{w}}_k + \bar{\mathbf{w}}_k,$$

where $\hat{\mathbf{w}}_k$ and $\bar{\mathbf{w}}_k$ are two independent Gaussian vectors in $\mathbb{R}^d$ distributed according to $\mathcal{N}\left(0, \left(1 - \theta^2\right) \mathbf{I}_d\right)$ and $\mathcal{N}\left(0, \theta^2 \mathbf{I}_d\right)$ respectively. We set

$$\theta = \frac{\delta}{2000 \, n^2 \sqrt{\log n}}.$$

Define the event $C_{\mathbf{w}}$ on a weight vector $\mathbf{w}$ as

$$C_{\mathbf{w}} = \left\{\left|\mathbf{w}^T \mathbf{x}_i - \alpha\right| \geq \frac{\delta}{100 n^2} : i \in [n]\right\}.$$

Using Fact C.1 and the fact that $\mathbf{x}_i$ are unit vectors,

$$\Pr_{\mathbf{w} \sim \mathcal{N}(0, (1-\theta^2)\mathcal{I}_d)} [C_{\mathbf{w}}] \geq 1 - \frac{\delta}{400 n \sqrt{1 - \theta^2}}. \tag{53}$$

Define the event $D_{\mathbf{w}}$ on a weight vector $\mathbf{w}$ as

$$D_{\mathbf{w}} = \left\{\left|\mathbf{w}^T \mathbf{x}_i\right| \leq \frac{\delta}{500 n^2} : i \in [n]\right\}.$$

Fact C.2 shows that,

$$\Pr_{\mathbf{w} \sim \mathcal{N}(0, \theta^2 \mathbf{I}_d)} [D_{\mathbf{w}}] \geq \left(1 - 2n \exp\left(\frac{-t^2}{2}\right)\right) \tag{54}$$

$$\geq 1 - \frac{2}{n^7} \tag{55}$$

$$\geq 0.5, \tag{56}$$

where we set $t = \frac{\delta}{500 n^2 \theta}$ to get Inequality 55.

We want $\hat{\mathbf{w}}_k$ to satisfy condition $C_{\hat{\mathbf{w}}_k}$ and $\bar{\mathbf{w}}_k$ to satisfy condition $D_{\bar{\mathbf{w}}_k}$. Since, $\hat{\mathbf{w}}_k$ and $\bar{\mathbf{w}}_k$ are independent of each other, we use Equation 53 and Equation 54 to get

$$P_{\bar{\mathbf{w}}_k, \hat{\mathbf{w}}_k}\left(C_{\hat{\mathbf{w}}_k} \text{ and } D_{\bar{\mathbf{w}}_k}\right) \geq 0.25.$$

Assuming both the conditions hold, it follows that $\mathcal{I}_{\hat{\mathbf{w}}_k^T \mathbf{x}_i \geq \alpha} = \mathcal{I}_{\mathbf{w}_k^T \mathbf{x}_i \geq \alpha}$. We will work conditioned on both the events.

Define a function $f$ as follows,

$$f(\mathbf{w}) = \sum_{i=1}^{n} \zeta_i \, \phi'(\mathbf{w}^T \mathbf{x}_i).$$

Note that

$$\nabla_{\mathbf{w}} f(\mathbf{w}) = \sum_{i=1}^{n} \zeta_i \, \phi''(\mathbf{w}^T \mathbf{x}_i) \mathbf{x}_i.$$

In the sequel, we will use $f'$ for $\nabla_{\mathbf{w}} f(\mathbf{w})$ and $f''$ for $\nabla^2_{\mathbf{w}} f(\mathbf{w})$. It is easy to see that the only discontinuities of the derivative of function $f$ are when $\mathbf{w}^T \mathbf{x}_i = \alpha$, since it is the only point of discontinuity for $\phi''$. Thus, assuming that $C_{\hat{\mathbf{w}}_k}$ and $D_{\bar{\mathbf{w}}_k}$ hold, we can apply Taylor expansion to $f(\hat{\mathbf{w}}_k)$ for a perturbation of $\bar{\mathbf{w}}_k$, ensuring that all the derivatives exist in the neighborhood of interest. Hence,

$$f(\mathbf{w}_k) = f(\hat{\mathbf{w}}_k) + \langle \bar{\mathbf{w}}_k, f'(\hat{\mathbf{w}}_k) \rangle + R_2(\mathbf{w}_k),$$

where $R_2$ denotes the second order remainder term in the Taylor expansion given by

$$R_2(\mathbf{w}_k) = \frac{1}{2} \int_{t=0}^{1} \left\langle f''(\hat{\mathbf{w}}_k + t\bar{\mathbf{w}}_k), \bar{\mathbf{w}}_k^{\otimes 2} \right\rangle dt.$$

Using $\nabla^2_{\mathbf{w}} f(\mathbf{w}) = \sum_{i=1}^{n} \zeta_i \mathbf{x}_i^{\otimes 2} \phi^{(3)}(z) \Big|_{z = \langle \mathbf{w}, \mathbf{x}_i \rangle}$, we have

$$R_2(\mathbf{w}_k) = \frac{1}{2} \int_{t=0}^{1} \sum_{i=1}^{n} \zeta_i \left(\langle \bar{\mathbf{w}}_k, \mathbf{x}_i \rangle\right)^2 \phi^{(3)}\left(\langle \hat{\mathbf{w}}_k + t\bar{\mathbf{w}}_k, \mathbf{x}_i \rangle\right) dt.$$

The magnitude of this term can be bounded as follows.

$$\left| R_2 \left( \mathbf{w}_k \right) \right| = \frac{1}{2} \sum_{i=1}^{n} \zeta_i \left( \langle \bar{\mathbf{w}}_k, \mathbf{x}_i \rangle \right)^2 \left( \int_{t=0}^{1} \phi^{(3)} \left( \langle \hat{\mathbf{w}}_k + t \bar{\mathbf{w}}_k, \mathbf{x}_i \rangle \right) dt \right)$$

$$\leq \frac{1}{2} \sqrt{\sum_{i=1}^{n} \left( \int_{t=0}^{1} \phi^{(3)} \left( \langle \hat{\mathbf{w}}_k + t \bar{\mathbf{w}}_k, \mathbf{x}_i \rangle \right) dt \right)^2} \sqrt{\sum_{i=1}^{n} \zeta_i^2 \left( \langle \bar{\mathbf{w}}_k, \mathbf{x}_i \rangle \right)^4} \qquad (57)$$

$$\leq \frac{1}{2} \sqrt{n} \left( \frac{\delta}{500 n^2} \right)^2 \qquad (58)$$

$$\leq \mathcal{O} \left( \frac{\delta^2}{n^{3.5}} \right),$$

where Inequality 57 uses Cauchy-Schwartz inequality, Inequality 58 uses the fact that all the derivatives of $\phi$ of order up to $r+1$ are bounded for $x \neq 0$ and $\bar{\mathbf{w}}_k$ satisfies condition $D_{\bar{\mathbf{w}}_k}$ and $\hat{\mathbf{w}}_k$ satisfies $C_{\hat{\mathbf{w}}_k}$. Thus we have

$$f(\mathbf{w}_k) = f(\hat{\mathbf{w}}_k) + \langle \bar{\mathbf{w}}_k, f'(\hat{\mathbf{w}}_k) \rangle + \mathcal{O} \left( \frac{\delta^2}{n^{3.5}} \right). \qquad (59)$$

Consider the following two cases.

**Case 1:** $\left| f(\hat{\mathbf{w}}_k) \right| < \frac{1}{2} \sqrt{\frac{0.01}{n}} \theta$.

First, we condition on the event that $\hat{\mathbf{w}}_k$ is picked so that $\left\| f'(\hat{\mathbf{w}}_k) \right\|_2^2 \geq \frac{0.01}{n}$ and $C_{\hat{\mathbf{w}}_k}$ holds true. We shall refer to this condition as $B_{\hat{\mathbf{w}}_k}$. By Claim K.5, we get that

$$\Pr_{\hat{\mathbf{w}}_k} \left[ B_{\hat{\mathbf{w}}_k} \right] = \Pr_{\hat{\mathbf{w}}_k} \left( \left\| f'(\hat{\mathbf{w}}_k) \right\|_2^2 \geq \frac{0.01}{n} \text{ and } C_{\hat{\mathbf{w}}_k} \right) \geq \Omega \left( \frac{\delta}{(1 - \theta^2) n^2} \right). \qquad (60)$$

$\langle \bar{\mathbf{w}}_k, f'(\hat{\mathbf{w}}_k) \rangle$ is a random variable following $\mathcal{N} \left( 0, \theta^2 \left\| f'(\hat{\mathbf{w}}_k) \right\|_2^2 \right)$. Thus applying Fact C.1,

$$\Pr_{\bar{\mathbf{w}}_k} \left( \left| \langle \bar{\mathbf{w}}_k, f'(\hat{\mathbf{w}}_k) \rangle \right| \geq \sqrt{\frac{0.01}{n}} \theta \ \middle| \ B_{\hat{\mathbf{w}}_k} \right) \geq \frac{1}{5}.$$

Now letting event $\bar{D}_{\bar{\mathbf{w}}_k}$ denote the complement of the event $D_{\bar{\mathbf{w}}_k}$ we have

$$\Pr_{\bar{\mathbf{w}}_k} \left( \left| \langle \bar{\mathbf{w}}_k, f'(\hat{W}_k) \rangle \right| \geq \sqrt{\frac{0.01}{n}} \theta \ \bigwedge \ D_{\bar{W}_k} \middle| B_{\hat{W}_k} \right)$$

$$= \Pr_{\bar{\mathbf{w}}_k} \left( \left| \langle \bar{\mathbf{w}}_k, f'(\hat{\mathbf{w}}_k) \rangle \right| \geq \sqrt{\frac{0.01}{n}} \theta \ \middle| \ B_{\hat{\mathbf{w}}_k} \right) - \Pr_{\bar{\mathbf{w}}_k} \left( \left| \langle \bar{\mathbf{w}}_k, f'(\hat{\mathbf{w}}_k) \rangle \right| \geq \sqrt{\frac{0.01}{n}} \theta \ \middle| \ B_{\hat{\mathbf{w}}_k}, \bar{D}_{\bar{\mathbf{w}}_k} \right) \Pr_{\bar{\mathbf{w}}_k} \left[ \bar{D}_{\bar{\mathbf{w}}_k} \right]$$

$$\geq \frac{1}{5} - \frac{2}{n^7}$$

$$\geq 1/10.$$

Hence, from Equation 59, we get

$$\Pr_{\bar{\mathbf{w}}_k} \left( \left| f(\mathbf{w}_k) \right| \geq \frac{1}{2} \sqrt{\frac{0.01}{n}} \theta \text{ and } D_{\bar{\mathbf{w}}_k} \middle| B_{\hat{\mathbf{w}}_k} \right) \geq 0.1.$$

Thus,

$$\Pr_{\mathbf{w}_k} \left( \left| f(\mathbf{w}_k) \right| \geq \frac{1}{2} \sqrt{\frac{0.01}{n}} \theta \right) \geq \Pr_{\mathbf{w}_k} \left( \left| f(\mathbf{w}_k) \right| \geq \frac{1}{2} \sqrt{\frac{0.01}{n}} \theta \text{ and } B_{\hat{\mathbf{w}}_k} \text{ and } D_{\bar{\mathbf{w}}_k} \right)$$

$$\geq \Pr_{\hat{\mathbf{w}}_k} \left[ B_{\hat{\mathbf{w}}_k} \right] \Pr_{\bar{\mathbf{w}}_k} \left( \left| f(\mathbf{w}_k) \right| \geq \frac{1}{2} \sqrt{\frac{0.01}{n}} \theta \text{ and } D_{\bar{\mathbf{w}}_k} \middle| B_{\hat{\mathbf{w}}_k} \right)$$

$$\geq \Omega \left( \frac{\delta}{n^2} \right)$$

**Case 2:** $\left|f(\hat{\mathbf{w}}_k)\right| \geq \frac{1}{2}\sqrt{\frac{0.01}{n}}\theta$.

We can upper-bound the magnitude of $f'(\hat{\mathbf{w}}_k)$ by $\mathcal{O}(\sqrt{n})$ as follows.

$$\left\|f'(\hat{\mathbf{w}}_k)\right\| = \left\|\sum_{i=1}^{n}\zeta_i\phi''(\hat{\mathbf{w}}_k^T\mathbf{x}_i)\mathbf{x}_i\right\| \leq \sqrt{\sum_{i=1}^{n}\zeta_i^2\phi''(\hat{\mathbf{w}}_k^T\mathbf{x}_i)^2}\sqrt{\sum_{i=1}^{n}\|\mathbf{x}_i\|^2} \leq \mathcal{O}(\sqrt{n}).$$

Here we use the fact that $\|\zeta\| = 1$, $\phi''$ is bounded by a constant at all $x \neq \alpha$ and $\|\mathbf{x}_i\| = 1$ for $i \in [n]$. Note that, this bound always holds true, irrespective of the value of $\hat{\mathbf{w}}_k$. Again, using the fact that $\langle\bar{\mathbf{w}}_k, f'(\hat{\mathbf{w}}_k)\rangle$ is a Gaussian variable following $\mathcal{N}(0, \theta^2\|f'(\hat{\mathbf{w}}_k)\|^2)$, Fact C.1 shows that,

$$\Pr_{\bar{\mathbf{w}}_k}\left(\left|\langle\bar{\mathbf{w}}_k, f'(\hat{\mathbf{w}}_k)\rangle\right| \leq \frac{1}{4}\sqrt{\frac{0.01}{n}}\theta \,\middle|\, C_{\hat{\mathbf{w}}_k}\right) \geq \frac{0.1}{4}\frac{2/3}{\sqrt{n}\|f'(\hat{\mathbf{w}}_k)\|} \geq \frac{0.1}{4}\frac{2/3}{n}.$$

Now,

$$\Pr_{\bar{\mathbf{w}}_k}\left(\left|\langle\bar{\mathbf{w}}_k, f'(\hat{\mathbf{w}}_k)\rangle\right| \leq \frac{1}{4}\sqrt{\frac{0.01}{n}}\theta \,\bigwedge\, D_{\bar{\mathbf{w}}_k} \,\middle|\, C_{\hat{\mathbf{w}}_k}\right)$$

$$= \Pr_{\bar{\mathbf{w}}_k}\left(\left|\langle\bar{\mathbf{w}}_k, f'(\hat{\mathbf{w}}_k)\rangle\right| \leq \frac{1}{4}\sqrt{\frac{0.01}{n}}\theta \,\middle|\, C_{\hat{\mathbf{w}}_k}\right) - \Pr_{\bar{\mathbf{w}}_k}\left(\left|\langle\bar{\mathbf{w}}_k, f'(\hat{\mathbf{w}}_k)\rangle\right| \leq \frac{1}{4}\sqrt{\frac{0.01}{n}}\theta \,\middle|\, C_{\hat{\mathbf{w}}_k}, \bar{D}_{\bar{\mathbf{w}}_k}\right)\Pr_{\bar{\mathbf{w}}_k}\left[\bar{D}_{\bar{\mathbf{w}}_k}\right]$$

$$\geq \frac{1}{60n} - \frac{2}{n^7}$$

$$\geq \Omega\left(\frac{1}{n}\right).$$

Hence, from Equation 59, we get

$$\Pr_{\bar{\mathbf{w}}_k}\left(\left|f(\mathbf{w}_k)\right| \geq \frac{1}{4}\sqrt{\frac{0.01}{n}}\theta \,\bigwedge\, D_{\bar{\mathbf{w}}_k}\middle|C_{\hat{\mathbf{w}}_k}\right) \geq \Omega\left(\frac{1}{n}\right).$$

Thus,

$$\Pr_{\mathbf{w}_k}\left(\left|f(\mathbf{w}_k)\right| \geq \frac{1}{4}\sqrt{\frac{0.01}{n}}\theta\right) \geq \Pr_{\mathbf{w}_k}\left(\left|f(\mathbf{w}_k)\right| \geq \frac{1}{4}\sqrt{\frac{0.01}{n}}\theta \text{ and } C_{\hat{\mathbf{w}}_k} \,\bigwedge\, D_{\bar{\mathbf{w}}_k}\right)$$

$$\geq \Pr_{\hat{\mathbf{w}}_k}\left[C_{\hat{\mathbf{w}}_k}\right]\Pr_{\bar{\mathbf{w}}_k}\left(\left|f(\mathbf{w}_k)\right| \geq \frac{1}{4}\sqrt{\frac{0.01}{n}}\theta \text{ and } D_{\bar{\mathbf{w}}_k}\middle|C_{\hat{\mathbf{w}}_k}\right)$$

$$\geq \Omega\left(\frac{1}{n}\right).$$

Thus, combining the two cases, we have

$$\Pr_{\mathbf{w}_k}\left(\left|f(\mathbf{w}_k)\right| \geq \Omega\left(\frac{\delta}{n^{2.5}\sqrt{\log n}}\right)\right) \geq \Omega\left(\frac{\delta}{n^2}\right). \tag{61}$$

Hence,

$$\Pr_{\mathbf{w}_k, a_k}\left(\left|\sum_{i=1}^{n}\zeta_i a_k\phi'(\mathbf{w}_k^T x_i)\right| \geq \Omega\left(\frac{\delta}{n^{2.5}\sqrt{\log n}}\right)\right) \geq \Omega\left(\frac{\delta}{n^2}\right). \tag{62}$$

owing to the fact that for a standard normal variate $\tilde{a}, |\tilde{a}|$ is at-least 1, with probability at-least 0.2 using Fact C.1. Applying a Chernoff bound over all $k \in [m]$, we get

$$\sum_{k=1}^{m}\left(\sum_{i=1}^{n}\zeta_i\phi'\left(\mathbf{w}_k^T\mathbf{x}_i\right)\right)^2 \geq \Omega\left(\frac{\delta^3 m}{n^7 \log n}\right)$$

with probability at least $1 - \exp(-\Omega(\frac{\delta m}{n^2}))$ with respect to $\{\mathbf{w}_k\}_{k=1}^m$ and $\{a_k\}_{k=1}^m$.. Applying an $\epsilon$-net argument over $\zeta \in \mathbb{S}^{n-1}$, with $\epsilon = \frac{\delta^{1.5}}{2n^4\sqrt{\log n}}$ and $\epsilon$-net size $\left(\frac{1}{\epsilon}\right)^n$, we get that

$$\sum_{k=1}^m \left( \sum_{i=1}^n \zeta_i \phi'(\mathbf{w}_k^T \mathbf{x}_i) \right)^2 \geq \Omega\left( \frac{\delta^3 m}{n^7 \log(n)} \right) \tag{63}$$

holds for all $\zeta \in \mathbb{S}^{n-1}$ with probability at least

$$1 - \left( \frac{2n^4\sqrt{\log n}}{\delta^{1.5}} \right)^n e^{-\Omega\left(\frac{m\delta}{n^2}\right)} \geq 1 - e^{-\Omega\left(\frac{m\delta}{n^2}\right)}$$

with respect to $\{\mathbf{w}_k\}_{k=1}^m$ and $\{a_k\}_{k=1}^m$., where we assume that $m > \Omega\left( \frac{n^3}{\delta} \log\left( \frac{n}{\delta^{1/3}} \right) \right)$. Now consider the following function,

$$f(\mathbf{w}) = \sum_{k=1}^m \sum_{i=1}^n \zeta_i a_k \phi'\left( \mathbf{w}_k^T x_i \right) \mathbf{x}_i$$

where $\zeta \in \mathbb{S}^{n-1}$. Note that $f(\mathbf{w})$ is a $d$-dimensional vector. Also, the above can be written as

$$f(\mathbf{w}) = \mathbf{Q}\mathbf{v}$$

where $\mathbf{Q} = [q_{ij}] \in \mathbb{R}^{d \times n}$ is defined by

$$q_{i,j} = \zeta_j x_{j,i}$$

and $\mathbf{v} \in \mathbb{R}^n$, defined by

$$v_i = a_k \phi'(\mathbf{w}_k^T \mathbf{x}_i).$$

Also, since $\|\zeta\| = 1$ and $\|\mathbf{x}_i\| = 1 \, \forall i \in [n]$, we have $\|\mathbf{Q}\|_F = 1$. Consider the following quantity $\sum_{j=1}^n q_{i,j} v_j$. This quantity denotes the dot product of a row vector of $\mathbf{Q}$ and $\mathbf{v}$. We can apply Equation 63 to get

$$\left| \sum_{j=1}^n \frac{q_{i,j}}{\|\mathbf{q}_i\|} v_j \right|^2 \geq \Omega\left( \frac{\delta^3 m}{n^7 \log n} \right)$$

holds true with probability at-least $1 - e^{-\Omega\left(\frac{\delta m}{n^2}\right)}$ with respect to $\{\mathbf{w}_k\}_{k=1}^m$ and $\{a_k\}_{k=1}^m$. Note that, the coefficients have been normalized to unit norm to satisfy the condition based on which Equation 63 was derived. We can take a union bound over all the rows of $\mathbf{Q}$ to get

$$\|f(\mathbf{w})\|^2 = \sum_{i=1}^d \left| \sum_{j=1}^n q_{i,j} v_j \right|^2 \geq \sum_{i=1}^d \|\mathbf{q}_i\|^2 \Omega\left( \frac{\delta^3 m}{n^7 \log n} \right)$$

$$= \|\mathbf{Q}\|_F^2 \, \Omega\left( \frac{\delta^3 m}{n^7 \log n} \right) = \Omega\left( \frac{\delta^3 m}{n^7 \log n} \right)$$

with probability at least

$$1 - d e^{-\Omega\left(\frac{\delta m}{n^2}\right)} \geq 1 - e^{-\Omega\left(\frac{\delta m}{n^2}\right)}$$

with probability at least $1 - e^{-\Omega\left(\frac{\delta m}{n^2}\right)}$ with respect to $\{\mathbf{w}_k\}_{k=1}^m$ and $\{a_k\}_{k=1}^m$, assuming that $m \geq \Omega\left( \frac{n^2 \log d}{\delta} \right)$. Thus, we can use Equation 4 to show that,

$$\sigma_{\min}(\mathbf{M}) \geq \Omega\left( \sqrt{\frac{\delta^3 m}{n^7 \log n}} \right).$$

Using the fact that $\lambda_{\min}\left(\mathbf{G}^{(0)}\right) = \frac{1}{m}\sigma_{\min}(\mathbf{M})^2$, we get that $\lambda_{\min}\left(\mathbf{M}^{(0)}\right) \geq \Omega(\frac{\delta^3}{n^7 \log n})$ with probability at least $1 - e^{-\Omega\left(\frac{\delta m}{n^2}\right)}$. □

ELU satisfies the conditions required for Theorem K.4. We state this explicitly in the following theorem.

**Corollary K.4.1.** *Assume* $\mathbf{w}_k^{(0)} \sim \mathcal{N}(0, \mathbf{I}_d)$ *and* $a_k^{(0)} \sim \mathcal{N}(0,1) \, \forall k \in [m]$, *if* $\phi(x) = \mathcal{I}_{x<0}(e^x - 1) + \mathcal{I}_{x\geq 0}x$, *we have that for the* $\mathbf{G}$-*matrix,*

$$\lambda_{\min}\left(\mathbf{G}^{(0)}\right) \geq \Omega\left(\frac{\delta^3}{n^7 \log n}\right)$$

*with probability at least* $1 - e^{-\Omega\left(\frac{\delta m}{n^2}\right)}$ *with respect to* $\left\{\mathbf{w}_k^{(0)}\right\}_{k=1}^{m}$, *given that* $m$ *satisfies*

$$m > \max\left(\Omega\left(\frac{n^3}{\delta}\log\left(\frac{n}{\delta^{1/3}}\right)\right), \Omega\left(\frac{n^2 \log(d)}{\delta}\right)\right).$$

**Claim K.5.** *Let the variables have the same meaning as in the theorem Theorem K.4. Then,*

$$\Pr_{\mathbf{w}\sim\mathcal{N}(0,c^2\mathbf{I}_d)}\left(\|f'(\mathbf{w})\|_2^2 \geq \frac{1}{4n} \text{ and } C_{\mathbf{w}}\right) \geq \Omega(\frac{\delta}{cn^2})$$

*for a variable* $c$ *that depends on* $n$ *and* $\delta$.

*Proof.* Let $i^*$ denote $\arg\max_{i\in[n]} \zeta_i$. We can split w as $\mathbf{w} = \mathbf{w}' + \mathbf{w}''$, where

$$\mathbf{w}' = (\mathbf{I}_d - \mathbf{x}_{i^*}\mathbf{x}_{i^*}^T)\mathbf{w} + \sqrt{1-\theta^2}g_1\mathbf{x}_{i^*}$$

and

$$\mathbf{w}'' = \theta g_2 \mathbf{x}_{i^*}$$

where $\theta = \frac{\delta}{n^2}$ and $g_1, g_2$ are two independent gaussians $\sim \mathcal{N}(0,c^2)$. Let $\mathcal{E}$ denote the following event.

$$\mathcal{E} = \left\{\left|\mathbf{w}'^T\mathbf{x}_{i^*} - \alpha\right| < \frac{\delta}{10n^2} \text{ and } \left|\mathbf{w}'^T\mathbf{x}_i - \alpha\right| > \frac{\delta}{4n^2} \, \forall i \in [n], i \neq i^* \text{ and } |\theta g_2| \in \left(\frac{\delta}{9n^2}, \frac{\delta}{5n^2}\right)\right\}$$

Event $\mathcal{E}$ satisfies condition $C_{\mathbf{w}}$ because for $i^*$,

$$\left|\mathbf{w}^T\mathbf{x}_i^* - \alpha\right| = \left|\mathbf{w}'^T\mathbf{x}_{i^*} + \mathbf{w}''^T\mathbf{x}_{i^*} - \alpha\right| > \left|\left|\mathbf{w}'^T\mathbf{x}_{i^*} - \alpha\right| - \left|\mathbf{w}''^T\mathbf{x}_{i^*}\right|\right| > \frac{\delta}{90n^2}$$

and for all $i \neq i^*$,

$$\left|\mathbf{w}^T\mathbf{x}_i - \alpha\right| = \left|\mathbf{w}'^T\mathbf{x}_i + \mathbf{w}''^T\mathbf{x}_i - \alpha\right| > \left|\left|\mathbf{w}'^T\mathbf{x}_i - \alpha\right| - \left|\mathbf{w}''^T\mathbf{x}_i\right|\right| > \frac{\delta}{20n^2}$$

Hence, we can write that

$$\Pr_{\mathbf{w}\sim\mathcal{N}(0,c^2\mathbf{I}_d)}\left(\|f'(\mathbf{w})\|_2^2 \geq \frac{1}{4n} \text{ and } C_{\mathbf{w}}\right)$$

$$\geq \Pr_{\mathbf{w}\sim\mathcal{N}(0,(1-\theta^2)\mathbf{I}_d)}\left(\|f'(\mathbf{w})\|_2^2 \geq \frac{1}{4n} \text{ and } C_{\mathbf{w}}\middle|\mathcal{E}\right) \Pr_{\mathbf{w}\sim\mathcal{N}(0,(1-\theta^2)\mathbf{I}_d)}\mathcal{E}$$

$$= \Pr_{\mathbf{w}\sim\mathcal{N}(0,(1-\theta^2)\mathbf{I}_d)}\left(\|f'(\mathbf{w})\|_2^2 \geq \frac{1}{4n}\middle|\mathcal{E}\right) \Pr_{\mathbf{w}\sim\mathcal{N}(0,(1-\theta^2)\mathbf{I}_d)}\mathcal{E}$$

$$\geq \frac{1}{2}\frac{0.2\delta}{cn^2} = \Omega(\frac{\delta}{cn^2}),$$

where we use Claim K.6 in the final step. $\square$

**Claim K.6.**

$$\Pr_{\mathbf{w}\sim\mathcal{N}(0,c^2\mathbf{I}_d)}\left(\|f'(\mathbf{w})\|_2^2\geq\frac{0.01}{n}\bigg|\mathcal{E}\right)\geq\frac{1}{2}$$

*and*

$$\Pr_{\mathbf{w}\sim\mathcal{N}(0,c^2\mathbf{I}_d)}(\mathcal{E})\geq\frac{0.08\delta}{c^2n^2}$$

*where $\mathcal{E}$ is defined and $\mathbf{w}$ has been split into $\mathbf{w}'$ and $\mathbf{w}''$ as in proof of* Claim K.5.

*Proof.* This follows from the proof of Claim K.2, with a slight change in distribution of $\mathbf{w}$ from $\mathcal{N}(0,\mathbf{I}_d)$ to $\mathcal{N}(0,c^2\mathbf{I}_d)$ and the function under consideration is changed to

$$f'(\mathbf{w})=\sum_{i=1}^n\zeta_i\phi''\left(\mathbf{w}^T\mathbf{x}_i\right)\mathbf{x}_i.$$

$\square$

### K.2.2 $\mathbf{J_2}$ FOR STANDARD SETTING

As before, we state the main theorem for standard initializations, defined in Appendix D, as corollaries.

**Corollary K.6.1** (Adapting Theorem K.4 for $\mathsf{Init}(\mathsf{fanin})$ setting)**.** *Let the condition on $\phi$ be satisfied for $r=1$. Assume, $\mathbf{w}_k^{(0)}\sim\mathcal{N}\left(0,\frac{1}{d}\mathbf{I}_d\right),b_k^{(0)}\sim\mathcal{N}(0,0.01)$ and $a_k^{(0)}\sim\mathcal{N}\left(0,\frac{1}{m}\right)\quad\forall k\in[m]$. Then, the minimum singular value of the $\mathbf{G}$-matrix satisfies*

$$\lambda_{\min}\left(\mathbf{G}^{(0)}\right)\geq\Omega\left(\frac{\delta^3}{n^7d\log n}\right)$$

*with probability at least $1-e^{-\Omega\left(\frac{\delta m}{n^2}\right)}$ with respect to $\left\{\mathbf{w}_k^{(0)}\right\}_{k=1}^m$, given that $m$ satisfies*

$$m>\max\left(\Omega\left(\frac{n^3}{\delta}\log\left(\frac{n}{\delta^{1/3}d^{1/9}}\right)\right),\Omega\left(\frac{n^2\log(d)}{\delta}\right)\right).$$

**Corollary K.6.2** (Adapting Theorem K.4 for $\mathsf{Init}(\mathsf{fanout})$ setting)**.** *Let the condition on $\phi$ be satisfied for $r=1$. Assume, $\mathbf{w}_k^{(0)}\sim\mathcal{N}\left(0,\frac{1}{m}\mathbf{I}_d\right),b_k^{(0)}\sim\mathcal{N}\left(0,\frac{1}{m}\right)$ and $a_k^{(0)}\sim\mathcal{N}(0,1)\quad\forall k\in[m]$. Then, the minimum singular value of the $\mathbf{G}$-matrix satisfies*

$$\lambda_{\min}\left(\mathbf{G}^{(0)}\right)\geq\Omega\left(\frac{\delta^2}{mn^4\log n}\right),$$

*with probability at least $1-e^{-\Omega(\delta m/n)}$ with respect to $\{\mathbf{w}_k^{(0)}\}_{k=1}^m$, $\{b_k^{(0)}\}_{k=1}^m$ and $\{a_k^{(0)}\}_{k=1}^m$, given that $m$ satisfies*

$$m\geq\Omega\left(\frac{n^2}{\delta}\log\frac{mn^5\log n}{\delta^2}\right).$$

## L    A NEW PROOF FOR THE MINIMUM EIGENVALUE FOR ReLU

**Theorem L.1** (Thm 3.3 in Du et al. (2019a)). *Consider the 2-layer feed forward network in Equation 1. Assume that $\mathbf{w}_k^{(0)} \sim \mathcal{N}(0,1)$ and $a_k^{(0)} \sim \mathbb{U}\{-1,+1\}$ $\forall k \in [m]$, Assumption 1 and Assumption 2 hold true and $|y_i| \leq C$, for some constant $C$. Then, if we use gradient flow optimization and set the number of hidden nodes $m = \Omega\left(\frac{n^6 \log \frac{m}{\kappa}}{\lambda_{\min}(\mathbf{G}^\infty)^4 \kappa^3}\right)$, with probability at least $1 - \kappa$ over the initialization we have*

$$\|\mathbf{u}_t - \mathbf{y}\|^2 \leq e^{-\lambda_{\min}(\mathbf{G}^\infty)t}\|\mathbf{u}_0 - \mathbf{y}\|$$

*Remark.* The above proof can be adapted for the gradient descent algorithm with a learning rate less than $\mathcal{O}\left(\lambda_{\min}(\mathbf{G}^\infty)/n^2\right)$, following the proof of Theorem 5.1 in Du et al. (2019b) and theorem 4.1 in Du et al. (2019a) without substantial changes in the bounds.

**Theorem L.2.** *Assume that we are in the setting of Theorem L.1. If the activation is ReLU and $m \geq \Omega\left(n^4 \delta^{-3} \log^4 n\right)$, then with probability at least $1 - \exp\left(-\Omega\left(\frac{m\delta^3}{n^2 \log^3 n}\right)\right)$*

$$\lambda_{\min}\left(\mathbf{G}^{(0)}\right) \geq \Omega\left(\left(\frac{\delta}{\log n}\right)^{1.5}\right).$$

*Remark.* Compare the previous theorem with Cor. K.2.1.

*Proof.* Using Lemma I.1, we have

$$\lambda_{\min}\left(\mathbf{G}^\infty\right) \geq \bar{c}_r^2\left(\phi'\right) \lambda_{\min}\left((\mathbf{X}^{*r})^T \mathbf{X}^{*r}\right), \quad \forall r \in \mathbb{Z}^+$$

where $\mathbf{X}$ denotes a $d \times n$ matrix, with its $i$-th column containing $\mathbf{x}_i$ and $\mathbf{X}^{*r} \in \mathbb{R}^{d^r \times n}$ denotes its order-$r$ Khatri-Rao product. Note that, by Assumption 1, the columns of $\mathbf{X}^{*r}$ are unit normalized euclidean vectors. If for any $i, j \in [n]$, $\mathbf{x}_i^T \mathbf{x}_j = \rho < 1$, then we can see that $\left(\mathbf{x}_i^{*r}\right)^T \mathbf{x}_j^{*r} = \rho^r$, where $\mathbf{x}^{*r}$ denotes the order-$r$ Khatri–Rao product of a vector $\mathbf{x}$. Also, $|\rho|$ can be shown to be at most $1 - \delta$, using Assumption 2. Thus, for the magnitude of $\left(\mathbf{x}_i^{*r}\right)^T \mathbf{x}_j^{*r}$, for any $i, j \in [n], i \neq j$, to be less than $\frac{1}{2n}$, we must have $r \geq r_0 = \frac{\log 2n}{\delta}$. Hence, for any $r \geq r_0$ the diagonal elements of $(\mathbf{X}^{*r})^T \mathbf{X}^{*r}$ are equal to 1 and magnitude of the non diagonal elements are less than $\frac{1}{2n}$. Thus, applying Fact C.10, we get that

$$\lambda_{\min}\left((\mathbf{X}^{*r})^T \mathbf{X}^{*r}\right) \geq \frac{1}{2}.$$

Using Equation 18, we see that for $r = \Theta\left(\frac{\log 2n}{\delta}\right)$,

$$\bar{c}_r^2\left(\phi'\right) = \Theta\left(\left(\frac{\log 2n}{\delta}\right)^{-1.5}\right).$$

Thus,

$$\lambda_{\min}\left(\mathbf{G}^\infty\right) \geq \Omega\left(\left(\frac{\log 2n}{\delta}\right)^{-1.5}\right).$$

For computing $\lambda_{\min}\left(\mathbf{G}^{(0)}\right)$, we bound the absolute difference in each element of $\mathbf{G}^\infty$ and $\mathbf{G}^{(0)}$ by $\frac{1}{2n}\lambda_{\min}(\mathbf{G}^\infty)$ using Hoeffding's inequality (Fact C.3) and 1-Lipschitzness of $\phi$ and then apply a union bound over all the indices. The bound stated in the theorem follows from the fact that

$$\lambda_{\min}\left(\mathbf{G}^{(0)}\right) \geq \lambda_{\min}\left(\mathbf{G}^\infty\right) - \left\|\left(\mathbf{G}^\infty - \mathbf{G}^{(0)}\right)\right\|_F$$
$$\geq \lambda_{\min}\left(\mathbf{G}^\infty\right) - \lambda_{\min}\left(\mathbf{G}^\infty\right)/2.$$

$\square$

Using the bound of $\lambda_{\min}(\mathbf{G}^{\infty})$ from Theorem L.2 in Theorem L.1, we get the following explicit rate of convergence for 2 layer feedforward networks.

**Theorem L.3** (Thm 3.3 in Du et al. (2019a))**.** *Assume that the assumptions in Du et al. (2019a) hold true. Then, if we use gradient flow optimization and set the number of hidden nodes* $m = \Omega\left(\frac{n^6 \log^6(n) \log \frac{m}{\kappa}}{\delta^6 \kappa^3}\right)$, *with probability at least* $1 - \kappa$ *over the initialization we have*

$$\|\mathbf{u}_t - \mathbf{y}\|^2 \leq \epsilon$$

$\forall t \geq \Omega\left(\frac{\log^{1.5} n}{\delta^{1.5}} \log \frac{n}{\epsilon}\right)$, *for an arbitrarily small constant* $\epsilon > 0$.

## M    EXTENSIONS AND ADDITIONAL DISCUSSION

In this section, we discuss proofs and extensions of our theorems, using adaptations from related work.

### M.1    POLYNOMIAL MINIMUM EIGENVALUE OF $\mathbf{G}$-MATRIX AT TIME $t$

The upcoming lemma shows that, if we restrict the change in weight matrices, the minimum eigenvalue of $\mathbf{G}^{(t)}$ stays close to the minimum eigenvalue of $\mathbf{G}^{(0)}$.

**Lemma M.1.** *If activation function $\phi$ is $\alpha$-lipschitz and $\beta$-smooth and $\left\| \mathbf{w}_r^{(t)} - \mathbf{w}_r^{(0)} \right\| \leq \frac{\lambda_{\min}\left(\mathbf{G}^{(0)}\right)}{4\alpha\beta n}$, $\forall r \in [m]$, then*

$$\lambda_{\min}\left(\mathbf{G}^{(t)}\right) \geq \frac{1}{2}\lambda_{\min}\left(\mathbf{G}^{(0)}\right)$$

*Proof.* The claim follows a similar proof as the proof of lemma B.4 in Du et al. (2019b) and has been repeated in Lemma M.11. □

The restriction is ensured by the large number of neurons we can choose for our neural network, as we mention in the next lemma.

**Lemma M.2.** *Let $S_t \subseteq [n]$ denote a randomly picked batch of size $b$. Denote $\nabla^{(t)}$ as $\nabla_{\mathbf{W}^{(t)}} \mathcal{L}\left(\left\{(\mathbf{x}_i, y_i)\right\}_{i\in[n]} ; \mathbf{a}, \mathbf{W}^{(t)}\right)$. Let the activation function $\phi$ used be $\alpha$-lipschitz and $\beta$-smooth. The GD iterate at time $t + 1$ is given by,*

$$\mathbf{W}^{(t+1)} = \mathbf{W}^{(t)} - \eta\nabla^{(t)}$$

*Let $\eta \leq \mathcal{O}\left(\frac{\lambda_{\min}\left(\mathbf{G}^{(0)}\right)}{\beta n^4 \alpha^4}\right)$. If*

$$m \geq \Omega\left(\frac{n^4 \alpha^4 \beta^2 \log m}{\lambda_{\min}\left(\mathbf{G}^{(0)}\right)^4}\right)$$

*then,*

$$\left\| \mathbf{y} - \mathbf{u}^{(t)} \right\|^2 \leq \epsilon$$

*for $t \geq \Omega\left(\frac{\log\left(\frac{n}{\epsilon}\right)}{\eta\lambda_{\min}(\mathbf{G}^{(0)})}\right)$, with probability at-least $1 - m^{-3.5}$ w.r.t. $\left\{\mathbf{w}_k^{(0)}\right\}_{k=1}^m$ and $\left\{a_k^{(0)}\right\}_{k=1}^m$. Moreover,*

$$\left\| \mathbf{w}_k^{(t)} - \mathbf{w}_k^{(0)} \right\| \leq \frac{\lambda_{\min}\left(\mathbf{G}^{(0)}\right)}{4\alpha\beta n}$$

*holds true $\forall k \in [m]$ and $\forall t \geq 0$.*

*Proof.* The claim follows a similar proof as the proof of lemma A.1 in Du et al. (2019b), where we keep the output vector $\mathbf{a}$ non trainable in GD update. □

*Remark.* The above lemmas are applicable for activation functions in $\mathbf{J_r}$ for $r \geq 2$. Similar lemmas can be proved for $\mathbf{J_1}$, along the lines of the proof of Theorem 4.1 in Du et al. (2019a).

### M.2    TRAINABLE OUTPUT LAYER

Similar to the proof of Theorem 3.3 in Du et al. (2019b), we can show that the GD dynamics depends on sum of two matrices, i.e.

$$\frac{d\mathbf{u}^{(t)}}{dt} = (\mathbf{G}^{(t)} + \mathbf{H}^{(t)})(\mathbf{y} - \mathbf{u}^{(t)}),$$

where the definition of $\mathbf{G}$ stays the same and $\mathbf{H}$ is given by

$$h_{ij} = \frac{1}{m} \sum_{r \in [m]} \sigma(\mathbf{w}_r^T \mathbf{x}_i) \sigma(\mathbf{w}_r^T \mathbf{x}_j),$$

implying $\mathbf{H}$ is p.s.d. For the positive results, e.g. Theorem 4.1, observe that $\lambda_{\min}(\mathbf{G} + \mathbf{H}) \geq \lambda_{\min}(\mathbf{G})$, hence a bound on $\lambda_{\min}(\mathbf{G})$ suffices in this case. For the negative results, Theorem 4.3 and Theorem 4.4 can be restated as follows.

**Theorem M.3.** *Let the activation function $\phi$ be a degree-$p$ polynomial such that $\phi'(x) = \sum_{l=0}^{p-1} c_\ell x^\ell$ and let $d' = \dim\left(\operatorname{span}\{x_1 \ldots x_n\}\right) = \mathcal{O}\left(n^{\frac{1}{p}}\right)$. Then we have*

$$\lambda_k\left(\mathbf{G}^{(0)} + \mathbf{H}^{(0)}\right) = 0, \quad \forall k \geq \lceil 2n/d' \rceil$$

**Theorem M.4.** *Let the activation function be* $\tanh$ *and let $d' = \dim\left(\operatorname{span}\{x_1 \ldots x_n\}\right) = \mathcal{O}\left(\log^{0.75} n\right)$. Then we have*

$$\lambda_k\left(\mathbf{G}^{(0)} + \mathbf{H}^{(0)}\right) \leq \exp(-\Omega(n^{1/2d'})) \ll 1/\mathsf{poly}(n), \quad \forall k \geq \lceil 2n/d' \rceil$$

*with probability at least $1 - 1/n^{3.5}$ over the random choice of weight vectors $\left\{\mathbf{w}_k^{(0)}\right\}_{k=1}^m$ and $\left\{a_k^{(0)}\right\}_{k=1}^m$.*

*Proof.* (Proof sketch for Theorem M.3 and Theorem M.4) Following the proof of Theorem 4.3 and Theorem 4.4 gives us similar lower bounds for $\mathbf{H}^{(0)}$ i.e. $n(1 - \frac{1}{d'})$ lower order eigenvalues are 0, if $\phi$ is a degree-$p$ polynomial and exponentially small in $n^{1/d'}$ with high probability, if $\phi$ is tanh. Thus, we can use Weyl's inequality (Fact C.11) to show that $n(1 - \frac{2}{d'})$ lower order eigenvalues of $G^{(0)} + H^{(0)}$ are 0, if $\phi$ is a degree-$p$ polynomial and exponentially small in $n^{1/d'}$ with high probability, if $\phi$ is tanh. $\square$

*Remark.* The lemmas in subsection M.1, that were proved for a network with non trainable output vector $\mathbf{a}$, can also be proved for a network with trainable output vector $\mathbf{a}$. And so, a polynomial lower bound on minimum eigenvalue of $\mathbf{G}^{(0)}$ implies polynomial lower bound on minimum eigenvalue of $\mathbf{G}^{(t)}$, under appropriate number of neurons and GD training.

## M.3 MULTI CLASS OUTPUT WITH CROSS ENTROPY LOSS

Let's say, we have a classification task, where the number of classes is $C$ and we use the following neural network for prediction.

$$f_q(\mathbf{x}; \mathbf{A}, \mathbf{W}) := \frac{c_\phi}{\sqrt{m}} \sum_{k=1}^m a_{k,q} \phi\left(\mathbf{w}_k^T \mathbf{x}\right), \quad \forall q \in [C]$$

where $\mathbf{x} \in \mathbb{R}^d$ is the input and $\mathbf{W} = [\mathbf{w}_1, \ldots, \mathbf{w}_m] \in \mathbb{R}^{m \times d}$ is the hidden layer weight matrix and $\mathbf{A} \in \mathbb{R}^{m \times C}$ is the output layer weight matrix. We define $\mathbf{u}(\mathbf{x}) \in \mathbb{R}^C$ as

$$\mathbf{u}(\mathbf{x}) = \operatorname{softmax}\left(\mathbf{f}(\mathbf{x}; \mathbf{A}, \mathbf{W})\right)$$

where $\operatorname{softmax}$ on a vector $\mathbf{v} \in \mathbb{R}^C$ denotes the following operation

$$\operatorname{softmax}(\mathbf{v})_i = \frac{e^{v_i}}{\sum_{j \in [C]} e^{v_j}}.$$

Given a set of examples $\{\mathbf{x}_i, y_i\}_{i=1}^n$, where $\mathbf{x}_i \in \mathbb{R}^d$ and $y_i \in [C] \ \forall i \in [n]$, we use the following cross entropy loss to train the neural network.

$$\mathcal{L}\left(\mathbf{A}, \mathbf{W}; \{\mathbf{x}_i, y_i\}_{i=1}^n\right) = -\sum_{i=1}^n \log\left(f_{y_i}(\mathbf{x}; \mathbf{A}, \mathbf{W})\right)$$

Let's denote the vector $\tilde{\mathbf{y}}$ as an $nC$ dimensional vector, whose elements are defined as follows.

$$\tilde{\mathbf{y}}_{C(i-1)+j} = \begin{cases} 0, & \text{if } j \neq y_i \\ 1, & \text{otherwise} \end{cases}$$

Also, let's define another vector $\tilde{\mathbf{u}} \in \mathbb{R}^{nC}$ as follows.

$$\tilde{\mathbf{u}}_{C(i-1)+j} = \text{softmax}\left(\mathbf{f}(\mathbf{x}_i; \mathbf{A}, \mathbf{W})\right)_j$$

All the network dependent variables have a superscript $t$, depending on the time step at which they are calculated.

Using chain rule and derivative of cross entropy loss w.r.t. output of softmax layer, we can show the following differential equation for gradient flow.

$$\frac{d\tilde{u}}{dt} = \tilde{\mathbf{G}}^{(\mathbf{t})}\left(\tilde{y} - \tilde{\mathbf{u}}\right)$$

where $\tilde{\mathbf{G}} \in \mathbb{R}^{nC \times nC}$ is a gram matrix defined by its elements as follows.

$$\tilde{g}_{pr} = \frac{1}{m} \sum_{k \in [m]} a_{k,q} a_{k,q'} \phi'\left(\mathbf{w}_k^T \mathbf{x}_i\right) \phi'\left(\mathbf{w}_k^T \mathbf{x}_j\right),$$

where $i = \lfloor \frac{p}{C} \rfloor + 1, j = \lfloor \frac{r}{C} \rfloor + 1, q = p \mod C$ and $q' = r \mod C$. Thus,

$$\frac{d\left\|\tilde{\mathbf{y}} - \tilde{\mathbf{u}}^{(t)}\right\|^2}{dt} = -\left(\tilde{\mathbf{y}} - \tilde{\mathbf{u}}^{(t)}\right)^T \tilde{\mathbf{G}}^{(\mathbf{t})}\left(\tilde{\mathbf{y}} - \tilde{\mathbf{u}}^{(t)}\right) \leq -\lambda_{\min}\left(\tilde{\mathbf{G}}^{(t)}\right)\left\|\tilde{\mathbf{y}} - \tilde{\mathbf{u}}^{(t)}\right\|^2$$

Again following the argument discussed in section 3, if there hasn't been much movement in the weights of the network due to large number of neurons, $\left(\tilde{\mathbf{G}}^{(t)}\right)$ stays close to $\left(\tilde{\mathbf{G}}^{(0)}\right)$ and hence, the rate of convergence depends on the gram matrix $\left(\tilde{\mathbf{G}}^{(0)}\right)$. We show that the gram matrix possesses a unique structure and is related to the gram matrix defined for a single output network.

$\tilde{\mathbf{G}}$ contains $C$ disjoint principal $n \times n$ blocks, denoted by $\{\mathbf{B}^q\}_{q=1}^C$, where $\mathbf{B}_q$ is defined as follows:

$$b_{ij}^q = \frac{1}{m} \sum_{k \in [m]} a_{k,q}^2 \phi'\left(\mathbf{w}_k^T \mathbf{x}_i\right) \phi'\left(\mathbf{w}_k^T \mathbf{x}_j\right).$$

As can be seen, each $\mathbf{B}^q$ is structurally identical to the gram matrix defined for a single output neural network with input weight matrix $\mathbf{W}$ and output weight vector $\mathbf{a}_q$ (Equation 2).

Let's denote the set of $C(C-1)$ remaining disjoint non diagonal blocks of $\tilde{\mathbf{G}}$ as $\{\mathbf{B}^{q,q'}\}_{q,q' \in [C], q \neq q'}$, where each block is defined as follows.

$$b_{ij}^{q,q'} = \frac{1}{m} \sum_{k \in [m]} a_{k,q} a_{k,q'} \phi'\left(\mathbf{w}_k^T \mathbf{x}_i\right) \phi'\left(\mathbf{w}_k^T \mathbf{x}_j\right).$$

Assuming that we have sufficient number of neurons, $\tilde{\mathbf{G}}^{(\mathbf{0})}$ can be shown to be close to the matrix $\tilde{\mathbf{G}}^\infty$ using Hoeffding's inequality, where $\tilde{\mathbf{G}}^\infty$ has the following diagonal $\{(\mathbf{B}^\infty)^q\}_{q=1}^C$ and non diagonal blocks $\{(\mathbf{B}^\infty)^{q,q'}\}_{q=1}^C$ defined as follows.

$$(\mathbf{B}^\infty)_{ij}^q = \mathbb{E}_{\mathbf{w} \sim \mathcal{N}(0,\mathbf{I}), \tilde{a} \sim \mathcal{N}(0,1)} \tilde{a}^2 \phi'\left(\mathbf{w}^T \mathbf{x}_i\right) \phi'\left(\mathbf{w}^T \mathbf{x}_j\right),$$

$$(\mathbf{B}^\infty)_{ij}^{q,q'} = \mathbb{E}_{\mathbf{w} \sim \mathcal{N}(0,\mathbf{I}), a,\tilde{a} \sim \mathcal{N}(0,1)} a\tilde{a} \phi'\left(\mathbf{w}^T \mathbf{x}_i\right) \phi'\left(\mathbf{w}^T \mathbf{x}_j\right).$$

Using independence of random gaussian variables $a$ and $\tilde{a}$ and random gaussian vector $\mathbf{w}$, we can show that $(\mathbf{B}^\infty)^{q,q'}$ are identically zero matrices. Also, the diagonal blocks $(\mathbf{B}^\infty)^q$ are identically equal to the $\mathbf{G}^\infty$ matrix defined for single output layer neural networks (Equation 2). Thus, $\lambda_{\min}(\tilde{\mathbf{G}}^\infty) = \lambda_{\min}(\tilde{\mathbf{G}}^\infty)$ and hence the bounds for eigenvalues of $\lambda_{\min}(\tilde{\mathbf{G}}^{(0)})$ can be derived from the bounds for eigenvalues of $\lambda_{\min}(\mathbf{G}^{(\mathbf{0})})$, defined for a single output layer neural network (Equation 2).

## M.4 FINE-GRAINED ANALYSIS FOR SMOOTH ACTIVATION FUNCTIONS

In this section, we show the behavior of the loss function under gradient descent, in the low learning rate setting considered by Du et al. (2019a), Du et al. (2019b) and Arora et al. (2019c). We consider the neural network given by Equation 1.

We assume that the activation function $\phi$ satisfies the following properties.

- $\phi \in C^3$,
- $\phi$ is $\beta$-lipschitz and $\gamma$-smooth.

Now, we state some important theorems from Du et al. (2019b), that we will use for the future analysis. There are some differences in our setting and the setting of Du et al. (2019b). a) Du et al. (2019b) work with a general $L$ layer neural network. Hence, we state their theorems for $L = 1$. b) For simplicity of presentation, we have assumed that $a_k$ has been kept fixed during gradient descent, which can be easily removed as in subsection M.2 and Du et al. (2019a).

**Theorem M.5** (Lemma B.4 in Du et al. (2019b))**.** *Assume that $\forall i \in [n], |y_i| = \mathcal{O}(1)$ and*

$$
m \geq \Omega \left( \max \left\{ \frac{n^4}{\left(\lambda_{\min}\left(\mathbf{G}^{(0)}\right)\right)^4}, \frac{n}{\kappa}, \frac{n^2 \log \frac{n}{\kappa}}{\left(\lambda_{\min}\left(\mathbf{G}^{(0)}\right)\right)^2} \right\} \right),
$$

*If we set step size as*

$$
\eta = \mathcal{O}\left( \frac{\lambda_{\min}\left(\mathbf{G}^{(0)}\right)}{n^2} \right)
$$

*then with probability at least $1 - \kappa$ over $\left\{ \mathbf{w}_k^{(0)} \right\}_{k=1}^m$, the following holds $\forall t \in \mathbb{Z}^+$.*

$$
\mathcal{L}\left(\mathbf{W}^{(t)}\right) \leq \left( 1 - \frac{\eta \lambda_{\min}\left(\mathbf{G}^{(0)}\right)}{2} \right)^t \mathcal{L}\left(\mathbf{W}^{(0)}\right)
$$

Note that Du et al. (2019b) consider $\lambda_{\min}(\mathbf{G}^\infty)$ in their arguments. However, in the overparametrized regime, with high probability with respect to $\{\mathbf{w}_k\}_{k=1}^m$, $\lambda_{\min}(\mathbf{G}^\infty)$ and $\lambda_{\min}\left(\mathbf{G}^{(0)}\right)$ differ only by a constant factor, as given by lemma B.4 in Du et al. (2019b). Thus, we show their theorems using $\lambda_{\min}\left(G^{(0)}\right)$.

**Theorem M.6** (Lemma B.6 in Du et al. (2019b))**.** *Assuming the setting in Theorem M.5, the following holds $\forall t \in \mathbb{Z}^+$ and $\forall k \in [m]$.*

$$
\left\| \mathbf{w}_k^{(t)} - \mathbf{w}_k^{(0)} \right\| \leq \mathcal{O}\left( \frac{n}{\sqrt{m}\lambda_{\min}\left(\mathbf{G}^{(0)}\right)} \right).
$$

**Theorem M.7.** *Assuming the setting in Theorem M.5, the following holds $\forall t \in \mathbb{Z}^+$.*

$$
\left\| \mathbf{G}^{(t)} - \mathbf{G}^{(0)} \right\|_2 \leq \mathcal{O}\left( \frac{n^2}{\sqrt{m}\lambda_{\min}\left(\mathbf{G}^{(0)}\right)} \right).
$$

*Proof.* We follow the proof of lemma B.5 of Du et al. (2019b). We will bound the change in each element of the $G$-matrix and then, take a sum over all the elements to get a bound over the

perturbation.

$$
\left| g_{ij}^{(t)} - g_{ij}^{(0)} \right| = \left| \frac{\langle \mathbf{x}_i, \mathbf{x}_j \rangle}{m} \left\{ \sum_{k=1}^{m} \phi'\left(\mathbf{w}_k^{(t)T}\mathbf{x}_i\right) \phi'\left(\mathbf{w}_k^{(t)T}\mathbf{x}_j\right) - \sum_{k=1}^{m} \phi'\left(\mathbf{w}_k^{(0)T}\mathbf{x}_i\right) \phi'\left(\mathbf{w}_k^{(0)T}\mathbf{x}_j\right) \right\} \right|
$$

$$
\leq \frac{1}{m} \sum_{k=1}^{m} \left| \phi'\left(\mathbf{w}_k^{(t)T}\mathbf{x}_i\right) \phi'\left(\mathbf{w}_k^{(t)T}\mathbf{x}_j\right) - \phi'\left(\mathbf{w}_k^{(0)T}\mathbf{x}_i\right) \phi'\left(\mathbf{w}_k^{(t)T}\mathbf{x}_j\right) \right|
$$

$$
+ \frac{1}{m} \sum_{k=1}^{m} \left| \phi'\left(\mathbf{w}_k^{(0)T}\mathbf{x}_i\right) \phi'\left(\mathbf{w}_k^{(t)T}\mathbf{x}_j\right) - \phi'\left(\mathbf{w}_k^{(0)T}\mathbf{x}_i\right) \phi'\left(\mathbf{w}_k^{(0)T}\mathbf{x}_j\right) \right|
$$

$$
= \frac{1}{m} \sum_{k=1}^{m} \left| \phi'\left(\mathbf{w}_k^{(t)T}\mathbf{x}_i\right) - \phi'\left(\mathbf{w}_k^{(0)T}\mathbf{x}_i\right) \right| \ \left| \phi'\left(\mathbf{w}_k^{(t)T}\mathbf{x}_j\right) \right|
$$

$$
+ \frac{1}{m} \sum_{k=1}^{m} \left| \phi'\left(\mathbf{w}_k^{(t)T}\mathbf{x}_j\right) - \phi'\left(\mathbf{w}_k^{(0)T}\mathbf{x}_j\right) \right| \ \left| \phi'\left(\mathbf{w}_k^{(0)T}\mathbf{x}_i\right) \right|
$$

$$
\leq 2\gamma \mathcal{O}(1) \left\| \mathbf{w}_k^{(0)} - \mathbf{w}_k^{(t)} \right\| \leq \mathcal{O}\left( \frac{\gamma n}{\sqrt{m}\lambda_{\min}\left(\mathbf{G}^{(0)}\right)} \right)
$$

where, we use Theorem M.6 and the fact that $\phi'$ is $\gamma$-smooth and is bounded by $\mathcal{O}(1)$ in the final step. Thus, we get

$$
\left\| \mathbf{G}^{(t)} - \mathbf{G}^{(0)} \right\|_F \leq \sqrt{ \sum_{i,j \in [n]} \left( g_{ij}^{(t)} - g_{ij}^{(0)} \right)^2 } \leq \mathcal{O}\left( \frac{\gamma n^2}{\sqrt{m}\lambda_{\min}\left(\mathbf{G}^{(0)}\right)} \right)
$$

as required. □

**Lemma M.8** (Claim 3.4 in Du et al. (2019a)). *In the setting of Theorem M.5,*

$$
\left\| \mathbf{y} - \mathbf{u}^{(0)} \right\| \leq \mathcal{O}\left( \sqrt{\frac{n}{\kappa}} \right)
$$

*holds with probability at least $1 - \kappa$ with respect to $\left\{ \mathbf{w}_k^{(0)} \right\}_{k=1}^{m}$.*

Now, we state the following theorem, which is a simple adaptation of theorem 4.1 in Arora et al. (2019c). Let $v_1, v_2, ..., v_n$ denote the eigenvectors of $G^{(0)}$, corresponding to its eigenvalues $\lambda_1, \lambda_2, ..., \lambda_n$.

**Theorem M.9.** *With probability at least $1 - \kappa$ over the random initialization, $\forall t \in \mathbb{Z}^+$, the following holds*

$$
\left\| \mathbf{y} - \mathbf{u}^{(t)} \right\| = \sqrt{ \sum_{i=1}^{n} \left( 1 - \eta c_\phi^2 \lambda_i \right)^{2t} \left( \mathbf{v}_i^T \left( \mathbf{y} - \mathbf{u}^{(0)} \right) \right)^2 } \pm \epsilon,
$$

*provided*

$$
m \geq \Omega\left( \frac{n^5}{\gamma\kappa\lambda_{\min}\left(\mathbf{G}^{(0)}\right)^4 \epsilon^2} \right)
$$

*and*

$$
\eta \leq \mathcal{O}\left( \frac{\lambda_{\min}\left(\mathbf{G}^{(0)}\right)}{c_\phi^2 n^2} \right)
$$

*Proof.* For each $i \in [n]$, we get

$$
u_i^{(t+1)} - u_i^{(t)} = \frac{c_\phi}{\sqrt{m}} \sum_{k=1}^{m} a_k \left\{ \phi\left( \mathbf{w}_k^{(t+1)T} \mathbf{x}_i \right) - \phi\left( \mathbf{w}_k^{(t)T} \mathbf{x}_i \right) \right\}
$$

$$
= \frac{c_\phi}{\sqrt{m}} \sum_{k=1}^{m} a_k \left\{ \phi\left( \left( \mathbf{w}_k^{(t)} - \eta \frac{\partial}{\partial \mathbf{w}_k} \mathcal{L}\left( \mathbf{w}_k^{(t)} \right) \right)^T \mathbf{x}_i \right) - \phi\left( \mathbf{w}_k^{(t)T} \mathbf{x}_i \right) \right\} \quad (64)
$$

$$
= \frac{c_\phi}{\sqrt{m}} \sum_{k=1}^{m} a_k \phi'\left( \mathbf{w}_k^{(t)T} \mathbf{x}_i \right) \left( -\eta \frac{\partial}{\partial \mathbf{w}_k} \mathcal{L}\left( \mathbf{w}_k^{(t)} \right) \right)^T \mathbf{x}_i + \epsilon_i\left( t \right) \quad (65)
$$

$$
= -\frac{\eta c_\phi^2}{m} \sum_{k=1}^{m} a_k^2 \sum_{j=1}^{n} \left( \phi'\left( \mathbf{w}_k^{(t)T} \mathbf{x}_i \right) \phi'\left( \mathbf{w}_k^{(t)T} \mathbf{x}_j \right) \right) \left( u_j^{(t)} - y_j \right) \langle \mathbf{x}_i, \mathbf{x}_j \rangle + \epsilon_i\left( t \right)
$$

$$
= -\eta c_\phi^2 \sum_{j=1}^{n} \mathbf{G}_{ij}^{(t)} \left( u_j^{(t)} - y_j \right) + \epsilon_i\left( t \right) \quad (66)
$$

where we use Taylor expansion of $\phi$ in Equation 65. $\epsilon_i\left( t \right)$ denotes the error term due to truncated Taylor expansion, whose norm can be bounded by

$$
\left| \epsilon_i\left( t \right) \right| \le \frac{c_\phi}{2\sqrt{m}} \sum_{k=1}^{m} \mathcal{O}(1) \left( \left( \eta \frac{\partial}{\partial \mathbf{w}_k} \mathcal{L}\left( \mathbf{w}_k^{(t)} \right) \right)^T \mathbf{x}_i \right)^2 \quad (67)
$$

$$
= \frac{c_\phi}{2m^{3/2}} \eta^2 \mathcal{O}(1) \sum_{k=1}^{m} \left( \sum_{j=1}^{n} a_k \phi'\left( \mathbf{w}_k^{(t)T} \mathbf{x}_i \right) \langle \mathbf{x}_i, \mathbf{x}_j \rangle \left( \mathbf{y} - \mathbf{u}^{(t)} \right)_j \right)^2 \quad 
$$

$$
\le \frac{c_\phi}{2m^{3/2}} \eta^2 \mathcal{O}(1) \sum_{k=1}^{m} \left( \sum_{j=1}^{n} a_k \phi'\left( \mathbf{w}_k^{(t)T} \mathbf{x}_i \right)^2 \langle \mathbf{x}_i, \mathbf{x}_j \rangle^2 \right) \left\| \mathbf{y} - \mathbf{u}^{(t)} \right\|^2 \quad (68)
$$

$$
\le \mathcal{O}\left( \frac{\eta^2 c_\phi n}{\sqrt{m/\log m}} \right) \left\| \mathbf{y} - \mathbf{u}^{(t)} \right\|^2 \quad (69)
$$

where we use the fact that $\left| \phi''(z) \right| \le \mathcal{O}(1) \forall z \in \mathbb{R}$ in Equation 67, use the Cauchy-Schwartz inequality in Equation 68 and use the fact that $\left| \phi'(z) \right| \le \mathcal{O}(1) \forall z \in \mathbb{R}$, $\langle \mathbf{x}_i, \mathbf{x}_j \rangle \le 1 \quad \forall i, j \in [n]$ and $\left| a_k \right| \le \sqrt{\log m}$ with high probability in Equation 69. Thus, this gives

$$
\mathbf{u}^{(t+1)} - \mathbf{u}^{(t)} = -\eta c_\phi^2 \mathbf{G}^{(t)} \left( \mathbf{u}^{(t)} - \mathbf{y} \right) + \epsilon(t)
$$

where

$$
\left\| \epsilon(t) \right\| = \sqrt{\sum_{i=1}^{n} \epsilon_i(t)^2} \le \mathcal{O}\left( \frac{\eta^2 c_\phi n^{3/2}}{\sqrt{m/\log m}} \right) \left\| \mathbf{y} - \mathbf{u}^{(t)} \right\|^2.
$$

Now, since $\mathbf{G}^{(t)}$ is close to $\mathbf{G}^{(0)}$, we can write

$$
\mathbf{u}^{(t+1)} - \mathbf{u}^{(t)} = -\eta c_\phi^2 \mathbf{G}^{(0)} \left( \mathbf{u}^{(t)} - \mathbf{y} \right) + \tau(t) \quad (70)
$$

where $\tau(t) = -\eta c_\phi^2 \left( \mathbf{G}^{(t)} - \mathbf{G}^{(0)} \right) \left( \mathbf{u}^{(t)} - \mathbf{y} \right) + \epsilon(t)$. The norm of $\tau(t)$ can be bounded as follows.

$$
\left\| \tau(t) \right\| \le \eta c_\phi^2 \left\| \left( \mathbf{G}^{(t)} - \mathbf{G}^{(0)} \right) \left( \mathbf{u}^{(t)} - \mathbf{y} \right) \right\| + \left\| \epsilon(t) \right\|
$$

$$
\le \eta c_\phi^2 \left\| \mathbf{G}^{(t)} - \mathbf{G}^{(0)} \right\| \left\| \mathbf{u}^{(t)} - \mathbf{y} \right\| + \left\| \epsilon(t) \right\|
$$

$$
\le \mathcal{O}\left( \eta \gamma \frac{n^2 c_\phi^2}{\sqrt{m} \lambda_{\min}\left( \mathbf{G}^{(0)} \right)} \right) \left\| \mathbf{y} - \mathbf{u}^{(t)} \right\| + \mathcal{O}\left( \frac{\eta^2 c_\phi n^{3/2}}{\sqrt{m/\log m}} \right) \left\| \mathbf{y} - \mathbf{u}^{(t)} \right\|^2
$$

$$
\le \mathcal{O}\left( \frac{\eta \gamma n^2 c_\phi^2}{\sqrt{m} \lambda_{\min}\left( \mathbf{G}^{(0)} \right)} \right) \left\| \mathbf{y} - \mathbf{u}^{(t)} \right\|.
$$

Thus, applying Equation 70 recursively, we get

$$\mathbf{u}^{(t)} - \mathbf{y} = \left(\mathbf{I} - \eta c_\phi^2 \mathbf{G}^{(0)}\right)^t \left(\mathbf{u}^{(0)} - \mathbf{y}\right) + \sum_{t'=0}^{t-1} \left(\mathbf{I} - \eta c_\phi^2 \mathbf{G}^{(0)}\right)^{t'} \tau(t-1-t'). \qquad (71)$$

We bound the norm of each term in the above equation. The norm of the first term can be given as follows.

$$\left\| - \left(\mathbf{I} - \eta c_\phi^2 \mathbf{G}^{(0)}\right)^t \left(\mathbf{u}^{(0)} - \mathbf{y}\right)\right\| = \left\| \left(\sum_{i=1}^n \left(1 - \eta c_\phi^2 \lambda_i\right)^t \mathbf{v}_i \mathbf{v}_i^T\right) \left(\mathbf{u}^{(0)} - \mathbf{y}\right)\right\|$$

$$= \sqrt{\sum_{i=1}^n \left(1 - \eta c_\phi^2 \lambda_i\right)^{2t} \left(\mathbf{v}_i^T \left(\mathbf{u}^{(0)} - \mathbf{y}\right)\right)^2}. \qquad (72)$$

Now, the norm of the second term can be bounded as

$$\| \sum_{t'=0}^{t-1} \left(\mathbf{I} - \eta c_\phi^2 \mathbf{G}^{(0)}\right)^{t'} \tau(t-1-t')\|_2$$

$$\leq \sum_{t'=0}^{t-1} \left\|\mathbf{I} - \eta c_\phi^2 \mathbf{G}^{(0)}\right\|_2^t \|\tau(t-1-t')\|_2$$

$$\leq \sum_{t'=0}^{t-1} \left(1 - \eta c_\phi^2 \lambda_{\min}\left(\mathbf{G}^{(0)}\right)\right)^{t'} \mathcal{O}\left(\frac{\eta \gamma n^2 c_\phi^2}{\sqrt{m}\lambda_{\min}\left(\mathbf{G}^{(0)}\right)}\right) \left\|\mathbf{u}^{(t-1-t')} - \mathbf{y}\right\|_2 \qquad (73)$$

$$\leq \sum_{t'=0}^{t-1} \left(1 - \eta c_\phi^2 \lambda_{\min}\left(\mathbf{G}^{(0)}\right)\right)^{t'} \mathcal{O}\left(\frac{\eta \gamma n^2 c_\phi^2}{\sqrt{m}\lambda_{\min}\left(\mathbf{G}^{(0)}\right)}\right) \left(1 - \frac{\eta c_\phi^2 \lambda_{\min}\left(\mathbf{G}^{(0)}\right)}{4}\right)^{t-1-t'} \mathcal{O}\left(\sqrt{n/\kappa}\right)$$

$$\leq t \left(1 - \frac{\eta c_\phi^2 \lambda_{\min}\left(\mathbf{G}^{(0)}\right)}{4}\right)^{t-1} \mathcal{O}\left(\frac{\gamma \eta n^{5/2} c_\phi^2}{\sqrt{m\kappa}\lambda_{\min}\left(\mathbf{G}^{(0)}\right)}\right).$$

In Equation 73, we use the following.

$$\left\|\mathbf{u}^{(s)} - \mathbf{y}\right\|_2 = \left(1 - \frac{\eta c_\phi^2 \lambda_{\min}\left(\mathbf{G}^{(0)}\right)}{4}\right)^s \left\|\mathbf{u}^{(0)} - \mathbf{y}\right\| \leq \left(1 - \frac{\eta c_\phi^2 \lambda_{\min}\left(\mathbf{G}^{(0)}\right)}{4}\right)^s \mathcal{O}\left(\sqrt{n/\kappa}\right)$$

Thus, combining the two terms, we have

$$\left\|\mathbf{u}^{(t)} - \mathbf{y}\right\| \leq \sqrt{\sum_{i=1}^n \left(1 - \eta c_\phi^2 \lambda_i\right)^{2t} \left(\mathbf{v}_i^T \left(\mathbf{y} - \mathbf{u}^{(0)}\right)\right)^2} \pm t \left(1 - \frac{\eta c_\phi^2 \lambda_{\min}\left(\mathbf{G}^{(0)}\right)}{4}\right)^{t-1} \mathcal{O}\left(\frac{\gamma \eta n^{5/2} c_\phi^2}{\sqrt{m\kappa}\lambda_{\min}\left(\mathbf{G}^{(0)}\right)}\right)$$

$$\qquad (74)$$

$$\leq \sqrt{\sum_{i=1}^n \left(1 - \eta c_\phi^2 \lambda_i\right)^{2t} \left(\mathbf{v}_i^T \left(\mathbf{y} - \mathbf{u}^{(0)}\right)\right)^2} \pm \underbrace{\mathcal{O}\left(\frac{\gamma n^{5/2}}{\sqrt{\kappa m}\lambda_{\min}^2\left(\mathbf{G}^{(0)}\right)}\right)}_{\spadesuit} \qquad (75)$$

where in Equation 74, we use the fact that

$$t \left(1 - \frac{\eta c_\phi^2 \lambda_{\min}\left(\mathbf{G}^{(0)}\right)}{4}\right)^{t-1} \leq \frac{4}{\eta c_\phi^2 \lambda_{\min}\left(\mathbf{G}^{(0)}\right)}.$$

Thus, for the term denoted by ♠ to be less than $\epsilon$, we need

$$m \geq \Omega \left( \frac{\gamma n^5}{\kappa \lambda_{\min} \left( \mathbf{G}^{(0)} \right)^4 \epsilon^2} \right)$$ □

## M.5 PROOF FOR SGD

The following theorem is an adaptation of Theorem 2 in Allen-Zhu et al. (2019), which asserts fast convergence of SGD for ReLU. The theorem below applies to activation in $\mathbf{J_r}$ for $r \geq 2$; the case of $\mathbf{J_1}$ can be handled by another adaptation of Theorem 2 in Allen-Zhu et al. (2019) which we do not discuss. Du et al. (2019b) analyzed gradient descent for this setting and mentioned the analysis of SGD as a future work.

**Theorem M.10.** *Let $S_t \subseteq [n]$ denote a randomly picked batch of size $b$. Let $\nabla^{(t)}$ denote $\frac{n}{b} \nabla_{\mathbf{W}^{(t)}} \mathcal{L} \left( \{(\mathbf{x}_i, y_i)\}_{i \in S_t} ; \mathbf{a}, \mathbf{W}^{(t)} \right)$. Let the activation function $\phi$ be $\alpha$-Lipschitz and $\beta$-smooth. The SGD iterate at time $t+1$ is given by,*

$$\mathbf{W}^{(t+1)} = \mathbf{W}^{(t)} - \eta \nabla^{(t)}$$

*Let $\eta \leq \mathcal{O} \left( \lambda_{\min} \left( \mathbf{G}^{(0)} \right) \frac{b^2}{\beta n^6 \alpha^4} \right)$. If*

$$m \geq \Omega \left( \frac{n^6 \alpha^4 \beta^2 \log m}{b^2 \lambda_{\min} \left( \mathbf{G}^{(0)} \right)^4} \right)$$

*then,*

$$\left\| \mathbf{y} - \mathbf{u}^{(t)} \right\|^2 \leq \epsilon$$

*for $t \geq \Omega \left( \frac{n^6 \alpha^4}{b^2 \lambda_{\min} \left( \mathbf{G}^{(0)} \right)^2} \beta \log \left( \frac{n}{\epsilon} \right) \right)$, with probability at least $1 - e^{-\Omega\left(n^2\right)} - m^{-3.5}$ w.r.t. random choice of $S_t$ for $t \geq 0$.*

*Proof.* Note that

$$\mathbb{E}_{S_t} \nabla^{(t)} = \nabla_{\mathbf{W}^{(t)}} \mathcal{L} \left( \{(\mathbf{x}_i, y_i)\}_{i \in [n]} ; \mathbf{a}, \mathbf{W}^{(t)} \right)$$

Using taylor expansion for each coordinate $i$, we have

$$
\begin{aligned}
& u_i^{(t+1)} - u_i^{(t)} \\
=& u_i \left( \mathbf{W}^{(t)} - \eta \nabla^{(t)} \right) - u_i(\mathbf{W}^{(t)}) \\
=& -\int_{s=0}^{\eta} \left\langle \nabla^{(t)}, u_i' \left( \mathbf{W}^{(t)} - s \nabla^{(t)} \right) \right\rangle ds \\
=& -\int_{s=0}^{\eta} \left\langle \nabla^{(t)}, u_i'(\mathbf{W}^{(t)}) \right\rangle ds + \int_{s=0}^{\eta} \left\langle \nabla^{(t)}, u_i'(\mathbf{W}^{(t)}) - u_i' \left( \mathbf{W}^{(t)} - s \nabla^{(t)} \right) \right\rangle ds \\
\triangleq& I_1^i(t) + I_2^i(t)
\end{aligned}
\tag{76}
$$

Writing the decrease of loss at time $t$, we have

$$
\begin{aligned}
\left\| \mathbf{y} - \mathbf{u}^{(t+1)} \right\|_2^2 =& \left\| \mathbf{y} - \mathbf{u}^{(t)} - (\mathbf{u}^{(t+1)} - \mathbf{u}^{(t)}) \right\|_2^2 \\
=& \left\| \mathbf{y} - \mathbf{u}^{(t)} \right\|_2^2 - 2(\mathbf{y} - \mathbf{u}^{(t)})^\top (\mathbf{u}^{(t+1)} - \mathbf{u}^{(t)}) + \left\| \mathbf{u}^{(t+1)} - \mathbf{u}^{(t)} \right\|_2^2 \\
=& \left\| \mathbf{y} - \mathbf{u}^{(t)} \right\|_2^2 - 2(\mathbf{y} - \mathbf{u}^{(t)})^\top \mathbf{I}_1 - 2(\mathbf{y} - \mathbf{u}^{(t)})^\top \mathbf{I}_2 + \left\| \mathbf{u}^{(t+1)} - \mathbf{u}^{(t)} \right\|_2^2
\end{aligned}
\tag{77}
$$

$$\tag{78}$$

where $\mathbf{I}_1 \in \mathbb{R}^n$ and its $i^{\text{th}}$ coordinate is given by $I_1^i$. Similarly, we define $\mathbf{I}_2$.

$I_1^i$ is given as,

$$
\begin{aligned}
I_1^i &= -\eta \left\langle \nabla^{(t)}, u_i'(\mathbf{W}^{(t)}) \right\rangle \\
&= -\eta \frac{n}{b} \sum_{j \in S_t} \left( u_j^{(t)} - y_j \right) \left\langle u_j'(\mathbf{W}^{(t)}), u_i'(\mathbf{W}^{(t)}) \right\rangle \qquad (79) \\
&\triangleq -\eta \frac{n}{b} \sum_{j \in S_t} \left( u_j^{(t)} - y_j \right) g_{ij}^{(t)} \qquad (80)
\end{aligned}
$$

where we use the definition of $\nabla^{(t)}$ in Equation 79.

That implies,

$$
\begin{aligned}
\|\mathbf{I_1}\| &= \eta \frac{n}{b} \left\| \mathbf{G}^{(t)} \mathbf{D}^{(t)} \left( \mathbf{u}^{(t)} - \mathbf{y} \right) \right\| \\
&\leq \eta \frac{n}{b} \left\| \mathbf{G}^{(t)} \right\|_2 \left\| \mathbf{y} - \mathbf{u}^{(t)} \right\| \\
&\leq \eta \frac{n}{b} n \alpha^2 \left\| \mathbf{y} - \mathbf{u}^{(t)} \right\| \qquad (81)
\end{aligned}
$$

where $\mathbf{D}^{(t)} \in \mathbb{R}^{n \times n}$ denotes a diagonal matrix that has 1 in $i^{\text{th}}$ diagonal element, iff $i \in S_t$ and 0 otherwise and $\|\mathbf{G}\|_2 \leq \|\mathbf{G}\|_F \leq nL^2$, since $\phi$ is $\alpha$-lipschitz.

Note that,

$$
\begin{aligned}
\mathbb{E}_{S_t} (\mathbf{y} - \mathbf{u}^{(t)})^\top I_1 &= \mathbb{E}_{S_t} \sum_{i \in [n]} \sum_{j \in S_t} \eta \frac{n}{b} \left( u_i^{(t)} - y_i \right) g_{ij}^{(t)} \left( u_j^{(t)} - y_j \right) \\
&= \eta \left( \mathbf{y} - \mathbf{u}^{(t)} \right)^\top \mathbf{G}^{(t)} \left( \mathbf{y} - \mathbf{u}^{(t)} \right) \\
&\geq \eta \lambda_{\min} \left( \mathbf{G}^{(t)} \right) \left\| \mathbf{y} - \mathbf{u}^{(t)} \right\|^2 \qquad (82)
\end{aligned}
$$

Also, we can bound $\mathbf{I}_2$ in the following manner.

$$
\left| I_2^i(t) \right| \leq \eta \left\| \nabla^{(t)} \right\|_F \max_{0 \leq s \leq \eta} \left\| u_i'(\mathbf{W}^{(t)}) - u_i' \left( \mathbf{W}^{(t)} - s \nabla^{(t)} \right) \right\|_F
$$

Since,

$$
\left\| \nabla^{(t)} \right\|_F^2 = \left( \frac{n/b}{\sqrt{m}} \right)^2 \sum_{r=1}^m \left\| \sum_{i \in S_t} \left( y_i - u_i^{(t)} \right) a_r \phi' \left( \mathbf{w}_r^T \mathbf{x}_i \right) \mathbf{x}_i \right\|^2 \leq \left( n^{1.5} \frac{\alpha}{b} \left\| \mathbf{y} - \mathbf{u}^{(t)} \right\| \right)^2
$$

and

$$
\begin{aligned}
\left\| u_i'(\mathbf{W}^{(t)}) - u_i' \left( \mathbf{W}^{(t)} - s \nabla^{(t)} \right) \right\|_F &= \frac{1}{\sqrt{m}} \sqrt{ \sum_{r=1}^m \left\| a_r \left( \phi' \left( \mathbf{w}_r^T \mathbf{x}_i \right) - \phi' \left( \mathbf{w}_r^T x_i - s \nabla_r^{(t)T} \mathbf{x}_i \right) \right) \mathbf{x}_i \right\|_2^2 } \\
&\leq \frac{1}{\sqrt{m}} \sqrt{ \sum_{r=1}^m \left\| a_r \beta \left( s \nabla_r^{(t)T} \mathbf{x}_i \right) \mathbf{x}_i \right\|_2^2 } \\
&\leq \frac{1}{\sqrt{m}} \beta \eta \sqrt{ \max_{r \in [m]} \left\| \nabla_r^{(t)} \right\|_2^2 \sum_{r=1}^m a_r^2 } \\
&\leq \beta \eta n^{1.5} \frac{\alpha}{b} \left\| \mathbf{y} - \mathbf{u}^{(t)} \right\| \qquad (83)
\end{aligned}
$$

where we use $\beta$-smoothness of the activation function $\phi$ in the first step, $\sum_r a_r^2 \le 5m$ with high probability and $\max_{r \in [m]} \left\| \nabla_r^{(t)} \right\|_2^2 \le \left\| \nabla_r^{(t)} \right\|_F^2$ in 3$^{\text{rd}}$ step.

That implies,

$$\left| I_2^i(t) \right| \le \beta \eta^2 n^3 \frac{\alpha^2}{b^2} \left\| \mathbf{y} - \mathbf{u}^{(t)} \right\|^2 \le \beta \eta^2 n^4 \frac{\alpha^2}{b^2} \left\| \mathbf{y} - \mathbf{u}^{(t)} \right\| \tag{84}$$

Also,

$$\left| u_i(\mathbf{W}^{(t)}) - u_i\left( \mathbf{W}^{(t)} - s\nabla^{(t)} \right) \right|^2 = \frac{1}{m} \left| \sum_{r=1}^m a_r \left( \phi\left( \mathbf{w}_r^T \mathbf{x}_i \right) - \phi\left( \mathbf{w}_r^T \mathbf{x}_i - s\nabla_r^{(t)T} \mathbf{x}_i \right) \right) \right|^2$$

$$\le \frac{1}{m} \left( \sum_{r=1}^m \left| a_r \alpha\left( s\nabla_r^{(t)T} \mathbf{x}_i \right) \right| \right)^2$$

$$\le \alpha^2 \eta^2 \left( \frac{1}{m} \sum_{r \in [m]} a_r^2 \right) \max_{r \in [m]} \left\| \nabla_r^{(t)} \right\|_2^2$$

$$\le \alpha^2 \eta^2 \max_{r \in [m]} \left\| \nabla_r^{(t)} \right\|_2^2$$

$$\le \alpha^2 \eta^2 \left( n^{1.5} \frac{\alpha}{b} \left\| \mathbf{y} - \mathbf{u}^{(t)} \right\| \right)^2 \tag{85}$$

Hence, using Equation 77, Equation 81, Equation 84 and Equation 85, we get

$$\left\| \mathbf{y} - \mathbf{u}^{(t+1)} \right\|_2^2 = \left\| \mathbf{y} - \mathbf{u}^{(t)} \right\|_2^2 - 2(y - u^{(t)})^\top I_1 - 2(\mathbf{y} - \mathbf{u}^{(t)})^\top I_2 + \left\| \mathbf{u}^{(t+1)} - \mathbf{u}^{(t)} \right\|_2^2 \tag{86}$$

$$= \left( 1 + 2\eta \frac{n^2 L^2}{b} + 2\beta \eta^2 n^{4.5} \frac{\alpha^2}{b^2} + \frac{\alpha^4}{b^2} \eta^2 n^4 \right) \left\| \mathbf{y} - \mathbf{u}^{(t)} \right\|^2 \tag{87}$$

$$= \mathcal{O}\left( \left( 1 + 2\eta \frac{n^2 \alpha^2}{b} \right) \left\| \mathbf{y} - \mathbf{u}^{(t)} \right\|^2 \right) \tag{88}$$

Taking log both the sides, we get

$$\log\left( \left\| \mathbf{y} - \mathbf{u}^{(t+1)} \right\|_2^2 \right) \le \mathcal{O}\left( \eta \frac{n^2 \alpha^2}{b} \right) + \log\left( \left\| \mathbf{y} - \mathbf{u}^{(t)} \right\|^2 \right)$$

By azuma-hoeffding inequality, we have

$$\log\left( \left\| \mathbf{y} - \mathbf{u}^{(t)} \right\|_2^2 \right) - E_{S_{t-1}} \log\left( \left\| \mathbf{y} - \mathbf{u}^{(t)} \right\|_2^2 \right) \le \sqrt{t} \mathcal{O}\left( \eta \frac{n^2 \alpha^2}{b} \right) n \tag{89}$$

with probability at-least $1 - e^{-\Omega(n^2)}$.

Also,

$$E_{S_t} \left\| \mathbf{y} - \mathbf{u}^{(t+1)} \right\|_2^2 = \left\| \mathbf{y} - \mathbf{u}^{(t)} \right\|_2^2 - E_{S_t} 2(\mathbf{y} - \mathbf{u}^{(t)})^\top \mathbf{I}_1 - E_{S_t} 2(\mathbf{y} - \mathbf{u}^{(t)})^\top \mathbf{I}_2 + E_{S_t} \left\| \mathbf{u}^{(t+1)} - \mathbf{u}^{(t)} \right\|_2^2$$

$$\le \left( 1 - \eta \lambda_{\min}\left( \mathbf{G}^{(t)} \right) + 2\beta \eta^2 n^{4.5} \frac{\alpha^2}{b^2} + \frac{\alpha^4}{b^2} \eta^2 n^4 \right) \left\| \mathbf{y} - \mathbf{u}^{(t)} \right\|^2 \tag{90}$$

$$\le \left( 1 - \frac{1}{2} \eta \lambda_{\min}\left( \mathbf{G}^{(0)} \right) + 2\beta \eta^2 n^{4.5} \frac{\alpha^2}{b^2} + \frac{\alpha^4}{b^2} \eta^2 n^4 \right) \left\| \mathbf{y} - \mathbf{u}^{(t)} \right\|^2$$

$$\le \left( 1 - \frac{1}{4} \eta \lambda_{\min}\left( \mathbf{G}^{(0)} \right) \right) \left\| \mathbf{y} - \mathbf{u}^{(t)} \right\|^2 \tag{91}$$

where we use Equation 82, Equation 84 and Equation 85 in Equation 90.

Taking log both the sides, we get

$$\log\left(E_{S_t}\left\|\mathbf{y}-\mathbf{u}^{(t+1)}\right\|_2^2\right) \leq \log\left(\left\|\mathbf{y}-\mathbf{u}^{(t)}\right\|^2\right) - \frac{1}{4}\eta\lambda_{\min}\left(\mathbf{G}^{(0)}\right)$$

Using Jensen's inequality, we get

$$E_{S_t}\log\left(\left\|\mathbf{y}-\mathbf{u}^{(t+1)}\right\|_2^2\right) \leq \log\left(\left\|\mathbf{y}-\mathbf{u}^{(t)}\right\|^2\right) - \frac{1}{4}\eta\lambda_{\min}\left(\mathbf{G}^{(0)}\right) \tag{92}$$

Thus, for $t \geq 0$, using Equation 89 and Equation 92, we get

$$\log\left(\left\|\mathbf{y}-\mathbf{u}^{(t)}\right\|_2^2\right) \leq \sqrt{t}\mathcal{O}\left(\eta\frac{n^2\alpha^2}{b}\right)n + \log\left(\left\|\mathbf{y}-\mathbf{u}^{(0)}\right\|_2^2\right) - \Omega\left(\frac{1}{4}\eta\lambda_{\min}\left(\mathbf{G}^{(0)}\right)\right)t$$

$$\leq \log\left(\left\|\mathbf{y}-\mathbf{u}^{(0)}\right\|_2^2\right) - \left(\sqrt{\eta\lambda_{\min}\left(\mathbf{G}^{(0)}\right)}\Omega\left(\sqrt{t}\right) - \mathcal{O}\left(\sqrt{\frac{\eta}{\lambda_{\min}\left(\mathbf{G}^{(0)}\right)}}\frac{n^3\alpha^2}{b}\right)\right)^2$$

$$+ \mathcal{O}\left(\sqrt{\frac{\eta}{\lambda_{\min}\left(\mathbf{G}^{(0)}\right)}}\frac{n^3\alpha^2}{b}\right)^2$$

$$\leq \log\left(\left\|\mathbf{y}-\mathbf{u}^{(0)}\right\|_2^2\right) - \left(\sqrt{\eta\lambda_{\min}\left(\mathbf{G}^{(0)}\right)}\Omega\left(\sqrt{t}\right) - \mathcal{O}\left(\sqrt{\frac{\eta}{\lambda_{\min}\left(\mathbf{G}^{(0)}\right)}}\frac{n^3\alpha^2}{b}\right)\right)^2 + 1$$

$$\leq \log\left(\left\|\mathbf{y}-\mathbf{u}^{(0)}\right\|_2^2\right) - \mathcal{I}\left[t \geq \frac{n^6\alpha^4}{b^2\lambda_{\min}\left(\mathbf{G}^{(0)}\right)^2}\right]\Omega\left(\eta\lambda_{\min}\left(\mathbf{G}^{(0)}\right)\right)t + 1$$

$$\leq \log\left(\left\|\mathbf{y}-\mathbf{u}^{(0)}\right\|_2^2\right) - \mathcal{I}\left[t \geq \frac{n^6\alpha^4}{b^2\lambda_{\min}\left(\mathbf{G}^{(0)}\right)^2}\right]\frac{b^2\lambda_{\min}\left(\mathbf{G}^{(0)}\right)^2}{\beta n^6\alpha^4}t + 1 \tag{93}$$

Hence, if $t \geq \Omega\left(\frac{n^6\alpha^4}{b^2\lambda_{\min}\left(\mathbf{G}^{(0)}\right)^2}\beta\log\left(\frac{n}{\epsilon}\right)\right)$, we have

$$\log\left(\left\|\mathbf{y}-\mathbf{u}^{(t)}\right\|_2^2\right) \leq \log\left(\mathcal{O}\left(n\right)\right) - \Omega\left(\log\left(\frac{n}{\epsilon}\right)\right) \leq \log\left(\epsilon\right)$$

implying $\left\|\mathbf{y}-\mathbf{u}^{(t)}\right\|_2^2 \leq \epsilon$. Also, let $T_0 = \frac{n^6\alpha^4}{b^2\lambda_{\min}\left(\mathbf{G}^{(0)}\right)^2}$. Then, applying Equation 93 in chunks of steps $T_0$, we get

$$\sum_{t=0}^{\infty}\left\|\mathbf{y}-\mathbf{u}^{(t)}\right\| \leq 2T_0\mathcal{O}\left(\sqrt{n}\right) + 2T_0\frac{\mathcal{O}\left(\sqrt{n}\right)}{2} + \frac{\mathcal{O}\left(\sqrt{n}\right)}{4} + ... = \mathcal{O}\left(\sqrt{n}T_0\right) \tag{94}$$

which implies

$$\sum_{t=0}^{\infty}\left\|\mathbf{w}_r^{(t+1)} - \mathbf{w}_r^{(t)}\right\| = \sum_{t=0}^{\infty}\eta\left\|\nabla_r^{(t)}\right\| = \sum_{t=0}^{\infty}\eta\left\|\sum_{i\in S_t}a_r\phi'\left(\mathbf{w}_r^T x_i\right)x_i\right\|$$

$$\leq \frac{\alpha n^{1.5}\eta\sqrt{\log m}}{b\sqrt{m}}\sum_{t=0}^{\infty}\left\|\mathbf{y}-\mathbf{u}^{(t)}\right\| \tag{95}$$

$$= \frac{\alpha n^2\eta\sqrt{\log m}}{b\sqrt{m}}T_0 \tag{96}$$

where in Equation 95 we use the fact that maximum magnitude of $a_r$ is $\sqrt{\log m}$ with high probability.

Also, for $\left\| \mathbf{G}^{(t)} - \mathbf{G}^{(0)} \right\|$ to be less than $\frac{1}{2}\lambda_{\min}\left(\mathbf{G}^{(0)}\right)$, we need to have (Lemma M.11)

$$\left\| \mathbf{w}_r^{(t)} - \mathbf{w}_r^{(0)} \right\| \leq \frac{\lambda_{\min}\left(\mathbf{G}^{(0)}\right)}{4\alpha\beta n}$$

Thus, for both the conditions to hold true, we must have

$$m \geq \Omega\left(\frac{n^6 \alpha^4 \beta^2 \log m}{b^2 \lambda_{\min}\left(\mathbf{G}^{(0)}\right)^4}\right) \tag{97}$$

$\square$

**Lemma M.11.** *If activation function $\phi$ is $\alpha$-lipschitz and $\beta$-smooth and $\left\| \mathbf{w}_r^{(t)} - \mathbf{w}_r^{(0)} \right\| \leq \frac{\lambda_{\min}\left(\mathbf{G}^{(0)}\right)}{4\alpha\beta n}, \forall r \in [m]$, then*

$$\lambda_{\min}\left(\mathbf{G}^{(t)}\right) \geq \frac{1}{2}\lambda_{\min}\left(\mathbf{G}^{(0)}\right)$$

*Proof.*

$$
\begin{aligned}
\left| g_{ij}^{(t)} - g_{ij}^{(0)} \right| &= \frac{1}{m}\left| \sum_{r=1}^m a_r^2 \phi'\left(\mathbf{w}_r^{(t)T}\mathbf{x}_i\right)\phi'\left(\mathbf{w}_r^{(t)T}\mathbf{x}_j\right) - a_r^2 \phi'\left(\mathbf{w}_r^{(0)T}\mathbf{x}_i\right)\phi'\left(\mathbf{w}_r^{(0)T}\mathbf{x}_j\right) \right| \\
&\leq \frac{1}{m}\left(\sum_{r=1}^m a_r^2\right) \max_{r\in[m]} \left| \phi'\left(\mathbf{w}_r^{(t)T}\mathbf{x}_i\right)\phi'\left(\mathbf{w}_r^{(t)T}\mathbf{x}_j\right) - \phi'\left(\mathbf{w}_r^{(0)T}\mathbf{x}_i\right)\phi'\left(\mathbf{w}_r^{(0)T}\mathbf{x}_j\right) \right| \\
&\leq \max_{r\in[m]}\left| \phi'\left(\mathbf{w}_r^{(t)T}\mathbf{x}_i\right)\left(\phi'\left(\mathbf{w}_r^{(t)T}\mathbf{x}_j\right) - \phi'\left(\mathbf{w}_r^{(0)T}\mathbf{x}_j\right)\right) \right| \\
&\quad + \left| \phi'\left(\mathbf{w}_r^{(0)T}\mathbf{x}_j\right)\left(\phi'\left(\mathbf{w}_r^{(t)T}\mathbf{x}_i\right) - \phi'\left(\mathbf{w}_r^{(0)T}\mathbf{x}_i\right)\right) \right| \\
&\leq \max_{r\in[m]} 2\alpha\beta\left\| \mathbf{w}_r^{(t)} - \mathbf{w}_r^{(0)} \right\|. \tag{98}
\end{aligned}
$$

Hence,

$$\left\| G^{(t)} - G^{(0)} \right\|_F \leq \max_{r\in[m]} 2L\beta n \left\| \mathbf{w}_r^{(t)} - \mathbf{w}_r^{(0)} \right\|$$

Since

$$\lambda_{\min}\left(\mathbf{G}^{(t)}\right) \geq \lambda_{\min}\left(\mathbf{G}^{(0)}\right) - \left\| \mathbf{G}^{(t)} - \mathbf{G}^{(0)} \right\|_F,$$

we have

$$\lambda_{\min}\left(\mathbf{G}^{(t)}\right) \geq \lambda_{\min}\left(\mathbf{G}^{(0)}\right) - \max_{r\in[m]} 2\alpha\beta n \left\| \mathbf{w}_r^{(t)} - \mathbf{w}_r^{(0)} \right\|.$$

Thus, for $\lambda_{\min}\left(\mathbf{G}^{(t)}\right) \geq \frac{1}{2}\lambda_{\min}\left(\mathbf{G}^{(0)}\right)$, we have

$$\max_{r\in[m]}\left\| \mathbf{w}_r^{(t)} - \mathbf{w}_r^{(0)} \right\| \leq \frac{1}{4\alpha\beta n}\lambda_{\min}\left(\mathbf{G}^{(0)}\right).$$

$\square$

# N    SOME BASIC FACTS ABOUT HERMITE POLYNOMIALS

For $\rho \in [-1, 1]$ we say that the Gaussian random variable $(v_0, v_1)$ is $\rho$-correlated if $(v_0, v_1) \sim \mathcal{N}\left(0, \begin{pmatrix} 1 & \rho \\ \rho & 1 \end{pmatrix}\right)$.

**Fact N.1** (Proposition 11.31 in O'Donnell (2014))**.**

$$\mathbb{E}_{(v_0,v_1)\,\rho\text{-correlated}}\, He_n\,(v_0)\, He_m\,(v_1) = \begin{cases} \rho^n & \text{if } n = m, \\ 0 & \text{otherwise.} \end{cases}$$

*where recall that $He_n$ denotes the degree-$n$ probabilists' Hermite polynomial given by* (10)*.*

The following fact follows immediately from the previous one.

**Fact N.2.** *For an activation function, define function $R : \mathbb{R} \to \mathbb{R}$ by*

$$R(\rho) := \mathbb{E}_{(v_0,v_1)\sim\,\rho\text{-correlated}}\, \phi(v_0)\phi(v_1).$$

*Then,*

$$R(\rho) = \sum_{a=0}^{\infty} \bar{c}_a^2\,(\phi)\,\rho^a,$$

*where $\bar{c}_a\,(\phi)$ is the $a$-th coefficient in the probabilists' Hermite expansion of $\phi$. The function satisfies the following two properties.*

- $\left|R(\rho)\right| \leq R(|\rho|)$,

- $R(\rho)$ *is increasing in* $(0, 1)$.

In the following we let $\Sigma := \begin{pmatrix} \rho_{00} & \rho_{01} \\ \rho_{01} & \rho_{11} \end{pmatrix}$.

**Lemma N.3.**

$$\mathbb{E}_{(v_0,v_1)\sim\mathcal{N}(0,\Sigma)}\, He_n\left(\frac{v_0}{\sqrt{\rho_{00}}}\right) He_m\left(\frac{v_1}{\sqrt{\rho_{11}}}\right) = \begin{cases} \left(\frac{\rho_{01}}{\sqrt{\rho_{11}\rho_{11}}}\right)^n & \text{if } n = m, \\ 0 & \text{otherwise.} \end{cases}$$

*Proof.* The proof follows from the proof of Fact N.1, by using the r.v. $(\tilde{v}_1, \tilde{v}_2)$, defined by

$$\tilde{v}_0 = \frac{v_0}{\sqrt{\rho_{00}}} \quad \tilde{v}_1 = \frac{v_1}{\sqrt{\rho_{11}}}$$

so that vector $(\tilde{v}_1, \tilde{v}_2) \sim \mathcal{N}\left(0, \Sigma'\right)$ where $\Sigma' := \begin{pmatrix} 1 & \left(\frac{\rho_{01}}{\sqrt{\rho_{00}\rho_{11}}}\right) \\ \left(\frac{\rho_{01}}{\sqrt{\rho_{00}\rho_{11}}}\right) & 1 \end{pmatrix}$. $\qquad\square$

**Lemma N.4.**

$$\mathbb{E}_{v\sim\mathcal{N}(0,\Sigma)}\, \phi\left(\frac{v_0}{\sqrt{\rho_{00}}}\right) \phi\left(\frac{v_1}{\sqrt{\rho_{11}}}\right) = \sum_{a=0}^{\infty} \bar{c}_a^2(\phi)\left(\frac{\rho_{01}}{\sqrt{\rho_{00}\rho_{11}}}\right)^a.$$

*Proof.*

$$\mathbb{E}_{v\sim\mathcal{N}(0,\Sigma)}\phi\left(\frac{v_0}{\sqrt{\rho_{00}}}\right) \phi\left(\frac{v_1}{\sqrt{\rho_{11}}}\right)$$

$$= \mathbb{E}_{v\sim\mathcal{N}(0,\Sigma)} \sum_{a=0}^{\infty} \sum_{a'=0}^{\infty} \bar{c}_a(\phi)\bar{c}_{a'}(\phi) He_a\left(\frac{v_0}{\sqrt{\rho_{00}}}\right) He_{a'}\left(\frac{v_1}{\sqrt{\rho_{11}}}\right)$$

$$= \sum_{a=0}^{\infty} \bar{c}_a^2(\phi)\left(\frac{\rho_{01}}{\sqrt{\rho_{00}\rho_{11}}}\right)^a = R\left(\frac{\rho_{01}}{\sqrt{\rho_{00}\rho_{11}}}\right)$$

where we use Lemma N.3 and Fact N.2 in the final step. $\qquad\square$

