# OpenReview forum: "Effect of Activation Functions on the Training of Overparametrized Neural Nets"
_ICLR.cc/2020/Conference — Accept (Poster)_

### Official Review · AnonReviewer2 · 2019-10-23
**Official Blind Review #2**

**Rating:** 8

**Review:**

The paper aims to characterize for different activation functions the minimum eigenvalue of a certain gram matrix that is crucial to the convergence rate for training over-parameterized neural networks in the lazy regime (small learning rate). On this front, the paper shows that for non-smooth activations the minimum eigenvalue is large under separation assumptions on the data improving on prior work. However, for smooth activations the authors show that the minimum eigenvalue can be exponentially small (or even 0 in case of polynomials) if the data has low-dimensional structure or the network is sufficiently deep. The authors experimentally validate the observations on synthetic data.

Overall, I vote to accept the paper. The paper does a thorough theoretical study of the behavior of the eigenvalues of the matrix corresponding to NTK which is crucial to the NTK analysis. The paper successfully makes the case for non-smooth activations versus smooth activations in the lazy regime. The authors use polynomial approximations and low-dimensionality in an interesting way to show an upper bound on the min eigenvalue for activations approximable by sufficiently low-degree polynomials. The paper is well written, self-contained and well-referenced.

Suggestions/Comments:
1. Please avoid referencing theorems in the appendix that do not have informal statements in the main paper. For example "We sketch the proofs of Theorem J.3, Theorem J.4 and Corollary J.4.1 showing that our results about the limitations of smooth activations are essentially tight when the data is smoothed.": Theorems/Corollary are not mentioned in the main text.
2. In real data as the authors point out, the dimension of the data is much larger than log n. In the setting of dimension being greater than log n, could you discuss how far the lower-bound from Theorem J.4.2 is from the upper bound from Theorem F.9. It would be useful to write it in similar notation.
3. The paper, in its current form, is long and probably hard for a general audience to parse. It would be useful to organize the appendix to emphasize the main techniques and ideas.

**Experience Assessment:**

I have published in this field for several years.

**Review Assessment: Checking Correctness Of Derivations And Theory:**

I assessed the sensibility of the derivations and theory.

**Review Assessment: Checking Correctness Of Experiments:**

I assessed the sensibility of the experiments.

**Review Assessment: Thoroughness In Paper Reading:**

I read the paper at least twice and used my best judgement in assessing the paper.

---

> ### Author Response · Authors · 2019-11-09
> **Response to Reviewer 2**
>
> Dear reviewer,
>
> Thank you very much for your constructive review and for your time and effort.
> We have uploaded a revised version.
>
> Responses to suggestions/comments:
>
> 1. We have changed the statement of Theorem 4.7 to be the informal statement of Cor. J.4.2 (Cor. I.4.2 in the revised version). We have dropped references to results whose informal statements do not appear in the main paper.
>
> 2. If the dimension of span of input is of the order n^{\gamma}, the lower-bound from Corollary J.4.2 (Cor. I.4.2 in the revised version) is approximately n^{-2/\gamma}, while the upper bound from Theorem F.9 is approximately (n^2 e^{-1/\gamma}). We will improve theorem statements to make comparisons easy.
>
> 3. We agree. Please see the first paragraph of our response to Reviewer #4.

---

### Official Review · AnonReviewer4 · 2019-11-03
**Official Blind Review #4**

**Rating:** 6

**Review:**

Summary: The authors of the paper examine how different activation functions affect training of overparametrized neural networks. They do their analysis in a general way such that it includes most activation functions such as ReLU, swish, tanh, polynomial, etc.

The main point of their analysis is that they examine a matrix called the G-matrix which is described in equation (2), and this (positive semi-definite) G-matrix can determine the rate of convergence to zero of the training error. Namely, the minimum eigenvalue of the G-matrix is inversely proportional to the time required to reach a desired amount of error (Theorem 3.1 (Theorem 4.1 from Du et al. (2019a)).

The main results separate into two cases: (1) activation functions with a kink (i.e. if the activation function is NOT in C^{r+1}, the space of r+1 continuously differentiable functions, for some finite r) and (2) smooth activation functions.

In the first case, the authors show that the minimum eigenvalue of the G-matrix is large, i.e. bounded away from zero after a few assumptions.

In the second case, the authors show that polynomial activations have many zero eigenvalues, and sufficiently smooth activations such as tanh or swish have many small eigenvalues, if the dimension of the span of data is sufficiently small.

The author’s initial problem setup works on a one-hidden-layer neural network where only the input layer is trained, but provide some extensions in the appendix.

The authors also provide some empirical experiments: one synthetic data, and on CIFAR10. The synthetic data experiments agreed with theory, but the experiment on CIFAR10 did have some gap between theory and experiment, although the CIFAR10 with ReLU experiment agreed with theory.


Stengths: I appreciate the author’s effort in providing needed theoretical analysis on how activation affects training error for deep neural networks. The authors also provide an extensive appendix that provide seemingly full proofs of the theorems (although this reviewer did not go into detail for most of the appendix). The authors also provide experiments that confirm the theory and also provide examples highlighting the gap (which this reviewer sees as a strength).


Weaknesses: A clear weakness of this paper is that the appendix is too long. The authors do provide a proof sketch of the main results and refer to the appendix, but I would have liked to have seen a more focused paper. Having extensions of the main results is nice, but it’s sometimes unclear what is being extended. Perhaps a list of extension would make clear what’s in the appendix.


Other comments: (i) I’d like to an explanation to why it’s called the DZXP setting.
(ii) On page 4, when explaining the M matrix, I think it should be grad_W F (instead of L)

(iii) On page 4, I think there is more to assumption 1, after looking at Allen-Zhu et al. (2019) (https://arxiv.org/pdf/1811.03962.pdf page 4, footnote 5)

(iv) the wording from page 4, “the matrix G^(t) does not evolve from its initial value G^(0)” is a bit awkward. Do you mean G^(t) does not change much from G^(0), and as to goes to infinity then G^(t) goes to G^(0)?

**Experience Assessment:**

I have read many papers in this area.

**Review Assessment: Checking Correctness Of Derivations And Theory:**

I assessed the sensibility of the derivations and theory.

**Review Assessment: Checking Correctness Of Experiments:**

I assessed the sensibility of the experiments.

**Review Assessment: Thoroughness In Paper Reading:**

I read the paper at least twice and used my best judgement in assessing the paper.

---

> ### Author Response · Authors · 2019-11-09
> **Response to Reviewer 4**
>
> Dear reviewer,
>
> Thank you very much for your constructive review and for your time and effort.
>
> We share your concern about the appendix being too long. We have added a table of contents (at the beginning of the appendix) to make it easier to navigate. For better organization, we have slightly changed the order of some sections (previous Sec. I (LOWER BOUND ON LOWEST EIGENVALUE FOR NON-SMOOTH FUNCTIONS) is now Sec. K) and improved section names. We are sure there are many more ways of further improving the readability of the paper (e.g. by adding more explanations in some proofs) and are working on it.
>
> Having said that, it seems to us that your specific concerns are already addressed in the paper as we now explain. We think a lengthy appendix is unavoidable largely due to the nature of the results; but perhaps the length issue is mitigated somewhat as we have tried to make the paper easy to navigate (e.g., the main paper provides a roadmap of the appendix). For focused presentation, we have highlighted the simplest case of one-hidden layer networks where only the input layer is trained so that the main ideas are clear. Large part of the paper is confined to this case. Section 5 on extensions provides a list of extensions along with pointers to the sections in the appendix for the results proven in the paper and mentions some others that are not proven. Please correct us if we have misunderstood your concerns. We welcome any further suggestions that might help improve readability further.
>
> It is perhaps relevant to mention here that another reason for the length of the paper is that reviewers of a previous shorter version of this submission asked for proofs of extensions and related results that we mentioned without proof. While the extension to SGD and one or two other results turn out to be straightforward adaptation of previous work, others are more substantive and together try to present a reasonably complete picture.
>
> We have uploaded a revised version.
>
> Responses to minor concerns:
>
> (i) This was an error on our part. DZXP should have been DZPS after the authors of Du et al. 2019a [1]. Fixed in the revised version.
>
> (ii) We have made the suggested change in the revised version.
>
> (iii) Our Assumptions 1 and 2, while not identical to those of Allen-Zhu et al. [2], are essentially equivalent. Our Assumption 2 is about the lower bound on angles between the data points, and Assumption 2.1 in Allen-Zhu et al. [2] is about the lower bound on the distance between data points. The requirement of the final coordinate being 1/sqrt(2) in footnote 5, page 4 of their paper, along with their Assumption 2.1 leads to the lower bound on angles. Thus our statements make the role of angles more explicit. We have added some clarification in our paper.
>
> (iv) We agree and have amended the wording in the new version from "does not evolve from its initial value" to "remains close to its initial value". Your interpretation is correct.
>
> If our responses satisfactorily address the weaknesses you mentioned, we hope that you would consider raising your score.
>
>
> [1] Simon S. Du, Xiyu Zhai, Barnabas Poczos, and Aarti Singh. Gradient descent provably optimizes over-parameterized neural networks. In ICLR, 2019.
> [2] Allen-Zhu, Zeyuan, Yuanzhi Li, and Zhao Song. A convergence theory for deep learning via over-parameterization.  arXiv preprint arXiv:1811.03962 (2018).

---

### Decision · Program_Chairs · 2019-12-19

**Decision:**

Accept (Poster)

**Comment:**

The article studies the role of the activation function in learning of 2 layer overparaemtrized networks, presenting results on the minimum eigenvalues of the Gram matrix that appears in this type of analysis and which controls the rate of convergence. The article makes numerous observations contributing to the development of principles for the design of activation functions and a better understanding of an active area of investigation as is convergence in overparametrized nets. The reviewers were generally positive about this article.